# Context value updating and multidimensional neuronal encoding in the retrosplenial cortex

Weilun Sun [1,2,8], Ilseob Choi[1,2,8], Stoyan Stoyanov [1], Oleg Senkov [1], Evgeni Ponimaskin [3], York Winter [4], Janelle M. P. Pakan [2,5,6✉] & Alexander Dityatev [1,2,7✉]

The retrosplenial cortex (RSC) has diverse functional inputs and is engaged by various sensory, spatial, and associative learning tasks. We examine how multiple functional aspects are integrated on the single-cell level in the RSC and how the encoding of task-related parameters changes across learning. Using a visuospatial context discrimination paradigm and two-photon calcium imaging in behaving mice, a large proportion of dysgranular RSC neurons was found to encode multiple task-related dimensions while forming context-value associations across learning. During reversal learning requiring increased cognitive flexibility, we revealed an increased proportion of multidimensional encoding neurons that showed higher decoding accuracy for behaviorally relevant context-value associations. Chemogenetic inactivation of RSC led to decreased behavioral context discrimination during learning phases in which context-value associations were formed, while recall of previously formed associations remained intact. RSC inactivation resulted in a persistent positive behavioral bias in valuing contexts, indicating a role for the RSC in context-value updating.

[1] Molecular Neuroplasticity, German Center for Neurodegenerative Diseases (DZNE), Magdeburg, Germany. [2] Center for Behavioral Brain Sciences (CBBS), Magdeburg, Germany. [3] Department of Cellular Neurophysiology, Hannover Medical School, Hannover, Germany. [4] Institute for Biology, Humboldt University, Berlin, Germany. [5] Institute of Cognitive Neurology and Dementia Research, Otto-von-Guericke University, Magdeburg, Germany. [6] German Center for Neurodegenerative Diseases (DZNE), Magdeburg, Germany. [7] Medical Faculty, Otto-von-Guericke University, Magdeburg, Germany. [8] These authors contributed equally: Weilun Sun, Ilseob Choi. ✉email: janelle.pakan@med.ovgu.de; Alexander.Dityatev@dzne.de

Animals must successfully navigate through their environment, assign value to behaviorally relevant locations, and later flexibly update these context-value associations. This context-specific behavior requires input from both sensory and motor systems, spatial awareness, and memory formation. Hence, these processes will inevitably involve multiple interacting brain regions, with information from various systems being constantly integrated and updated. In this regard, the retrosplenial cortex (RSC) is ideally positioned to integrate both spatial and contextual information from the environment[1,2], since it is densely connected with the hippocampal and parahippocampal formation, as well as visual, posterior parietal, prefrontal, and entorhinal cortices[1,3,4]. Indeed, the RSC has been shown to be essential in spatial navigation[5–9] and contextual associations[10,11]. Additionally, the RSC plays a role in visualizations of future actions and scene processing in humans[1,12], with recent rodent studies revealing the involvement of the RSC in similar cognitive functions[13–20]. Although anatomical and behavioral studies strongly suggest an integrative role for the RSC[21,22], how this is functionally implemented at the cellular level is unknown. This is important as the precise cellular dynamics may offer a significant computational advantage for a region with such diverse input-output functions.

Previous studies of single-cell activity in mammalian RSC have revealed that specific subpopulations of RSC neurons can encode spatial information[23–25], possess properties of head direction cells[26,27] or place cells[28], exhibit responses to visual properties[29–32], including representations of visual landmarks[18,20,33] and can encode reward locations[34,35] as well as reward history[36]. Therefore, it is clear that RSC neurons exhibit diverse functional responses; however, the extent to which these inputs are integrated on the single-cell level, and if this offers a specific advantage for particular learning tasks, remains unknown. Although these dynamics are not clear in the RSC, single-cell responses tuned to multiple task-related parameters are prevalent in the prefrontal cortex, predictive of behavior, and thought to specifically offer a computational advantage to support complex cognitive tasks and flexibility[37–40]. A traditional way of probing cognitive flexibility, or the ability to rapidly change behavior in the face of changing circumstances, is to implement reversal learning within a behavioral paradigm, which serves to evaluate adaptive responses when stimulus-outcome associations are altered[41]. Importantly, this cognitive capacity is also disrupted in many neurological disorders[42–44].

Here, we establish a simple visuospatial training paradigm with multiple task-related parameters to investigate context-discrimination behavior and the associated contribution of integrative neuronal responses to the encoding of information following learning and reversal learning. We performed in vivo two-photon calcium imaging in the dysgranular RSC in head-fixed mice navigating in a virtual reality (VR) environment while mice learned to discriminate between virtual contexts by associating a water reward with a specific location in a particular visually defined context. With training, mice demonstrated stereotyped running behaviors in relation to the rewarded context. We evaluated the capacity of individual RSC neurons to encode information crucial for spatial navigation and context discrimination, such as the animal's position and speed, as well as the contextual identity, including its visual features and the associated reward-value. Interestingly, the RSC contained a large proportion of multidimensional neurons encoding multiple task-related parameters: context, position, and speed. Their fraction increased after reversal learning, suggesting these multi-dimensional neurons may play a role in updating context-value associations and cognitive flexibility. We further verified the importance of the RSC by demonstrating that chemogenetic inactivation during learning and reversal learning impaired

behavioral context discrimination but recall of previously formed context-value associations remained intact.

## Results

**Context discrimination in a virtual environment.** To examine the neuronal dynamics underlying context discrimination and context-value associations in the RSC, Thy1-GCaMP6f transgenic mice were implanted with cranial windows, head-fixed, and pretrained to reliably run on an air-cushioned spherical treadmill (Fig. 1a, b). We then performed two-photon Ca$^{2+}$ imaging of the expressed genetically encoded calcium indicator GCaMP6f in the dysgranular RSC in layer 2/3 (Fig. 1d–f) while mice performed a contextual discrimination task in a virtual environment (Fig. 1b, c). Animals were presented with three different contexts across a series of days, which were defined by the parameters of the virtual environment—including the spatial properties of the linear corridor and the visual pattern present along the length of the corridor. Each context had identical geometry (2-meter-long linear corridor) but unique visual patterns on the virtual corridor walls (Fig. 1c). Following an initial baseline imaging day, a specific context was associated with a water reward that was given at a fixed location (180 cm from the beginning of the corridor). Changes in fluorescence were imaged (i.e., $\Delta F/F$) and the corresponding position and speed of each animal within the VR environment were recorded (Fig. 1g).

First, we validated that mice can learn to discriminate between the different virtual environmental contexts. The aim was to establish an experimental paradigm that would passively, but consistently, pair a particular context with a reward, thereby assigning value to a specific context through associative learning —as may occur when animals are navigating through an environment and discover a food source at a particular location. Therefore, rewards were given to the animals consistently at a default location in a particular context without the need for the animal to perform an extraneous behavior to receive the reward. Within each experimental day, this context is then inseparable from the parameters of the virtual environment (e.g., visual pattern on the corridor walls) and its behaviorally relevant value; however, on subsequent days a different context is paired with reward (i.e., reversal learning) to assess changes in context-value associations and to dissociate the effects of the reward-association from responses specific to the visual pattern along the virtual corridor (Fig. 1i).

This training paradigm has the benefit that learning occurs quickly, relative to when specific operant behaviors need to be trained (often taking days to weeks) but with the drawback that there is no explicit measure of success rate. However, we found that after an initial decrease in speed across all contexts, animals naturally increased their speed in the non-rewarded context (i.e., context 2) and decreased their speed selectively in the rewarded context (i.e., context 1), presumably in preparation to consume the water reward ($n = 5$ mice, Figs. 1h, j, 2a, and Supplementary Fig. 1). This change in speed across contexts was maximally represented in the 10 cm before the water reward was given, in the reward 'anticipation zone' (AZ), and was consistently reproduced across experimental groups (Fig. 2a and Supplementary Fig. 1). Additionally, we found consistent speed trajectories across trials, with mice starting to show alterations in running speed after the first ~40 cm along the corridor (Fig. 1j). Using linear discriminant analysis, we predicted the occurrence of a reward at the end of a trial from the mean speed in the anticipation zone for all rewarded and non-rewarded trials (contexts 1 and 2) across both learning and reversal learning. The accuracy of prediction was $84.8 \pm 1.1\%$ ($n = 5$), which was significantly higher than when the mean speed along the entire

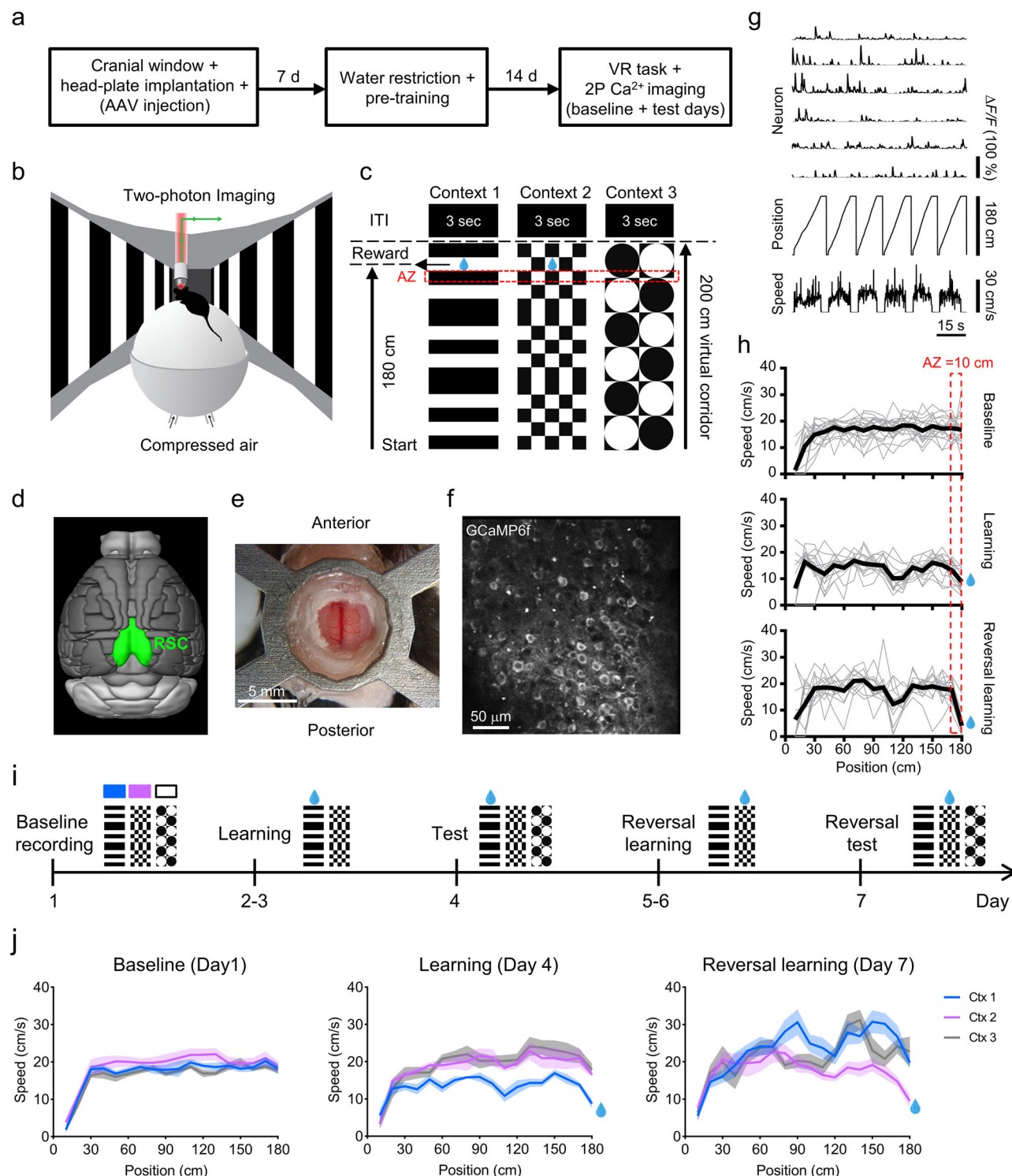

virtual corridor was used (71.9 ± 4.4%, $n = 5$, $p = 0.022$). We also found a clear shift in the distribution of speeds within the anticipation zone on a trial-by-trial basis across various learning stages (Fig. 2b) as well as within learning sessions (Supplementary Fig. 2). Although mice slowed down in the anticipation zone across all contexts following training (due to the enforced stationary period at the end of each trial), this anticipation zone represents the culmination of the difference in speeds across the length of the corridor (Fig. 1j, Supplementary Fig. 2). Therefore, in this type of passive training paradigm that has no explicit measure of success rate per se, we found that this change in speed

along the virtual corridor, culminating in a maximal difference between rewarded versus non-rewarded contexts at the anticipation zone, was a reliable behavioral metric of learning and context discrimination.

Using this behavioral metric, we then examined how mice perform during reversal learning. Here, the reward-association was switched from context 1 to context 2 while the neutral context (context 3) was not presented during training days and, hence, remained neutral on the test day (day 7). In the reversal task, mice need to additionally devalue the previously rewarded context (context 1) and then form a new reward association with

**Fig. 1 Two-photon calcium imaging in the retrosplenial cortex (RSC) of head-fixed mice during a context-discrimination paradigm in a virtual environment. a** Experimental timeline for chronic calcium imaging of RSC neurons. **b** Schematic of the virtual reality (VR) system for two-photon imaging in head-fixed awake behaving mice. Movement of the spherical treadmill is recorded via a rotary encoder and the VR environment is displayed on a surrounding monitor system covering ~270° of the horizontal visual field. **c** Three 200 cm virtual contexts were created with different visual patterns on the virtual corridor walls and a sucrose water reward (blue drop) was given at a fixed reward position (180 cm) either in context 1 or in context 2. The intertrial interval (ITI) consisted of 3 s of black screens. In both the reward period and the ITI the treadmill was stationary with external brakes applied. The reward anticipation zone (AZ; indicated in red) was defined as 10 cm before the reward position. **d** Schematic showing the site for imaging highlighted in green (dysgranular RSC), derived from the Brain Explorer® 2 software based on the Allen Mouse Brain Atlas (https://mouse.brain-map.org/static/brainexplorer). © 2015 Allen Institute for Brain Science. Allen Brain Atlas API. Available from: brain-map.org/api/index.html. **e** Illustration of the metal head-plate for fixation and example cranial window. **f** Example image of GCaMP6f-expressing neurons in layer 2/3 of the RSC in a Thy1-GCaMP6f transgenic mouse. This experiment was repeated independently in 5 mice and 3 sessions for each mouse with similar results. **g** Changes in fluorescence ($\Delta F/F$) signals were synchronized with recordings of position and speed in the virtual environment. **h** Running speed from one example animal as a function of position in the VR environment indicating the speed of each trial (gray) and the mean speed of all trials (black) for all contexts on baseline day (top) and the rewarded context during learning (middle) and after reversal learning (bottom); reward anticipation zone is indicated in red. **i** Experimental timeline. Contexts 1–3 (Ctx 1–3) are indicated by their visual patterns and associated color legend. Rewarded context is indicated by a blue drop. **j** Speed trajectories averaged across animals ($n = 5$ mice), shown for each context and across learning phases. Solid lines indicate the mean speed across animals and shading indicates the corresponding SEM. Source data are provided as a Source Data file.

the previously non-rewarded context (context 2). Similar to the initial learning progress, we found that even on the second day of exposure to the new task, mice already showed significantly slower running speed for the newly rewarded context (i.e., context 2) compared to the non-rewarded context (i.e., context 1; Fig. 2a; see Supplementary Fig. 2). By the testing day (day 7), mice again showed significantly slower running speeds in the rewarded context when compared to baseline days and the end of the initial learning phase, and the same faster running speed in both the non-rewarded and neutral contexts (Fig. 2a, c). Therefore, mice altered their running behavior during this reversal learning task, indicating that they devalued the previously rewarded context to match the neutral context, and flexibly formed a new reward association with the previously non-rewarded context.

**Decoding of context and context-value associations from neuronal activity in the RSC.** Since the RSC is known to play a role in various task-related elements, such as contextual discrimination[35], spatial navigation[25,28], and the persistent encoding of value-related signals during learning[36], we investigated the underlying activity in the dysgranular RSC during this learning and reversal learning paradigm to determine how information related to these combined factors was represented within the RSC. We divided each trial within the virtual environment into task-related periods (Fig. 2d). First, the period when the visual context was displayed and animals had to run along the virtual corridor to reach the end (termed corridor). Next, the period where external breaks were applied (and therefore the treadmill was blocked so all animals were stationary), the reward spout was extended for all trials, a water reward was dispensed following only the appropriate context and animals consumed the reward, and the reward spout was retracted for all trials (collectively termed the reward period). Finally, a 3-second intertrial-interval (ITI) period for all trials, where the treadmill remained blocked and the VR screens were disabled, resulting in a dark environment; note that this ITI period was consistent across all trial types with no systematic behavioral differences during this period between rewarded and non-rewarded trials.

We first examined the neuronal activity along the virtual corridor and found that RSC cells displayed heterogeneous response patterns for both the rewarded and the non-rewarded context, with no difference in the average activity across contexts (Fig. 2e, Supplementary Fig. 4a). This was observed during both the learning phase (day 4, $p = 0.958$) and after the specific rewarded context was switched in reversal learning (day 7, $p = 0.088$; averaged responses across trials, Fig. 2e; see also

Supplementary Fig. 6). To determine how the context (i.e., parameters of the virtual environment) and the context-value associations (i.e., specific context linked to reward, no reward, or neutral value associations) were represented in this RSC activity, we examined the success of decoding the individual contexts at baseline (day 1) and the change in context-value associations formed with learning (day 4) and after reversal learning (day 7) across trials. We used this decoding accuracy as a metric to quantify the extent to which external and task-related variables are represented by the neuronal population activity within the RSC. To do this, backward stepwise linear discriminant analysis was employed to select the linear model with a minimal number of neurons (see methods). We found that while decoding accuracy was high for all contexts (>80%), when we neuronal activity along the virtual corridor there were no systematic differences in context-value decoding accuracy between the rewarded or non-rewarded contexts following learning or reversal learning, either within or across days (Fig. 3a). Although, we did find that the neutral context had lower decoding accuracy on the baseline day and on day 7 (see below). Additionally, we found no change in the overall proportion of cells that significantly contributed to the prediction of which context the animal was in ($p < 0.05$; i.e., context-encoding cells), which was also independent across days (~70%; Fig. 3b) and, hence, did not differ across learning stages with altered specific context-value associations.

We did observe a minor drop in accuracy for decoding the neutral context (context 3; on days 1 and 7; Fig. 3a). This may reflect differences in the spatial organization of visual features in context 3, which contains fewer dark-light transitions, indicating that the RSC responds strongly to fundamental visual features of the VR environment (see Supplementary Figs. 3 and 8). Previous studies have shown that the RSC receives substantial inputs from the visual cortex[29,32] and some neurons in the RSC reliably respond to visual input, e.g., to moving gratings with a specific orientation[29,30,45]. Therefore, the generally high accuracy of context encoding in RSC neurons along the corridor may result largely from responses to the visual properties of the corridor walls. In order to more directly separate out the responses of individual neurons to the specific visual pattern of the virtual environment versus the context-value associations, we followed the neuronal responses for a subset of neurons longitudinally across days 1, 4, and 7. Using principal component analysis we found that representations of neuronal activity during both learning phase and context-value associations were separable (e.g., Supplementary Fig. 5a). Therefore, we further performed

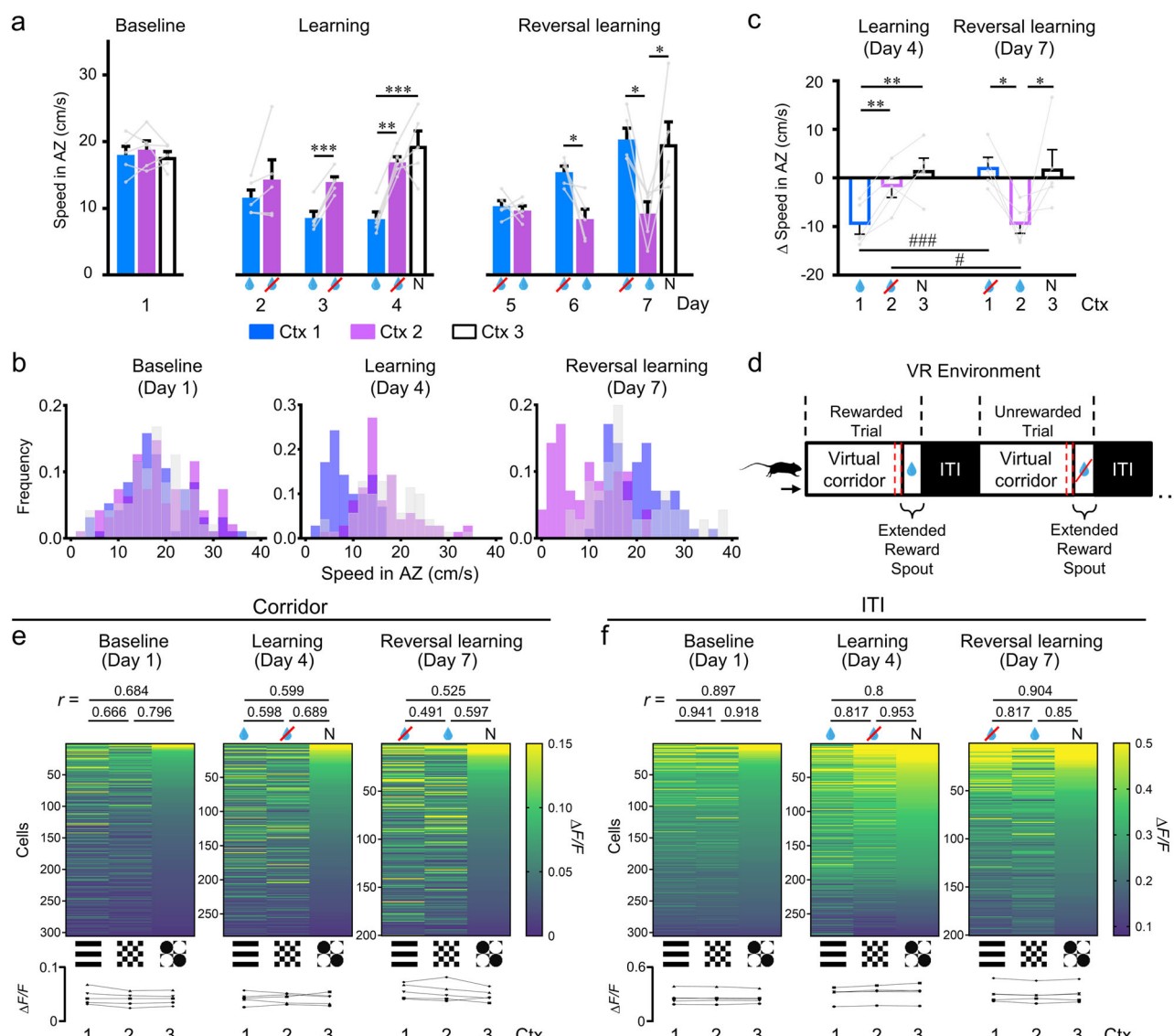

**Fig. 2 Mice exhibit context discrimination by differentiated running speed and distinct patterns of neuronal activity in dysgranular RSC neurons across rewarded and non-rewarded contexts. a** Mean running speed in the anticipation zone across days and learning phases. Rewarded context (blue drop); Non-rewarded context (drop with red cross); Neutral context (N). **b** Distribution of speeds within the anticipation zone on a trial-by-trial basis across learning phases (baseline-day 1, learning-day 4, reversal learning-day 7). **c** Change in speed relative to baseline for learning (day 4–day 1) and reversal learning (day 7–day 1). **d** Schematic of the trial-by-trial task. The virtual corridor is presented with a specific contextual pattern on the corridor walls (Ctx 1–3) followed by a zone where mice get rewarded or not depending on the specific context, then the ITI consists of a black screen for 3 s. In both the reward period and the ITI the treadmill was stationary with external brakes applied. The anticipation zone is indicated as a red dashed area. **e** Mean $\Delta F/F$ for each neuron averaged along the length of the virtual corridor for each context (Ctx 1–3) and across learning phases (baseline-day 1, learning-day 4, reversal learning-day 7). Cells are sorted for each day according to mean $\Delta F/F$ for the neutral (N) context. Line plots at the bottom show mean $\Delta F/F$ across the entire population of cells for each context. Cross-correlations (Spearman $r$ value) of neuronal activity between contexts are reported on top of the color maps. **f** Same as **e** but during the ITI. For all, $n = 5$ mice. For **a** and **c**, data are presented as mean values + SEM, and the data points are shown as gray dots with the data from the same animals linked by gray lines. For **a**, one-way RM ANOVA and post-hoc Holm–Sidak test were applied for the days with three contexts, and paired $t$-test was used for the days with two contexts, two-sided. For **c**, two-way RM ANOVA and post-hoc Student–Newman–Keuls tests were used. Within-day comparisons: *$p < 0.05$, **$p < 0.01$, ***$p < 0.001$. Exact $p$-values can be found in Supplementary Dataset 1. Source data are provided as a Source Data file.

linear discriminant analysis to test the decoding accuracy for the context value from neuronal responses, i.e., % of correct prediction of the rewarded context (context 1 on day 4, and context 2 on day 7) in relation to all other non-rewarded contexts across all learning phases. We focused on responses along 40–180 cm of the virtual corridor, which corresponds to the length of the virtual environment where we found the divergence

of speed trajectories in the running behavior across contexts with training (see Fig. 1j, Supplementary Fig. 2). We found that the decoding accuracy of the rewarded context-value association was significantly higher for observed compared to shuffled data ($83.7 \pm 3.5\%$ vs $55.5 \pm 1.2\%$, $n = 4$; $p = 0.006$; see Supplementary Fig. 5c). Thus, a large proportion of the RSC neuronal population, even in naïve animals, showed context-specific patterns of activity

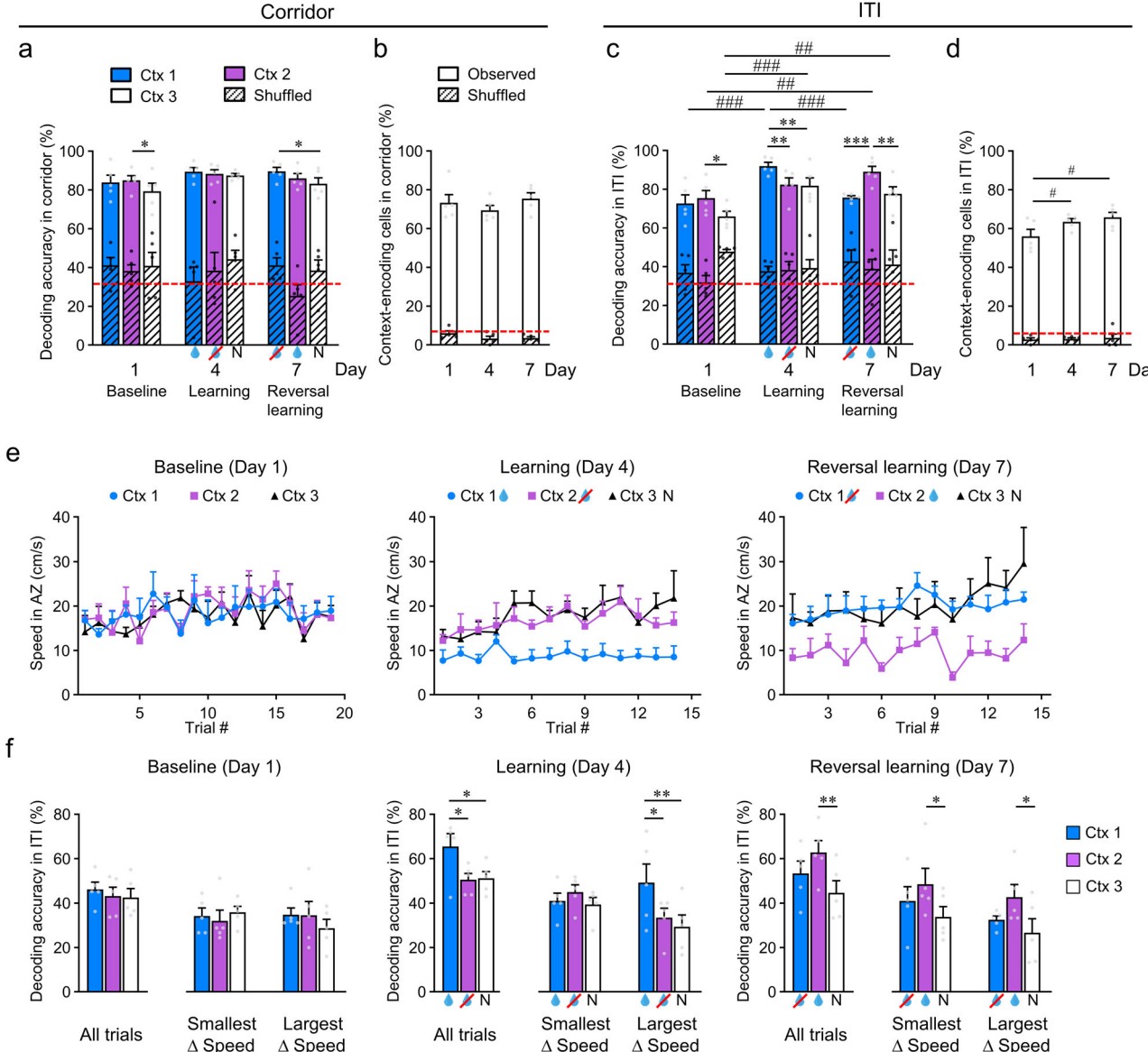

**Fig. 3 Context-decoding accuracy during the intertrial interval (ITI) is higher for rewarded contexts across learning phases and shows intertrial variability correlating with within-session changes in running speed across contexts. a** Decoding accuracy for each context along the virtual corridor (0–180 cm), across learning phases. Results after random shuffling of the raw data (diagonal stripes) and red dashed line indicating the chance level (33.3%) are shown. **b** Proportion of context-encoding cells from responses along the virtual corridor across learning phases. **c** Same as **a** but during ITI. **d** Same as **b** but during ITI. **e** Within session trial-by-trial mean speed in the anticipation zone for each context across learning phases, averaged across animals. Rewarded context (blue drop), non-rewarded context (drop with red cross), and neutral context (N). **f** Decoding accuracy within the ITI after deconvolution across learning phases for all trials and for the two trials with the smallest and the two trials with the largest difference in speed between the rewarded and the non-rewarded context. For all, $n = 5$ mice and data are presented as mean values + SEM. For **a–d** and **f**, experimental data points are shown as gray dots and shuffled data points are shown as black dots, two-way RM ANOVA and post-hoc Holm–Sidak tests were used, *$p < 0.05$, **$p < 0.01$. Within-day comparisons: *$p < 0.05$, **$p < 0.01$, ***$p < 0.001$; between-day comparisons: #$p < 0.05$, ##$p < 0.01$, ###$p < 0.001$. Exact $p$-values can be found in Supplementary Dataset 1. Source data are provided as a Source Data file.

along the virtual corridor and allowed for accurate discrimination of all virtual environments. Moreover, the response properties of individual neurons across days could be predictive of context-value associations across learning phases during periods when behavior also diverged across contexts.

Interestingly, we also found that when we performed the linear discriminant analysis across days using longitudinal neuronal responses within the ITI period, the decoding accuracy for this reward association was significantly higher than shuffled data (72.3 ± 3.0% versus 60.6 ± 1.1%, $n = 4$; $p = 0.025$; see Supplementary Fig. 5b, c). Within this ITI period animals no longer saw the

context-specific visual stimulus, but rather a black screen; therefore, it is possible that the context-value-related information may be less obscured by activity related to visuospatial information on the population level. Indeed, when we examined the pattern of neuronal activity in the ITI period, we found a clear difference in the pattern of activity between the rewarded and non-rewarded contexts (Fig. 2f), which switched with reversal learning; revealing a change in activity beyond responses to the individual contexts (or virtual environments) alone and in relation to context-value associations across days. Here the activity of individual neurons was more similar for contexts that

were non-rewarded and neutral, and differed for the context which was rewarded (Fig. 2f). We cross-validated this pattern of activity by comparing odd and even trials, which resulted in comparable patterns (Supplementary Fig. 4b) and quantified the relationship across contexts using cross-correlation (Supplementary Fig. 7). We found that in the ITI period, the non-rewarded and neutral contexts were more significantly correlated to each other than they were to the rewarded context (Fig. 2f, Supplementary Fig. 7, see the alternate pattern of responses along the virtual corridor in Fig. 2e, Supplementary Fig. 6). This was also reflective of the behavioral changes in running speed observed across learning phases, in that the non-rewarded and neutral contexts were not significantly different from each other, but both were significantly different to the rewarded context (see Fig. 2c). Accordingly, we also found a higher decoding accuracy for the rewarded context (Fig. 3c) during the ITI period as well as a significantly higher proportion of context-encoding cells (Fig. 3d), following both learning (day 4) and reversal learning (day 7), indicating a change in decoding accuracy with the change in context-value associations across days.

Due to the nature of calcium imaging (i.e., decay-time kinetics of GECIs), it is possible that these reward-related responses in the ITI period could be remnants of simple reward-onset evoked neuronal activity in the preceding reward period. However, we found no difference in the average activity across all cells between the rewarded, non-rewarded, or neutral contexts (Fig. 2f). Since reward responses have been shown to be potentiating in the RSC[18,34,35] as well as in many other connected brain regions[46–48] one would expect an increase in the average $\Delta F/F$ for the rewarded context if this activity was simply a residual response to the reward-onset itself. Further, when we used deconvolution methods in the ITI to remove potential effects of the remnant calcium decay signal (Fig. 3f; Supplementary Figs. 10, 11), the pattern of results in the ITI period was consistent with the non-deconvolved data. We sorted the average activity of each neuron according to their responses during the reward period and found that the activity within the reward period is only moderately correlated to that within the ITI period, and only increases in the later trials (Supplementary Figs. 9, 10). Since mice displayed a within-session learning curve on day 4 according to our behavioral metric (Fig. 3e), we compared the context-specific decoding accuracy on trials with the smallest speed difference between contexts ('worst' performance) versus trials with the largest speed difference between contexts ('best' performance). We found that the worst performance trials showed no significant difference in decoding accuracy across contexts based on the neural activity in the ITI period. However, for the best performance trials, the rewarded context had significantly higher decoder accuracy compared to the non-rewarded and neutral contexts (Fig. 3f). Interestingly, on day 7, the now highly trained animals showed a consistent difference in speed across contexts beginning from the first trials (Fig. 3e) and accordingly we did not detect significant changes in decoding accuracy between trials (Fig. 3f). Hence, if the behavioral performance differed between contexts, the RSC neuronal activity consistently resulted in significantly higher decoding accuracy for the rewarded context. Therefore, the pattern of RSC neuronal activity within the ITI period was reflective of our behavioral readout of context discrimination and led to increased decoding accuracy of the behaviorally relevant context compared to non-rewarded contexts within this delay period following each trial.

Since there was no systematic difference in context-value association decoding across days along the virtual corridor, this processing within ITI was also not just a straightforward reflection or 'replay' of the previous corridor activity (see also Supplementary Fig. 9). Further, since mice received the reward in context 1 across all trials on day 4, independent of behavioral performance, the difference in the RSC decoding accuracy across contexts (Fig. 3f) represents more than simple reward-evoked dynamics. Therefore, it is plausible that this ITI activity reflects a more complex post-trial processing of multiple task-dependent parameters. Hence, we also examined the activity of RSC neurons in relation to other task dimensions, namely spatial position along the virtual corridor and speed.

**Decoding of spatial information from neuronal activity in the RSC.** The RSC has been shown to contain both place cells and head direction cells, which play a role in spatial navigation[26–28]. Therefore, we examined the relationship between RSC neuronal activity and the animal's position along the virtual corridor in different contexts. We observed that some neurons were preferentially active at one or more spatial positions during each trial (Fig. 4a). We used a linear model to decode the animal's position in each of eighteen 10 cm-long spatial bins using the RSC neuronal activity for each animal. Backward stepwise linear regression analysis was used to select the model with a minimal number of cells. The linear model provided accurate decoding of the animal's position in a given context, as illustrated by a linear relationship between the predicted and observed values of position (Fig. 4b). On average, $R^2$ (which is the measure of model fit, i.e., the fraction of position variance explained by neuronal activity) was ~0.50 across all contexts (Fig. 4c; compared to ~0.1 for shuffled data) with no significant advantage for the rewarded context across days. We found that ~30% of neurons showed a significant contribution ($p < 0.05$) to the encoding of positional information in each context, independent of day and reward association (Fig. 4d; compared to ~10% for shuffled data). Therefore, position-related responses were not affected by reward-associations and, on the population level, remained consistent across learning.

Although there was no specific change in responses with rewarded/non-rewarded contexts, we did find that a large portion (~40%) of position-encoding cells only encoded position in a single context, while ~20% contributed to encoding in two contexts, and a small proportion (~5%) showed position encoding in all three tested contexts (Fig. 4e). Therefore, many neurons in the RSC demonstrated the potential for encoding spatial information in a context-specific manner, and these responses were largely unaffected by the behavioral relevance, or context-value association, of the particular context. In fact, we found a significant increase in the decoding accuracy ($R^2$) across days only for the neutral context (context 3), which started off significantly lower on the baseline day in comparison to the other two contexts—this may have been due to the less salient edges associated with this visual pattern (see Supplementary Fig. 8). Therefore, even though position encoding in the RSC was not specifically altered with reward, it may still be refined in an experience-dependent manner for less salient spatially defined environments.

**Decoding of speed information from neuronal activity in the RSC.** Previous studies have suggested that RSC neurons might also encode speed selective signals[22,26,49] and we found that running speed was a reliable behavioral metric of learning during the context discrimination task; therefore, we investigated whether the underlying neuronal activity in the RSC was related to the animal's speed. Indeed, we observed that in some cells neuronal activity was specifically modulated by speed (e.g., Fig. 4f). When we used a linear model to decode the animal speed based on the neuronal activity, we found a linear relationship between predicted and observed values of speed (Fig. 4g). On average, $R^2$

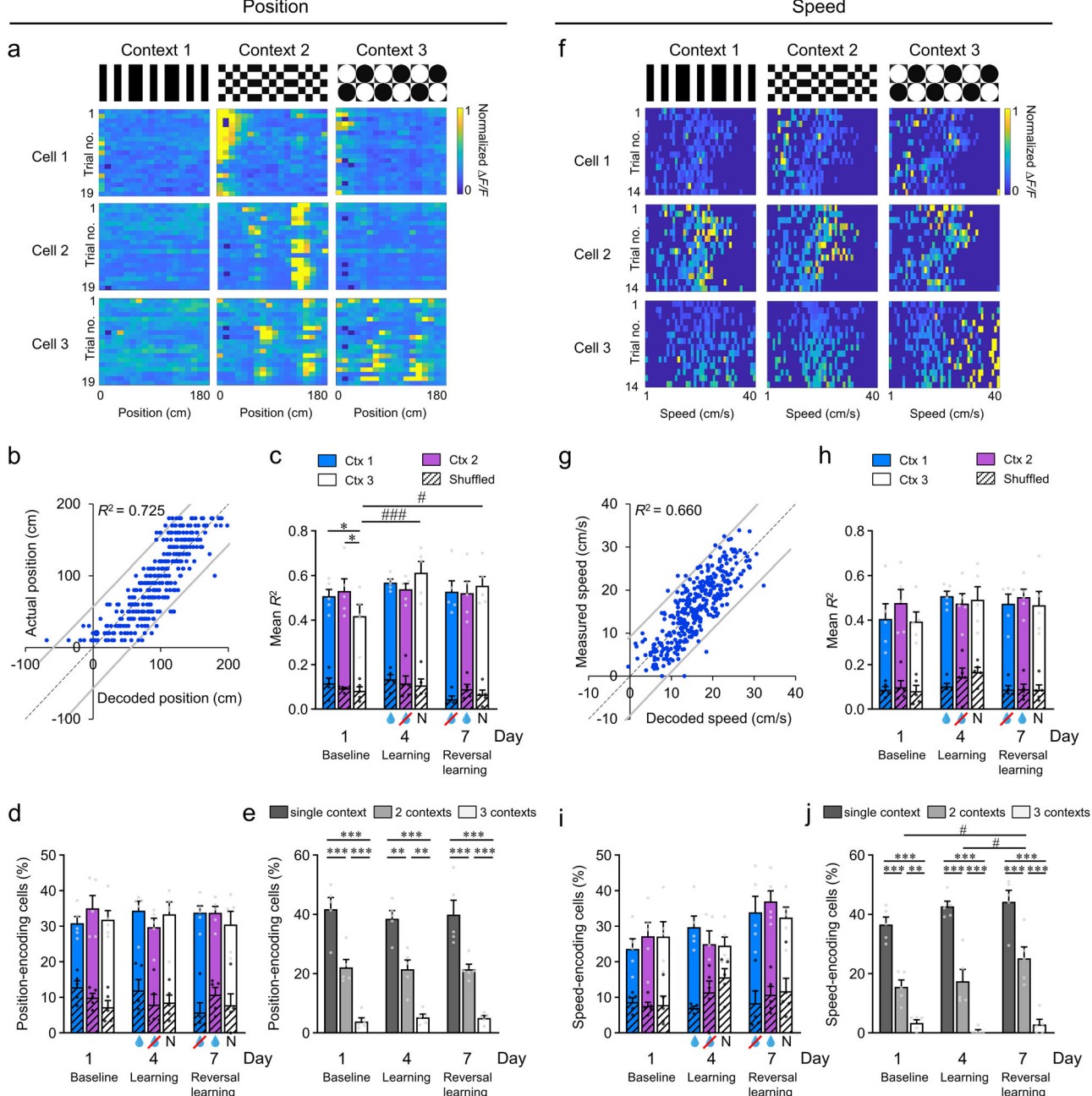

**Fig. 4 Neurons in the dysgranular RSC encode position-related and speed-related information. a** Normalized response ($\Delta F/F$) at positions along the virtual corridor (0–180 cm) of three position-encoding neurons across multiple trials. **b** Example linear regression plot for the actual position and the position decoded from the neuronal activity for one animal and one context; a high $R^2$ value (0.725) indicates a good fit of the linear regression model to the observed data. **c** Mean $R^2$ of position decoded in each context on baseline (day 1), learning (day 4), and reversal learning (day 7). **d** Proportion of position-encoding cells across contexts and days. **e** Context specificity of position-encoding cells on each day. **f** Normalized response ($\Delta F/F$) of three speed-encoding neurons in the RSC across multiple trials as a function of running speed. **g** Example linear regression plot for actual speed versus speed decoded from the neuronal activity for one animal and one context; a high $R^2$ value (0.66) indicates a good fit of the linear regression model to the observed data. **h** Mean $R^2$ of actual and decoded speeds in each context across days. **i** Proportion of speed-encoding cells across contexts and days. **j** Context specificity of speed-encoding cells on each day. For all, $n = 5$ mice. For **c**–**e** and **h**–**j**, data are presented as mean values + SEM. For **c**, **d**, **h**, and **i**, results after random shuffling of the raw data are shown in a diagonal-stripe pattern. For **c**–**e** and **h**–**j**, experimental data points are shown as gray dots and the shuffled data points are shown as black dots. Two-way RM ANOVA and post-hoc Holm–Sidak tests were used. Within-day comparisons: *$p < 0.05$, **$p < 0.01$, ***$p < 0.001$; between-day comparisons: #$p < 0.05$, ###$p < 0.001$. Exact $p$-values can be found in Supplementary Dataset 1. Source data are provided as a Source Data file.

was 0.4–0.5 (Fig. 4h; compared to ~0.10–0.15 for shuffled data) with no significant differences across learning phase or specific context. Approximately 30% of cells showed a significant contribution ($p < 0.05$) to the encoding of speed information in each of the contexts, independent of learning phase (Fig. 4i; in contrast to ~10% for shuffled data). Interestingly, similar to position encoding, we also found that ~40% of speed cells were specific to a single context, while ~20% contributed to decoding in two contexts, and only a small proportion of cells were speed encoding in all three tested contexts (Fig. 4j). Notably, there was a moderate but significant increase in the proportion of neurons encoding speed in two contexts after reversal learning when compared to baseline and initial learning sessions (Fig. 4j), suggesting that speed encoding may not be entirely unaffected by previous learning.

**Multidimensional encoding by RSC neurons.** Given our findings that RSC neuronal activity encodes multiple task-related aspects, and additionally that both position and speed-encoding cells can also be context-specific, we next wanted to investigate the potential for neurons to encode information along these multiple dimensions. Further, since neurons encoding across multiple task parameters have been shown to provide a computational advantage in the prefrontal cortex during complex cognitive tasks[37], we wanted to examine how these dynamics change in the RSC across learning phases, considering reversal learning requires increased cognitive flexibility[41]. For each individual neuron imaged within the RSC population, we determined if it displayed activity sufficient to decode context, position and/or speed as individual parameters (according to the criteria described above; e.g., see Fig. 5a). We found that a relatively large proportion of neurons encoded information across all three task dimensions, and that this proportion remained consistent from the baseline day ($28.40 \pm 6.57\%$) to after the initial learning phase ($28.90 \pm 4.21\%$), while there was a significant increase in the proportion of multidimensional neurons after reversal learning ($39.69 \pm 6.05\%$; Fig. 5b, c).

We also analyzed the imaging dataset to specifically follow the responses of individual cells across days (in 4 out of 5 mice; see Methods and Supplementary Fig. 13), and found that the proportion of multidimensional cells was comparatively stable across learning, and a significant proportion of individual cells remapped from being responsive to a single parameter on baseline to then become multidimensional neurons following reversal learning (Fig. 5d). Since there is no change in the dynamics of how the task parameters are related to each other across learning, (i.e., visuomotor factors may influence both speed and context for instance, but they are entangled to the same degree across all learning phases), this transition from non-encoding or single parameter encoding to multidimensional encoding during reversal learning, suggests that these neurons may be contributing to more complex cognitive processing, beyond simple sensory or motor input-output functions.

To investigate this further, we isolated the population of multidimensional cells versus the population of single dimension encoding cells and examined the pattern of activity (Fig. 5e, g, Supplementary Fig. 12) as well as the decoding accuracy of the rewarded context in the ITI period (Fig. 5f, h). Here, we found that the population of multidimensional cells had significantly higher decoding accuracy specific to the rewarded, or behaviorally valued, context across learning, whereas the population of single-dimensional cells did not show any significant bias for higher decoding accuracy within the ITI period (Fig. 5f, h). Therefore, this multidimensional cell population within RSC is particularly important during the period following the reward, and has the potential to encode context-value associations in this context discrimination paradigm.

**RSC inactivation leads to deficits in acquisition and updating context-value associations.** Given our results, we wanted to investigate if inactivating the RSC would lead to behavioral differences during learning and reversal learning and whether context-value associations were altered when the RSC was perturbed. To do this, we used a chemogenetic approach where mice were injected with an AAV driving the expression of inhibitory DREADDs (Designer Receptors Exclusively Activated by Designer Drugs) (rAAV8-hSyn-hM4Di-mCherry; e.g., see Supplementary Fig. 14) into the RSC in order to inhibit neuronal activity in this area by systemic administration of clozapine-N-oxide (CNO). In the first learning phase with CNO-on (i.e., RSC inactivated; Day 2–4), mice ran at the same speed for all contexts and this speed was slower than on the baseline day (Fig. 6a). Therefore, mice assigned a positive-value bias to all contexts across this learning phase and failed to devalue the non-rewarded and neutral contexts (Fig. 6d; see also control data Fig. 2a). This positive-value bias seems to be the 'default' state in early learning (i.e., see also Day 2 of control data Fig. 2a), but RSC inactivation prevented the behavior to be updated according to the relevant rewarded context-value association. Hence, RSC inactivation prevented learning acquisition; this was also reflected on the next day when mice were tested with CNO-off and they failed to show a behavioral pattern to indicate successful context discrimination across the RSC-inactivated learning phase (Fig. 6a and Supplementary Figs. 15–17). The mice then remained off CNO and went through the learning phase again, demonstrating a behavioral change and clear context discrimination for the rewarded context when RSC neuronal activity was intact (Day 8; Fig. 6b, d). To examine if this learning was preserved when the RSC was inactivated again during memory retrieval, the next day mice were tested with CNO-on and we found that the context-value associations remained intact; hence, RSC inactivation did not disrupt the behavioral performance of a previously learned context-value association during retrieval (Day 9; Fig. 6a). This also indicates that mice could still recognize the individual contexts (i.e., as defined by the visual stimuli presented on the corridor walls) when RSC was inactivated, hence, learning deficits in the initial phase with CNO-on cannot be explained simply by a general deficit in visual perception.

We then continued with the reversal learning task with the RSC inactivated (CNO-on, Day 10–12). If mice showed a generalized inability to update contextual associations indiscriminately with RSC inactivated, we would see a similar pattern of behavior to Day 8/9, with mice continuing to run faster through the previously non-rewarded context 2. However, we found again that mice had assigned a positive-value bias to both contexts in this early relearning phase, similar to what we saw on Day 5 in control mice (see Fig. 6a and Supplementary Figs. 15–17). As training days proceeded, when the RSC was inactivated, in contrast to controls, mice failed to devalue the previously rewarded context (context 1) and therefore did not run significantly faster. Instead, the previously rewarded context 1 is maintained as valued throughout the CNO-on phase of reversal learning (see Supplementary Figs. 15–17) and, therefore, there was no significant difference in running speed between the previously and newly rewarded contexts (context 1 and 2) by the last testing day (Day 12). With regard to the neutral context, with the RSC inactivated, we do not see context 3 adopt a significant positive-value bias as it did during the initial learning phase with CNO-on (Day 4), but the association of this context as a neutral context from Day 8 was maintained on the test day even after

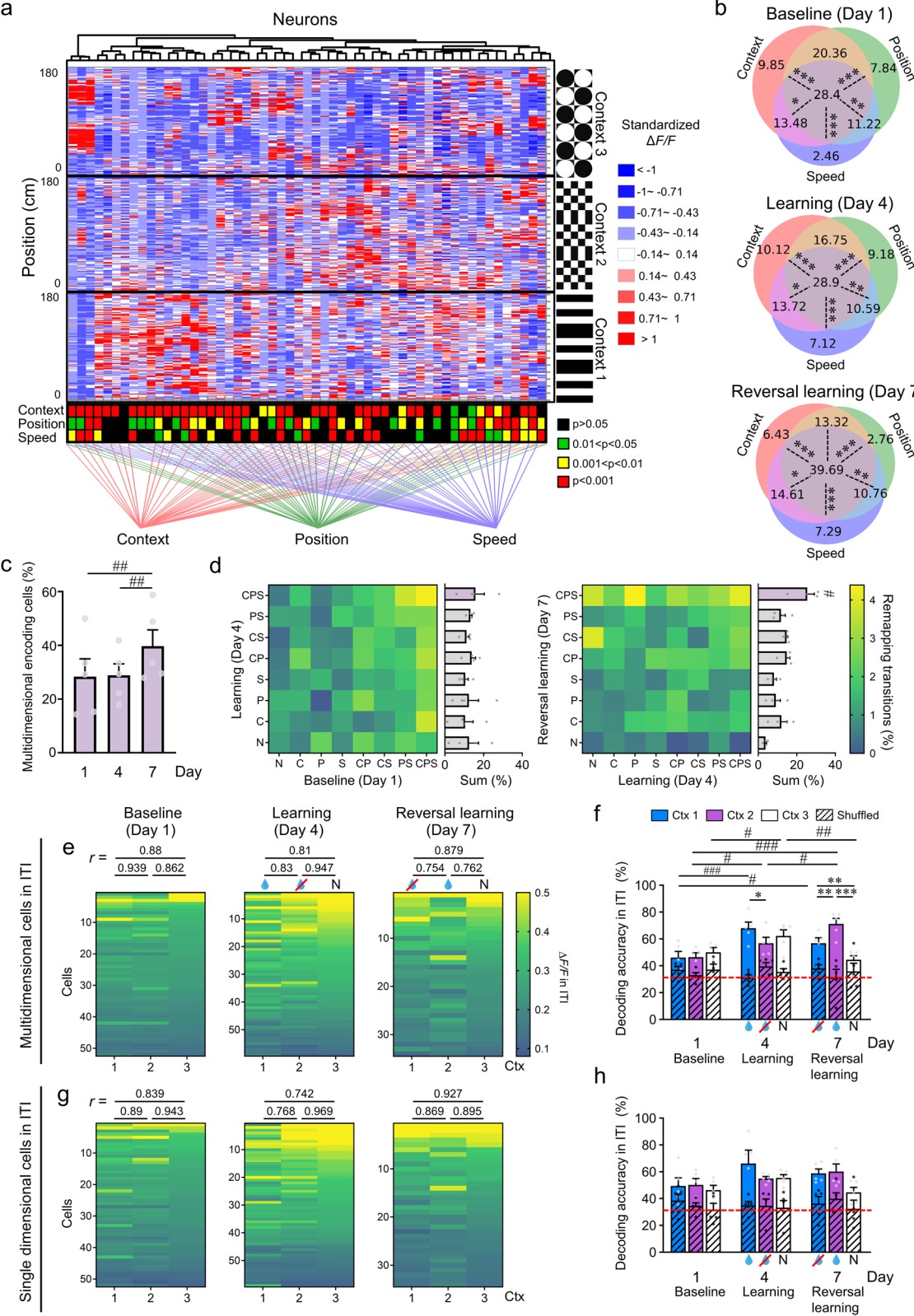

RSC inactivation (Day 12), similar to the retrieval day (Day 9; Fig. 6a). In the current paradigm, there is no need for this neutral context to be 'updated' across learning and reversal learning phases (Day 8 to Day 12), as it remains neutral throughout, but our results show that again this information is retained across the reversal learning phase even with the RSC inactivated (Fig. 6a). Finally, in the last phase, successful reversal learning proceeded

with the same pattern of behavioral changes when CNO was off and the RSC again had intact neuronal activity (Fig. 6a, c, e), and a final testing day with CNO-on again confirmed that once the new reversed context-value relationships were acquired, they could be recalled even with the RSC inactivated (Day 17; Fig. 6a).

Taken together, we found that when the RSC was inactivated, there was a deficit in forming and updating context-value

**Fig. 5 Neurons in the RSC can encode for multiple task-related parameters and the proportion of these multidimensional neurons is increased after reversal learning. a** Normalized response ($\Delta F/F$) of all imaged RSC neurons from one animal in three contexts (neurons in columns, positions along the corridor in rows). Hierarchical cluster analysis links categories of neurons that encoded information across multiple dimensions. Significance values of their contributions to the encoding of context, position, and speed are indicated below. **b** Percentage of neurons encoding various task-related parameters across learning phases (days 1, 4, and 7). **c** Proportion of multidimensional (context, position, and speed) encoding neurons across learning phases. **d** Percentage of cells followed across days that have changed their encoding properties (remapping transitions; i.e., from either non-encoding [N], context [C], speed [S], or position [P] encoding and all combinations of C, S, and/or P) from baseline (day 1) to learning (day 4; left) and then learning to reversal learning (day 7; right). Bar graphs show the sum of the percentage of cells in each encoding category after learning (left) and reversal learning (right; 361 cells from $n = 4$ mice). **e** Mean $\Delta F/F$ during the intertrial interval (ITI) of each neuron classified as multidimensional (CPS category) for each context (Ctx 1–3) across learning phases. Cells are sorted according to the mean $\Delta F/F$ of the response to the neutral context (context 3). Cross-correlations (Spearman $r$ value) of neuronal activity between contexts are reported on top of the color maps. **f** Decoding accuracy of multidimensional neurons for each context during the ITI across learning phases (days 1, 4, and 7). **g** Same as **e** but for single-dimensional cells (C, P and S categories). **h** Same as **f** but for single-dimensional cells. For all panels except **d** (4 mice), $n = 5$ mice. For **c**, **d**, **f**, and **h**, data are presented as mean values + SEM and the experimental data points are shown as gray dots while the shuffled data points are shown as black dots. For **f** and **h**, results after random shuffling of the raw data are shown in a diagonal-stripe pattern. For **b**, **d**, **f**, and **h**, two-way RM ANOVA and post-hoc Holm–Sidak tests were applied. For **c**, one-way RM ANOVA and post-hoc Holm–Sidak tests were applied. Within-day comparisons: $*p < 0.05$, $**p < 0.01$, $***p < 0.001$; between-day comparisons: $^{\#}p < 0.05$, $^{\#\#}p < 0.01$, $^{\#\#\#}p < 0.001$. Exact $p$-values can be found in Supplementary Dataset 1. Source data are provided as a Source Data file.

associations during our paradigm, and particularly in devaluing previously formed associations with reversal learning, indicating a disruption in cognitive flexibility. Note there were no significant differences in the pattern of behavioral responses between animals with CNO-off and the previous control animals that had control rAAV8 injections (rAAV8-hSyn-mCherry lacking the inhibitory G-protein-coupled receptor, hM4Di) plus CNO treatment (Fig. 2a; Supplementary Fig. 18), or animals with no AAV injection (Supplementary Fig. 1); thus, the performance was not affected by rAAV8 transfection or CNO injection alone.

## Discussion

In this study, we utilized a visuospatial context discrimination paradigm in a virtual environment to investigate how multiple task parameters are encoded across learning in the RSC, a brain region known for its diverse functional inputs[1,12]. We found that a large proportion of individual neurons in the RSC integrate multiple aspects of task-related information while forming context-reward-value associations across learning. These multidimensional neurons contribute to the encoding of context-specific visual features of the environment, the context-associated reward value as well as the spatial position and speed of the animal. Interestingly, a more complex reversal learning task resulted in an increased proportion of these multidimensional encoding neurons and this population of cells showed higher decoding accuracy for behaviorally relevant context-value associations during the ITI period of the task. Further, we found that behavioral context discrimination decreased when the RSC was inactivated during learning phases in which context-reward-value associations were formed. However, recall of previously formed associations remained intact after RSC inactivation. RSC inactivation resulted in a persistent positive behavioral bias in valuing contexts, suggesting that the RSC may play a role in contextual value updating by assigning a negative value to non-rewarded and neutral contextual cues.

Neurons within the dysgranular RSC predominantly and concurrently encoded multiple aspects of information during our context-reward-association paradigm. As the RSC is densely interconnected with a wide range of brain regions and involved in an array of cognitive functions, previous studies have proposed a critical translational function of the RSC in the integration and transformation of information[1,34]. This proposed translational function is supported by multiple behavioral studies showing that the RSC is involved in integrating navigational information[1], visuomotor processing[18,20], forming associations between multiple sensory stimuli[50], encoding multiple variables with distinct temporal dynamics[51], and processing the conjunction between allocentric and egocentric spatial frames[25]. Along the virtual corridor, we found that neuronal activity related to context, position, and speed within each experimental day resulted in high decoding accuracy of these parameters within the imaged RSC population, with no systematic bias towards the behaviorally relevant context. However, during the delay period between trials (ITI) when no visual stimulation was present, the rewarded context-value associations were more accurately decoded and the pattern of neuronal activity was similar for the non-rewarded and neutral contexts, but unique for the rewarded context — reflective of the observed behavioral changes in running speed.

Longitudinal analyses of neuronal activity revealed a significant remapping of task parameters encoded by individual neurons, leading to more multidimensional responses following reversal learning. However, this remapping did not preclude accurate decoding of the rewarded context-value association across learning phases. Interestingly, neurons that showed multidimensional responses along the corridor also more accurately encoded the context-reward associations during the delay period between trials (ITI) compared to neurons encoding for a single dimension along the corridor. This supports a translational role for the RSC in forming context-value associations based on integrative information. This is also in agreement with a recent study demonstrating the role of the RSC in value-based decision making with a dynamic foraging task, in which persistent population activity in the RSC enabled consistent decoding of value-related variables even throughout periods spanning the ITI and pretrial ready phases[36]. The RSC has also recently been implicated in associating environmental contexts to motor planning signals in a virtual T-maze task[51], with anticipatory responses to appropriate motor choices. During our ITI period, since the animal has already consumed the reward, the spout has been removed, and the treadmill is held stationary, we do not expect any systematic motor or behavioral differences across contexts beyond fundamental differences that are integral to the learning process (e.g., attention and arousal), which may affect the general behavioral state. However, in this regard, determining the contribution of specific neuromodulatory influences in anticipation of and during this ITI time period will be of interest as a potential underlying mechanism of plasticity across learning phases[52].

The increase in the proportion of multidimensional cells after reversal learning may represent the recruitment of additional resources for cognitive flexibility. Neurons that respond to multiple task-related aspects in the prefrontal cortex have been shown

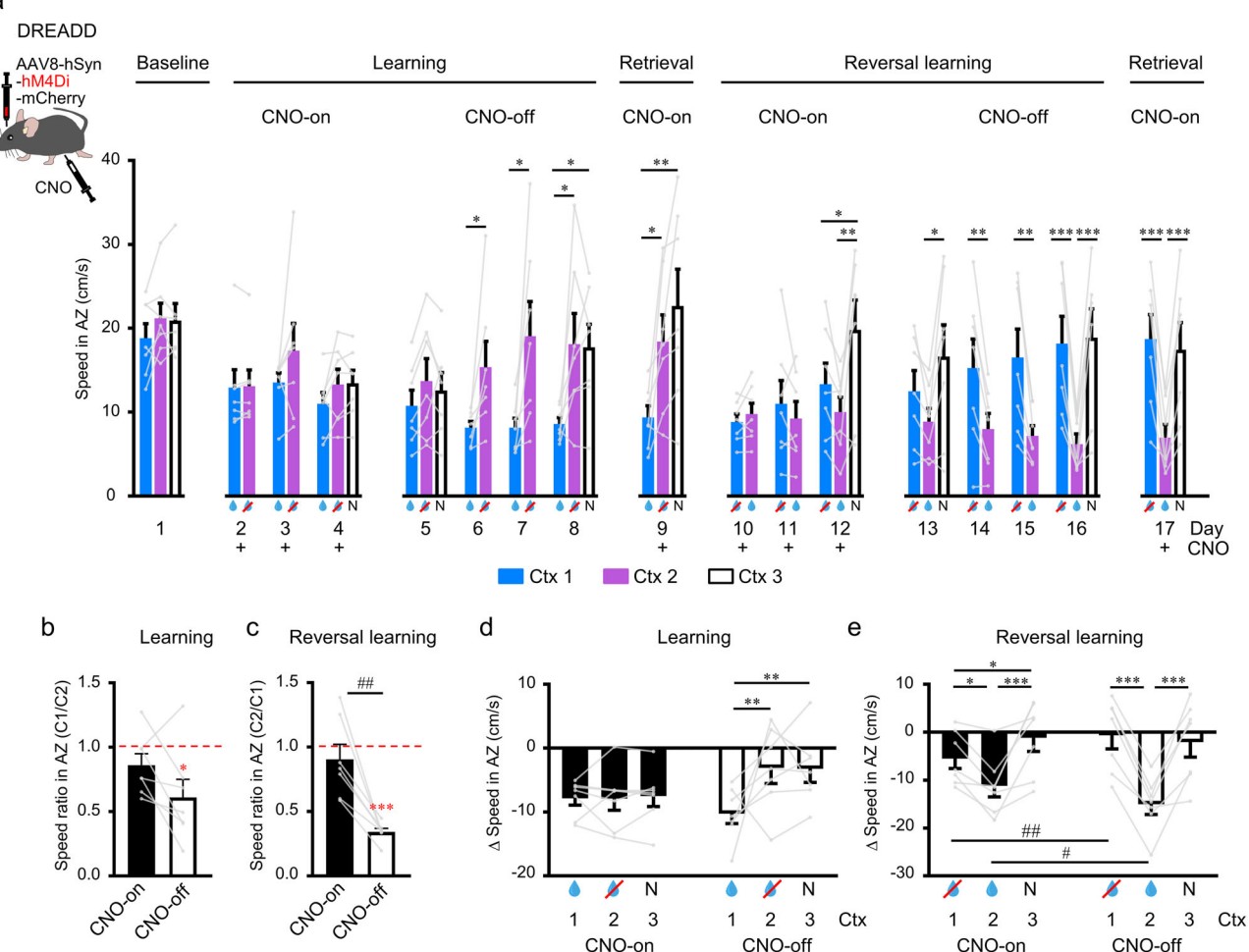

**Fig. 6 Behavioral context discrimination is impaired by chemogenetic inactivation of the RSC. a** Mean running speed in the anticipation zone for mice injected with AAV8-hSyn-hM4Di-mCherry to express inhibitory DREADD. CNO was applied either during the memory formation process (days 2–4 and 10–12) or after the formation of memory (days 9 and 17; indicated by +). **b** Ratio of speed in the anticipation zone between rewarded context (C1) and non-rewarded context (C2) after CNO-on learning (day 4) and CNO-off learning (day 8). **c** Same as **b** but after CNO-on reversal learning (day 12) and CNO-off reversal learning (day 16). **d** Change in speed relative to baseline day after CNO-on learning (day 4–day 1) and CNO-off learning (day 8–day 1). Rewarded context (blue drop), non-rewarded context (drop with red cross), and neutral context (N). **e** Same as **d** but after CNO-on reversal learning (day 12–day 1) and CNO-off reversal learning (day 16–day 1). For all, n = 7 mice. For **a**, one-way RM ANOVA and Holm–Sidak post-hoc test were used when three contexts were presented; paired *t*-test was used for days when two contexts were presented. For **b** and **c**, paired *t*-test was used for comparisons between CNO-on and CNO-off conditions; one-sample *t*-test was used to compare to the hypothesized population mean (1.00). For **d** and **e**, two-way RM ANOVA and post-hoc Holm–Sidak tests were used. For all, data are presented as mean values + SEM, and the data points are shown as gray dots with the data from the same animals linked by gray lines. Within-day comparisons: *$p < 0.05$, **$p < 0.01$, ***$p < 0.001$; between-day comparisons: #$p < 0.05$, ##$p < 0.01$. Exact *p*-values can be found in Supplementary Dataset 1. Source data are provided as a Source Data file.

to have a significant computational advantage in terms of the complexity of input-output functions that they can represent, and this dimensionality is predictive of behavior[37,53]. Therefore, a larger proportion of multidimensional representations in the RSC may allow animals to flexibly update context-value associations. Through iterations of learning and reversal learning with and without RSC inactivation, we show that inactivating the RSC disrupted the ability to devalue previously rewarded contexts, while animals maintained the previously acquired value of the neutral context. The RSC has recently been shown to be important for history coding of value-based decisions[36], and our data suggest that inactivation of the RSC explicitly affects the ability to assign a negative value to a context. This reversal learning task creates a conflict between the previously established value of the context and a new reality, resulting in the need to update the previous context-value association to resolve this conflict. The use of multiple parameters in the environment indeed provides a

strategy to resolve conflicting value-based information. This is demonstrated here on the behavioral level as well as reflected in the pattern of activity within the RSC. It is noteworthy that previous studies have shown the RSC is also engaged in tasks dealing with other forms of conflicting information, for instance, conflicting proximal and distal cues[54] or isolation of spatial information in a place avoidance task[55]. Although it can be difficult in visuospatial tasks to fully separate out the influence of multiple task parameters, for instance, changes in running speed will also alter the optic flow created by the visual stimulus on the corridor walls, these parameters are inherently entangled in natural navigation and yet do not change in quality between learning and reversal learning in this paradigm. Therefore, the increase in multidimensional neurons with reversal learning represents population level plasticity, likely in response to a more complex cognitive demand. Further studies investigating the underlying circuit and synaptic dynamics of this experience-

dependent plasticity may provide insights into how individual neurons get recruited to represent multidimensional aspects during these context-value associations.

Using chemogenetic inactivation, we found that the RSC was required for initial learning and updating context-value associations, but not necessary for retrieval of previously learned associations. This is seemingly in contrast with some studies examining the role of the RSC in contextual fear conditioning, which have demonstrated that RSC lesions impaired recall of contextual memory[17,56], and optogenetic reactivation of activity patterns occurring during contextual fear conditioning were sufficient to drive contextual memory recall[57]. However, contextual learning may differ in relation to fear conditioning versus reward conditioning. These involve an appetitive motivational system that mediates approach versus an aversive motivational system that reinforces escape and, while there may be complex relationships between these two systems[58], they are in fact mutually inhibitory[59,60]. Although the RSC undoubtedly plays a role in both of these motivational systems[61], these pathways may diverge such that context-value associations in our paradigm are formed in the RSC but the reward-associated contextual memory traces themselves are not necessarily exclusively consolidated there. This is supported by our observation that decoding accuracy of context, position and/or speed do not change along the virtual corridor with learning or reversal learning and the activity in the RSC does not develop a significant bias towards the rewarded context while the animal is within the context itself. In this regard, it is interesting that we do find responses specific to the rewarded context in the ITI period, which are not simple reflections of responses to the reward-onset event itself. This role for the RSC to encode behaviorally significant contexts has also been demonstrated in rats performing a spatial context-discrimination foraging task, which also showed heterogeneous responses to a reward-event that were not specifically event-timed (in contrast to reward responses observed in the hippocampus) and the development of these responses even preceded significant improvements in behavioral performance across training sessions[35]. In our study, this reward-linked activity in the delay period between trials may represent persistent local integration of previous task-related activity within the RSC or RSC updating through reciprocally connected cortical regions, such as the prefrontal[62], orbitofrontal[63], primary sensory[64], and/or posterior parietal[65–69] cortex. It is additionally possible that due to the nature of our visuospatial paradigm where multiple task-related aspects are represented together, the RSC is more specifically permissive during the formation and updating of these context-value associations. Indeed, even in fear conditioning paradigms, while the RSC is not necessary for the formation of a fear response to a single cue, it is required for the acquisition of fear responses that necessitate the integration of multiple cues and forming associations between various environmental stimuli[50,56,70]. Lastly, our inactivation of the RSC was not spatially selective for either the granular or dysgranular regions (which have been shown to have functional differences[71]) or cell-type specific; therefore, future studies examining more precise RSC inactivation, particularly in relation to cell populations encoding information across multiple parameters, will provide further insight into the integrative processing dynamics within the RSC in relation to memory formation and recall.

Taken together, our study demonstrates that the RSC plays a role in context-value association and updating, and that a large proportion of RSC neurons encode for multiple task-related dimensions. These populations show more efficient encoding of reward-associated contexts during intertrial periods and increase in proportion when task parameters require increased cognitive flexibility across learning. Considering the RSC is an early-onset

center for dysfunction in neurodegenerative disorders such as Alzheimer's disease[72–75], further studies investigating changes in multidimensional encoding and specific disruptions of learning and memory processes, may provide a useful behavioral marker for neurodegenerative progression in both human and animal studies.

## Methods

**Animals.** Eighteen Thy1-GCaMP6f mice with C57BL/6 J genetic background (C57BL/6J-Tg [Thy1-GCaMP6f] GP5.5Dkim/J, Jackson Laboratory, USA; RRID: IMSR_JAX: 024276) were used for experiments (5- to 6-month-old males). All mice were cared and treated strictly following the ethical animal research standards defined by the Directive of the European Communities Parliament and Council on the protection of animals used for scientific purposes (2010/63/EU) and were approved by the Ethical Committee on Animal Health and Care of Saxony-Anhalt state, Germany (license number: 42502-2-1346). Mice were group-housed until surgery under a reversed 12 h light/dark cycle with water and food available ad libitum. The ambient temperature is kept at 22 °C with 65% humidity. For the first baseline experiment, six mice were used, which did not receive any adeno-associated virus (AAV) injection. For the next control experiment, five mice were injected with a control virus AAV8/hSyn-mCherry (AV6443, UNC GTC Vector Core, USA; RRID: Addgene_114472; titer: $4.6 \times 10^{12}$ vg/mL), which drives mCherry expression. For the last inactivation experiment, a chemogenetic approach (DREADD) was used to inactivate the RSC in seven mice. Here, AAV8/hSyn-hM4Di-mCherry (AV5630D, UNC GTC Vector Core, USA; RRID: Addgene_50475; titer: $7.4 \times 10^{12}$ vg/mL) was injected into the RSC followed by clozapine-N-oxide (CNO)-induced neuronal inhibition at specific time periods during the experiment. In control experiments, mice expressing mCherry received CNO at the same time points as hM4Di plus mCherry expressing mice.

**Surgical procedures.** Surgical procedures were modified from previous studies[76,77]. Anesthesia was induced with 4% isoflurane (Baxter, Germany) before the mouse was fixed in the stereotaxic apparatus (SR-6M, Narishige Scientific Instrument Lab, Japan) and adjusted to 1.5–2% during surgery with 0.4 L/min O$_2$. Ophthalmic ointment was applied to protect the eyes (Bepanthen, Bayer, Germany). To decrease cortical stress response and avoid cerebral edema, dexamethasone (2 mg/kg body weight; Mephamesone, Mepha Pharma, Switzerland) was subcutaneously injected, and carprofen (5 mg/kg body weight; Rimadyl, Pfizer, USA) was intraperitoneally injected to reduce inflammation. Mice were placed on a controlled heating pad, and the temperature was maintained at 37 °C (ATC1000, World Precision Instruments, USA). The scalp was shaved and cleaned with 70% ethanol followed by an incision, and the surface of the skull was cleaned with 10% povidone-iodine (Dynarex, USA) and 3% hydrogen peroxide solution (Sigma–Aldrich, Germany).

A craniotomy over the retrosplenial cortex (4 mm in diameter, centered over the midline and −2.0 mm from Bregma; Fig. 1d) was performed with a high-speed dental drill (Eickemeyer, Germany). To avoid heat-induced damage, the drilling procedure was stopped periodically and sterile cold saline was applied to the skull during the interval. AAVs were bilaterally injected into the RSC (ML, ±0.4 mm; AP, −2.0 mm from Bregma) using a 10 µL NanoFil syringe with a 35-gauge beveled needle, attached to an Ultra Micro Pump (UMP3) with Micro 4 MicroSyringe Pump Controller (World Precision Instruments, USA) at a speed of 100 nL/min at two different depths (~200 and 700 µm; 500 nL per site)[16,78]. After each injection, the needle was kept in place for five minutes before it was slowly withdrawn. A 5 mm in diameter circular glass coverslip (Thermo Fisher Scientific, Germany) was then used to cover the craniotomy sealed by cyanoacrylic glue. A custom-built 3D printed metal head-plate (Fig. 1e; i.materialise, Belgium) was implanted on the exposed skull with cyanoacrylic glue and dental acrylic (Paladur, Heraeus Kulzer, Germany). Animals were returned to their home cage after recovery from anesthesia, and training began 1 week after surgery. Mice were treated once per day with carprofen to prevent inflammation and reduce pain (5 mg/kg body weight, i.p.) for the first three days following surgery.

**Virtual reality setup.** The virtual reality (VR) system and environment were modified from previous studies[79–81]. Experiments were performed using a JetBall-TFT VR system (PhenoSys, Berlin, Germany), which consisted of a TFT surround monitor system (6 monitors) covering ~270° of the horizontal visual field of the mouse and an air-cushioned spherical treadmill with two XY-motion sensors that translate the movements of the sphere into VR coordinates (Fig. 1b), with a gain of 1 (physical distance to VR distance). The VR system recorded the XY-coordinates at a 100 Hz sampling rate via two optoelectronic XY-sensors with a direct USB connection and mean running speed was calculated from recorded data using custom-written code (MATLAB, MathWorks, MA, USA). A retractable reward spout connected to a peristaltic pump was put into position when animals arrived at a specified VR location for every trial (regardless of whether a reward was dispensed or not) and was retracted either 1.5 s after the dispensing of the reward or after 1.5 s if no reward was given. A 4 µL droplet (10% sucrose in water) was dispensed on reward trials if the animal licked at the predefined reward position.

PhenoSoft VR software was used to establish and perform the presentation of the VR environment and record the related behavioral data (PhenoSys, Berlin, Germany). Three linear virtual corridors were used in this study (context 1, context 2, and context 3), which each consisted of different visual stimuli along the virtual corridor walls (Fig. 1c). The total length of each corridor was 200 cm, and the potential reward position was located at 180 cm from the starting point. After the reward spout was retracted for each trial, the screens turned black for 3 s as an intertrial interval (ITI), and the animal was 'teleported' back to the starting point of the linear track for the next trial. The interval of 10 cm before the reward position (i.e., 170–180 cm) was defined as the anticipation zone (Fig. 1c, h).

**Behavioral training.** The initial habituation to the recording environment and pretraining for reliable running performance began 1 week after surgery. Mice were head-fixed at the same height as the center of the TFT monitors and able to move freely for 1 h/day on the spherical treadmill (30 cm diameter; Phenosys, Berlin, Germany) in darkness without any virtual environment presented. Mice were water restricted (1.5 ml of water/day) from the first day of pretraining to the end of the experiment and weighed daily to ensure they were ≥~85% of their prewater-restriction weight. During pretraining sessions, water was given via the reward spout, so the mice associated licking the spout to receiving a water reward. After 14 days of pretraining (~3 weeks after cranial window surgery and AAV injection), mice showed fast, straight and constant running on the spherical treadmill, and the experimental protocol and imaging began.

The experimental timelines are shown in Fig. 1a, i. For these experiments, on the first imaging day (day 1; baseline recording), each of the three virtual corridors were presented for 20 trials in a random sequence without any reward (reward spout was not extended). During subsequent learning sessions (day 2–3), contexts 1 and 2 were randomly presented for 40 trials each and the reward spout was extended for each context but the water reward was given only during context 1 trials at the fixed reward position (180 cm). The following day was a test session (day 4), in which all three contexts were presented for 15 trials each, and water reward was given only in context 1, as during the previous learning sessions. After this initial learning phase (day 2–4), reversal learning sessions were performed (day 5–6), in which context 1 and 2 were randomly presented again, but the rewarded context was reversed from context 1 to context 2; this was followed by a reversal test session (day 7) where all three contexts were presented and the water reward still given for context 2 trials only (see Fig. 1i). For the control experiments, CNO (10 mg/kg, 0.1 mL/10 g, body weight) was intraperitoneally injected (i.p.) 15 min prior to the virtual environment presentation on each day except for the baseline day (day 1).

For the inactivation experiments, the experimental timeline was derived from the same scheme, but with modifications to dissect the role of the RSC in acquisition versus recall of memories. Here, initial learning sessions were first performed in the presence of CNO (day 2–4), then behavior was tested (day 5). The next learning sessions were done in the absence of CNO (day 6–7), and mice were tested again without (day 8) and with (day 9) CNO to examine the effect of neuronal inhibition during memory formation processes. This was then repeated during reversal learning (learning-day 10–12 with CNO; testing day 13; learning-day 14–15 without CNO; testing day 16 with and day 17 without CNO) to determine the effects on recall of formed memories. For the days without CNO application, the same volume of 0.9 % NaCl was injected (i.p.) as the control.

**In vivo two-photon calcium imaging.** Two-photon calcium imaging (Fig. 1b) was performed using resonant scanning two-photon microscopy (B-scope; Thorlabs, USA), and a Ti:Sapphire pulsing laser (Chameleon Ultra II, Coherent, USA) tuned to 920 nm. GCaMP6f fluorescence emission was isolated by a band-pass filter (525/50, Semrock, USA) and detected by a GaAsP photomultiplier tube (Hamamatsu, Germany). Images were acquired through a ×20 water immersion objective (1.00 N.A.; Olympus, Japan) with a frame rate of 14.7 Hz (real-time averaging by 4) for bidirectional scanning at a resolution of 256 × 256 pixels (300 × 300 μm field of view) and controlled by ThorImageLS imaging software (version 2.4). In order to prevent light leakage from the VR displays into the microscope, a custom-made black foam ring was used between the microscope objective and the head-plate. Imaging and behavioral data were synchronized by custom-written code (MATLAB, MathWorks, MA, USA). Images were collected at a single L2/3 focal plane per animal at cortical depths between 120 and 180 μm, and the same RSC region was imaged across multiple days (Fig. 1f).

**Histology.** Procedures for histology were performed based on previous protocols[82,83]. Animals were deeply anesthetized with isoflurane (Baxter, Germany) and transcardially perfused first with PBS (0.1 M, pH 7.4) and followed with 4% paraformaldehyde (PFA). The brains were removed and postfixed in 4% PFA at 4 °C for 24 h and then transferred to 30% sucrose for cryoprotection until the solution had infiltrated into the whole brain (~48 h). The brains were then frozen in 100% 2-methylbutane (kept at −80 °C) and stored at −80 °C until sectioning. Forty-micrometer thin coronal sections were cut using a cryostat (Leica CM1950, Germany). Floating sections were kept in cryoprotection solution (1 mL ethylene glycol (Carl Roth, 6881), 1 mL glycerine (Carl Roth, 3783), and 2 mL of 1× PBS (Life Technologies, 10010056), pH 7.4).

The sections were first washed in phosphate buffer (PB; 3 × 10 min, at room temperature (RT) with gentle shaking) and then permeabilized with 0.5% Triton X-100 (Sigma–Aldrich, T9284) in PB for 10 min at RT. Next, the sections were incubated for 1 h (at RT with gentle shaking) in a blocking solution containing 5% normal donkey serum (NDS, Jackson ImmunoResearch, 017-000-121), 0.4% Triton X-100 and 0.1% glycine in PB. Subsequently, sections were incubated for 48 h (at 4 °C with gentle shaking) with the primary antibodies (chicken anti-GFP, 1:500, abcam, ab13970, RRID: AB_300798; goat anti-mCherry, 1:200, SICGEN, AB0040-200, RRID: AB_2333092; mouse anti-NeuN, 1:500, Merck Millipore, MAB377, RRID: AB_2298772) in blocking solution. The sections were then washed 3 × 10 min at RT in PB and incubated on a shaker for 3 h at RT with the secondary antibodies (Alexa Fluor 488 donkey anti-chicken, 1:500, Jackson ImmunoResearch, 703-545-155, RRID: AB_2340375; Alexa Fluor 568 donkey anti-goat, 1:500, abcam, ab175704, RRID: AB_2725786; Alexa Fluor 647 donkey anti-mouse, 1:500, ThermoFisher, A31571, RRID: AB_162542). Finally, the sections were washed 3 × 10 min at RT with washing buffer and 1 × 10 min at RT with PB and mounted on SuperFrost glasses (Thermo SCIENTIFIC, J1800AMNZ) with Fluoromount medium (Sigma–Aldrich, F4680). Images were acquired using a confocal laser-scanning microscope (LSM 700, Carl Zeiss, Germany) and ZEN software (Carl Zeiss, Germany).

### Data analysis

*Two-photon calcium imaging.* Two-photon imaging datasets were acquired during days 1 (baseline), 4 (after learning), and 7 (after relearning) for control experiments, where 200–300 neurons from five mice were analyzed. Motion artifacts were corrected for by a technique based on nonlinear optimization and discrete Fourier transform (DFT) in low noise imaging[84] for the case of uniform motion artifacts, where the quality of image registration was assessed using normalized root-mean-square (NRMS) between reconstructed and reference images[85] using the first image frame as the reference. Alternatively, we used a template matching method that split the field-of-view into spatially overlapping patches according to user-determined dimensions, registered corresponding patches of the template separately and then merged the registered subpatches to each other[86] for the case of nonuniform motion artifacts. Image registration quality was measured by the image crispness defined as the Frobenius norm of image gradient vector and image magnitude.

To extract the neuronal change in fluorescence, we used a method that automatically identified ROIs (including spatially overlapped ones), denoised signals, and when comparing the ITI period to the immediately preceding reward period (Fig. 3f; Supplementary Figs. 10, 11) deconvolved signals[87] with open source and adapted MATLAB code (MathWorks, MA, USA). Briefly, this method uses constrained non-negative matrix factorization (cNMF) to isolate spatially and temporally independent fluorescent signals, approximating a parametric model for continuous timeseries calcium transients as the impulse response of an autoregressive process, and then estimates the spiking signal from the sparsest non-negative neural activity signal. This method can denoise the spatiotemporal imaging set and model the background activity in each image frame by averaging the spatiotemporal background over ROIs. The temporal trace of each ROI was expressed as $\Delta F/F$, raw fluorescence trace divided by background activity. The default parameters were used with few exceptions (the order of AR process p was set to 2, temporal downsampling factor "tsub" was set to 4, spatial downsampling "ssub" was set to 2). Between 50 and 70 ROIs corresponding to the somata of neurons were identified per animal per session using this approach after manual confirmation.

To follow the changes in the encoding pattern of individual cells across days during learning and reversal learning to determine how neurons remapped response categories across non-encoding, single-, double-, or multidimensional encoding, ROIs were also automatically identified using suite2p[88] simultaneously across days 1, 4, and 7 for four out of the five mice used in the control group (the remaining mouse was excluded due to the field-of-view on baseline day not matching precisely to the following chronic imaging days).

*Context and context-value association decoding.* A linear discriminant analysis (XLSTAT, Addinsoft) was used for decoding the relationship between neuronal activity ($\Delta F/F$ values) and the identity of the virtual context on fixed days. Therefore, for each observed set of neuronal activities, discriminant functions were calculated for each of i-th contexts using the Eq. (1):

$$w_{i0} + \sum w_{ij}\, a_{jpt}, \text{ for } j = 1, 2, ..., \text{ number of neurons,} \qquad (1)$$

where $a_{jpt} = \Delta F/F$ for neuron $j$ at position $p$ for trial $t$, and $w_{ij}$ are the weights optimized to provide the highest value to the discriminant function corresponding to the right context. To predict context along the length of the virtual corridor, mean $\Delta F/F$ values were computed per neuron for 18 consecutive bins, 10 cm long each, and the context was predicted for each of these bins in all trials. In the backward stepwise method for cell selection, initially, all cells are included in the model, and then cells contributing the least to the context prediction are removed one-by-one if the quality of context prediction is not significantly changed ($P > 0.05$). Further, the contributions of each neuron to the context prediction were estimated by type III sum of squares analysis and neurons with $p < 0.05$ were identified as context-encoding cells. The quality of context prediction was

characterized by % of well-classified observations, which is the ratio of the number of observations for which the context was correctly predicted over the total number of observations (number of trials × 18). A cross-validation method was used to estimate the accuracy of context decoding. Cross-validation allows one to see the prediction for a given observation if it is left out of the estimation sample. Additionally, we split observations into two independent subgroups and predicted context for 100 observations using discriminant functions estimated on the rest of data, which provided a very similar outcome as the cross-validation method. To demonstrate that the linear discriminant model extracts meaningful information about the context-specific neuronal activity, we performed the analysis after random shuffling of contexts in each dataset.

Similarly, we used linear discriminant analysis for decoding of the relationship between longitudinal neuronal activity across days and context-value associations, which were alternated across learning phases (i.e., reward was associated with context 1 on day 4 and context 2 on day 7 versus no reward in all contexts on day 1, contexts 2 and 3 on day 4 and contexts 1 and 3 on day 7). As our behavioral results suggested animals show a change in speed associated with a particular context-value association beginning only after the first ~40 cm of the virtual corridor (see Supplementary Fig. 2), for this analysis we used the mean value of neuronal activity for 40–180 cm along the virtual corridor.

To compress and visualize the neuronal activity of all cells per animal in all contexts and days in 2D space, a principal component analysis was performed (XLSTAT, Addinsoft), which revealed that the first three principal components explain in total about 70% of cell activity variance and hence provide a highly informative representation of neuronal activity.

**Position and speed decoding**. A linear regression analysis (XLSTAT, Addinsoft) was employed for decoding of the relationship between neuronal activity ($\Delta F/F$ values) and the animal's position (18 bins) in a defined virtual context, i.e., the position was predicted using the Eq. (2):

$$w_0 + \sum w_j\, a_{jpt}, \text{ for } j = 1, 2, ..., \text{ number of neurons,} \qquad (2)$$

where $a_{jpt} = \Delta F/F$ for neuron $j$ at position $p$ for trial $t$, and $w_j$ are the weights optimized using the least squares method to minimize the difference between the actual and predicted position values for all observations.

In the backward stepwise method, initially, all cells are included in the regression model, and then cells contributing the least to the position prediction are removed one-by-one (using Student's $t$-test if, $p > 0.05$). Corrected $R^2$, which corresponds to the fraction of the dependent variable (i.e., distance) variance that is explained by the linear model, was used as a standard measure to report the fitting quality. Further, the contributions of each neuron to the prediction of position were estimated by type III sum of squares analysis and neurons with $p < 0.05$ were identified as position-encoding cells. Similarly, we performed decoding of the animal's speed after averaging of $\Delta F/F$ for speed bins of 10 cm/s using the same procedures. The position and speed-decoding analyses were performed separately for each context and experimental day and the outcomes were combined for the analysis of individual cell properties, i.e., to determine if a cell was significantly contributing to the encoding of position or speed or both, and if it was contributing in none, 1, 2, or 3 contexts. To demonstrate that the linear discriminant model extracts meaningful information about position- and speed-specific neuronal activity, we performed the analysis after randomly shuffling position and speed bins in each dataset.

**Visual cell analysis**. To search for visual feature-encoding neurons in the RSC, we described our VR environments by simple dark-to-light transition (DLT) functions having the value 1 at the spatial bins in which there was a dark-to-light transition in the VR environment and value 0 at other spatial bins. Then a cross-correlation function was computed between the DLT functions and activities of single neurons (averaged $\Delta F/F$ values across all trials per day per context). Considering that our VR environments were composed of repeated elements, resulting in periodic DLT functions, the cross-correlation functions were computed for time lags in the range between −3 to +3 bins (corresponding to the period of the slow-changing DLT function in the context 3). The maximal value of the Pearson coefficient of cross-correlation was used to determine at which lag the best fit between neuronal activity and the DLT function is achieved, and the $\Delta F/F$ signal was accordingly aligned. A cell was classified to be visually responsive if it has the maximal coefficient of cross-correlation significantly different from 0 with $p < 0.05$ (after Bonferroni correction for multiple comparisons) and at least two peaks aligned with one of 3 DLT functions corresponding to the 3 contexts used.

**Statistics**. In this study, unless indicated otherwise, all values reported represent mean + SEM with $n$ indicating the number of animals. All statistical comparisons were performed using SigmaPlot (version 13; Systat Software, USA) and described in the corresponding figure legends. In figures, "*" and "#" were used to indicate statistical significance ($p$ presents the level of statistical significance, $*p < 0.05$, $**p < 0.01$, $***p < 0.001$; $^{\#}p < 0.05$, $^{\#\#}p < 0.01$, $^{\#\#\#}p < 0.001$). All statistical comparisons of position- and speed-decoding parameters derived by linear regression analysis and context-decoding parameters evaluated by the discriminant analysis were performed by the two-way repeated measures ANOVA with Holm–Sidak

post-hoc method (XLSTAT, Addinsoft). All statistical tests are two-sided. Exact $p$-values can be found in Supplementary Dataset 1.

**Reporting summary**. Further information on research design is available in the Nature Research Reporting Summary linked to this article.

## Data availability
The source data underlying all figures are provided as a Source Data file. The raw $\Delta F/F$ and speed values at each position bin along the virtual corridor, the raw $\Delta F/F$ and the amplitude after deconvolution at each time bin during ITI used in this study are publicly available in the Mendeley Data database at https://data.mendeley.com/datasets/cpgrd49h85/2. All two-photon images are publicly available in the Mendeley Data database at https://data.mendeley.com/datasets/dcv885kzf2/1 (day 1), https://data.mendeley.com/datasets/k9z74vfgwf/1 (day 4), and https://data.mendeley.com/datasets/hjgtsmyyv5/1 (day 7). Source data are provided with this paper.

## Code availability
The code used to synchronize two-photon imaging with behavioral data, and to calculate the speed from Phenosys data are available on GitHub (https://github.com/seobseob/MATLAB-code).

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

## Acknowledgements

We thank Katrin Boehm for the technical support and Artem Turetskyy for providing a part of the custom codes. This work has been supported by the State Scholarship Fund of the China Scholarship Council (grant number 201406170032 to W.S.), the European Regional Development Fund (ERDF: Center for Behavioral Brain Sciences ZS/2016/04/78113 to J.P.), the Deutsche Forschungsgemeinschaft (project B14 in SFB779 to A.D., SFB1436 with Project-ID 425899996 in project A05 to A.D. and B06 to J.P.; Po732 to E.P.), and funds from the German Center for Neurodegenerative Diseases (to A.D.).

## Author contributions

W.S., S.S., O.S., and A.D. designed the study. W.S., S.S., and A.D. contributed to the establishment of the methods and setup for the 2PM-VR system. W.S. performed all experiments. W.S., J.P., and A.D. wrote the manuscript. W.S. and I.C. analyzed the data with contributions from J.P., E.P., and A.D. Y.W. contributed to the implementation of the virtual reality paradigm. All authors edited the manuscript.

## Funding

## Competing interests

Y.W. owns PhenoSys equity. The remaining authors declare no competing interests.
