## [Peer Review File · Nature Communications]

Context Value Updating and Multidimensional Neuronal Encoding in the Retrosplenial CortexReviewers' comments:

Reviewer #1 (Remarks to the Author):

The authors investigated in mice the role of the retrosplenial cortex in the acquisition and recall of spatial contextual associations. Using a head-fixed VR assay, they had mice move repeatedly through 1 of up to 3 distinct visual VR environments (contexts) to reach a goal location with distinct reward associations (rewarded or not-rewarded contexts). Following repeated exposure to the environments, the mice begin to slow down their running in proximity of the goal location and do so selectively for the rewarded environment, a reward location association which can be changed by changing the rewarded environment. Interestingly, using chemogenetic silencing, they show that RSC activity is associated with acquisition of these associations (they don't learn to slow down upon silencing) but not with the recall of previously-learned associations (they still slow upon silencing). The authors then go on using 2-photon calcium imaging characterizing cellular activity in RSC during the task with focus on encoding of sensory and task variables.

This is a carefully-done study on an interesting topic. The approach is sound and the data interesting. The manuscript is well written. The behavioral experiments, thanks to the VR approach, have a level of control that is rare in the field. The task put forward by the authors is simple and has potential for broader use. The chemogenetics experiments provide an indication of a role a role of the RSC in the task, which raises the interest substantially. While this may not be entirely unexpected, the specificity to acquisition is interesting, and it's probably the first time that such a direct link is demonstrated with such a level of experimental control, which increases the potential impact. However, I have several concerns with the analysis of the behavioral data which I lay out below.

The results of the calcium imaging seem comparatively weak. There are issues with analysis and experimental design, and with how the neural coding questions are addressed, which I also lay out below. Moreover, the overall motivation, the relation to the first part off the paper, and the significance of the results are not clearly explained. The focus being solely on encoding of task variables with little link with the animal's behavior the relation to the first part of the paper seems tenuous. The lack of link to behavior performance feels like a lost opportunity given the likely richness of the acquired data.

Main comments

1. The study uses average running speed across trials as sole behavioral readout. This is convenient but the metric is not adequately validated and has clear weaknesses. With regards to validation, different speed trajectories could lead to the same average speed which can result in ambiguities. Very little behavioral data is shown and the extent to which the metric captures the differences between experimental conditions is not shown. As a result, the extent to which the behavior is affected by learning or the chemogenetic manipulations is not clear. Furthermore, the behavior during the inter-trial interval is not characterized although it is used in later parts of the paper. It would be important to show more of the behavioral data, more trajectories at different time points during learning and more trajectories under distinct chemogenetic manipulations along with the associated behavioral quantification and to include the inter-trial interval in the data visualization and quantification.
2. With regard to weakness, the metric relies on trial averaging and therefore does not quantify single trial performance but it also does not take trial-to-trial variance into account hence and there is no quantification of overlap of behavior between experimental conditions. Trajectory shapes and trial-to-trial variance could vary significantly between experimental conditions and from the metric we would not know about it. Addressing this issue is important to understand the nature of the effects of learning and the chemogenetic manipulations. Furthermore, a better metric could potentially draw a direct link between neural activity and behavior .
3. The paper makes a point that RSC activity during the intertrial interval in darkness is predictive of the visual environment that the animal just visited presenting this activity as evidence of contextual

encoding. There are 2 issues with this. a) First, the experiments cannot distinguish between context and the behavioral responses to the reward. RSC receives a wide range of inputs (somatosensory, motor, neuromodulatory) and the reward may have long lasting effects on behavior such as increased movement speed, whisker activity, neuromodulatory inputs, which could be reflected in the activity. Here it would be important to show the behavior during those trials. b) Second, calcium responses themselves can be long lasting and responses to the reward likely are visible within the intertrial interval. I could not find it in the methods but it does not look that the calcium time courses have been deconvolved which would have removed some of the temporal dependencies.

4. The paper makes a point that the RSC shows activity linked to position in VR and that this activity is also informative of the specific VR environment the animal is visiting. The paper also makes multiple times the same point that cells that encode non-contextual variables such as running speed also carry information about context. The overall motivation behind these measurements is not made clear. RSC representations are intermingled so one expects some level of cross-talk between representations. Importantly, no links are drawn to behavior or learning which reduces to the overall interest.

Second, the experimental design cannot distinguish between context encoding and sensory responses to the visual environments which bear distinct visual features and therefore are expected to evoke distinct visual responses from RSC neurons. The experiments also cannot distinguish visually-related modulations, behavioral responses to the reward, and contextual modulations so those points are of limited value.

Finally, one needs to be extra careful when applying decoding approaches to calcium imaging data which is inherently ambiguous particularly for questions about modulatory influences. Dendritic signals, neuropil contamination, motion artifacts, for instance, could be correlated with context and induce spurious correlations that could lead to erroneous conclusions. While I trust the authors have been careful with analysis, there is not raw data shown that convince that the effects observed with the decoding actually reflect cellular spiking activity of the targeted population.

6. Some comments about figures, data visualization, particularly in the analysis of the behavioral data. These visualizations are too complex and need to be streamlined/optimized so the reader can quickly get through the essential points made by the data. The current figures are crowded and make a painful, time-consuming read. The illustrations of the experimental conditions in Figure 2a,c for example have too much redundancy in them which makes it unnecessarily difficult to read. The triple bar plots used to show the behavioral data (Figure 2, Figures S1-2) are extremely difficult to parse asking the reader to make multiway comparisons. The summary figures aim to address that (Fig. 2e-h, Figure S3) but they are not simple enough to allow efficient (e.g. symbol order changes unnecessarily across panels) and given the issues with the metric how they should be interpreted is not obvious.

Details:

Supplementary Videos: I could not find captions for these videos.

Supplementary Video 1. The images shown don't appear to have been corrected for brain movement. If the images are the images obtained post-registration, then the registration algorithms appear to have failed to have removed the motion artifacts.

Supplementary Video 2 is not particularly informative. There is no indication of progress within the corridor and when the reward is delivered, and it is not possible to assess whether the animals is slowing down in anticipation of the reward.

Page 4. To study neuronal activity in the RSC during context discrimination, Thy1-GCaMP6f transgenic mice were implanted with cranial windows, head-fixed, and pre-trained to reliably run on an air-cushioned spherical treadmill (Figures 1a, 1b, also see supplementary video 1). ... Via

the cranial window (Figure 1e), changes in fluorescence ($\Delta F/F$) were imaged using the genetically encoded calcium indicator GCaMP6f (Figures 1f, 1g, also see supplementary video 2) and signals were analyzed as a function of context, position, and speed during the sessions in the VR environment (Figure 1g).

References to supplementary videos 1 and 2 appear to have been inverted and don't match the submitted manuscript items.

Page 9: These data clearly demonstrate that these mice can perform the context discrimination if learning was already acquired, hence, DREADDs-mediated inhibition did not affect the ability of the mice to recognize the visual stimuli.

The latter part of this sentence is incorrect. It is unclear how the ability to recognize visual stimuli is involved. The metric is slow down in anticipation of reward delivery.

Page 9: Notably, on the first day of reversal learning while the RSC was inactivated (day 10), mice again reduced their speed in the newly rewarded context 2 to the same level as the previously rewarded context 1. This reward-related reduction of speed was specific to the anticipation zone (Figure S2c).

This statement does not seem correct. S2c shows a decrease in speed outside the activation zone relative to baseline (day 10 vs. day 1).

Page 10. Supplementary Figure 2. Mean running speed is not modulated by context or reward.

This statement is inaccurate. First, mean speed is consistently higher in the baseline session relative to subsequent sessions. Second, there are statistically significant differences in mean speeds between contexts in some sessions.

Page 10. Considering these results during learning and reversal learning sessions, we suggest that inactivation of the RSC affects the ability of mice to update the context value to make it negative.

The reference to 'learning' and 'reversal learning' in the first part of the sentence is confusing because the results of the 'learning' and 'reversal learning' phases are entirely consistent with one another. An important point to make seem to be that 'context 3' has been assigned a neutral value at day 8 (which corresponds to the learning phase) during which no CNO was applied. With regards the latter part, the parsimonious interpretation seem to be that inactivation prevents all updates, not only the negative ones.

Page 18. Additionally, there was no change in the modulation of the neuronal responses between context 1 and 2 (context modulation index; absolute value of $\text{context1} - \text{context2} / \text{context1} + \text{context2}$) across learning phases (Figure 3d),

The graph in Figure 3d actually shows a trend of increase although it might not be significant.

Reviewer #2 (Remarks to the Author):

In their manuscript (MS), Sun et al. study the behavioral role of retrosplenial cortex (RSC) during reward-associated contextual learning in a simple head-fixed virtual reality (VR) visual discrimination paradigm. They go on to record the neuronal activity of RSC in their paradigm to understand the potentially underlying changes in neuronal encoding of task-relevant features.

The main behavioral finding is that RSC is necessary during both acquisition of the initial contextual memory as well as during reversal learning of a newly rewarded context. In contrast to previous RSC lesion studies of contextual fear conditioning (Keene & Bucci, 2008), recall of these memories was not affected by inhibitory DREADD expression in the presence of CNO. Interestingly, CNO+DREADD animals were able to anticipate rewards but not to distinguish rewarded from non-rewarded contexts, which the authors term 'positive value bias'. Their main neuronal finding is that coding of three task-related features in dysgranular RSC (i.e. visual context, position, and speed) is largely unchanged during learning and reversal learning with the exception of a modest increase in the already high

fraction of 'multidimensional' neurons encoding all three analyzed task dimensions.

The RSC has started to draw considerable attention in recent years due to its newly appreciated role in learning and memory (L&M), navigation, and neurological disorders. The current study therefore is timely and both the behavioral and neuronal data are interesting and novel in their own regard. However, it is unclear how these findings relate to each other, to which extent the main neuronal finding of widespread mixed selectivity is non-trivial, why no longitudinal single-cell data is presented, and why the authors refrained from using standard head-fixed VR procedures like VR decoupling, VR gain modulation, and lick-scoring. I discuss these points below.

Main points

- The behavioral and neurophysiological part of the study appear disjunct and the narrative of the MS is sometimes meandering. Most importantly, it is unclear how minor changes in the encoding of different task dimensions relate to the behavioral finding of a role of RSC in context value updating. The main learning-related neuronal changes the authors observe are i) a change in context discriminability in the intertrial intervals (Fig. 3), and ii) an increase in the fraction of neurons with mixed selectivity specifically after reversal learning (Fig. 6). Whereas the first finding is unlikely to contribute to contextual encoding because it is easily explainable by ongoing post-reward activity, it is unclear to me how the second finding of an increase in context-unspecific multidimensional coding relates to the observed behavior. In fact, the probably more interesting finding matching their behavioral observation is the absence of observable context-related decoding differences during learning and relearning (Fig. 3cdi). Setting aside the fact that the chemogenetic interventions seem to address both the granular and dysgranular part of RSC (Fig. S7), a lack of persistent learning-related changes is consistent with the observation of a lack of effect of DREADD/CNO on the expression of contextual memory. RSCdys may therefore have a permissive role during learning of the paradigm but may not be part of the actual consolidated contextual memory trace. I suggest to phrase the interpretation of the neuronal data far more carefully and refrain from highly speculative interpretations in the context of the ill-described concept of 'cognitive flexibility' like "This [use of multiple parameters in the environment] is a role previously assigned to RSC [...], and we have now demonstrated this on a neuronal level."

- Neurons in RSCdys have frequently been observed to encode visual, self-motion, head-direction, vestibular, and positional information and combinations thereof (the most recent reports (Fischer, Mojica Soto-Albors, Buck, & Harnett, 2020; Mao, Molina, Bonin, & McNaughton, 2020; Powell et al., 2020; Vélez-Fort et al., 2018; Voigts & Harnett, 2020) are not mentioned in the MS but should now be cited and, when necessary, discussed). The most important new finding therefore would be the high percentage (30%) of 'multidimensional neurons' and their increase (to 40%, Fig. 6) after reversal learning. Mixed selectivity is certainly not surprising for a structure with afferent and efferent connectivity as varied as the RSC's. But indeed: a gain in mixed selectivity could be an interesting finding in the context of L&M, given its postulated role in increasing the richness of potential input/output transformations (Rigotti et al., 2013). However, I am skeptical if the authors show mixed representations or rather were unable to experimentally untangle the highly correlated task-dimensions they chose to investigate. Running speed is a covariate of visual responsiveness via visual flow. Similarly, position along a linear VR track is notoriously difficult to disentangle from simple visual responses to VR features. For instance, the double-peaked periodicity of many putative place cells in Fig. S6 could also be the result of on/off responses to the non-uniform visual features present in all contexts. Furthermore, vision is modulated by behavioral context such as running speed as early as in the thalamus (and potentially even on the level of retinal ganglion cells). It is not clear to me, why the authors did not try to take advantage of their VR setting to at least partially disentangle visual and self-motion signals using decoupled VR replay and treadmill-to-VR gain modulation as is common practice in the field, as has just recently repeatedly been shown also for visuo-spatial coupling in RSCdys (Fischer et al., 2020; Mao et al., 2020).

- It is unclear to me why the authors chose to not take advantage of their recording modality of choice

and analyze the short- and long-term stability of their representational classes. Chronic two photon Ca²⁺ imaging should have easily enabled them to ask whether or not the encoding classes they observe are stable across trials and days and whether or not neurons are recruited / silenced, stabilized / destabilized during learning and reversal learning. If the authors have this data, I strongly suggest to present it, also to distinguish this report from the four other recent publications (see above) with similar focus.

- Along similar lines: I do not understand why the authors chose to use a reduction in running speed before reward spout insertion as readout for reward anticipation. Running speed, again, is a covariate to vision and location. As the authors also state, the standard approach would have been to quantify reward-anticipation with licking as the motor output most closely related to reward consumption. Given that the authors state that their lick spout is also a lick sensor, the authors should explain their rationale for not using this data in the MS and ideally show lick responses when available. It would also be interesting to know why the lickspout was not present throughout the trials, enabling continuous lick recordings.

Minor points

- The finding of a role of RSC during memory formation but not expression is in disagreement with the conventionally assigned role of RSC as relevant for recall but not encoding during contextual fear conditioning (Corcoran et al., 2011; Keene & Bucci, 2008). This difference should be discussed in greater depth. Furthermore, it is unclear given these and other widely known studies why the authors claim that "[o]ur study is the first to explore the contribution of RSC in the contextual rather than cued learning".

- In the discussion the authors state that "single neurons in the RSC integrate multiple aspects of information while forming and updating context-reward value associations." This is misleading. None of the single cell data show robustly changing context-reward association (except in terms of post-reward ITI activity).

- The chemogenetic intervention addresses both granular and dysgranular RSC – the recordings, however, are exclusively from dysgranular RSC. The authors should state this clearly throughout and not speak of RSC in general when referring to their recordings in L2/3 of RSC_{dys}.

- The figures are often difficult to read. For example, most figures have recording day labels. It would be helpful if additionally the learning state would be indicated (e.g. day 1, baseline; day 4 post-learning, day 7 post-reversal). Furthermore, some figures display highly derived data to the extent of being cryptic (e.g. Fig. 3e, h). The authors should consider leaving these out.

- The linear models and procedures for cell selection and cross validation should be described in much greater depth in the methods.

- Authors should refrain from indicating 'barely non-significant' comparisons in their figures (e.g. Fig. S1c)

- When feasible, single observation points should be shown together with the bar graphs. Especially in the case of repeated measures data, individual connected animal datapoints are crucial to be able to judge across-animal variance.

- Neither in figures nor legends the authors indicate cell numbers and only show animal numbers. Most of the displayed data is subsampled. Especially in the case of the linear models where sum-of-squares analysis was used, the number of significantly coding neurons is important information. Does this change over the course of learning (e.g. does sparsity increase)?

- It would be interesting to see the within session trial-wise running speed data (similar as in Fig. 1) for

at least reversal learning in the presence of CNO (d9 vs. d10). Do mice slow down instantly after receiving reward in the new context? How does this develop during d10?

- Given the VR wall patterning used, visual responses to the three different visual contexts shown could easily be highly context-specific even with simple spatio-temporal visual receptive fields (S4e). It is unclear why the authors come to the conclusion “[...] indicating complex context-specific processing of visual information at the RSC rather than generalized responses to light [...]”.

- Fig S6: What do the authors mean with ‘classical place cell’ - I assume cells with only a single place response on the linear track? But many cells in Fig.S6 show clear double peaks.

- ‘Snake plots’ (peak-sorted positional responses) as in Fig.S6 should be sorted on one half of the trials and displayed using that sort-index for the other half (see (Saleem, Diamanti, Fournier, Harris, & Carandini, 2017)).

Literature

- Corcoran, K. A., Donnan, M. D., Tronson, N. C., Guzman, Y. F., Gao, C., Jovasevic, V., ... Radulovic, J. (2011). NMDA Receptors in Retrosplenial Cortex Are Necessary for Retrieval of Recent and Remote Context Fear Memory. *Journal of Neuroscience*, 31(32), 11655–11659. <https://doi.org/10.1523/JNEUROSCI.2107-11.2011>
- Fischer, L. F., Mojica Soto-Albors, R., Buck, F., & Harnett, M. T. (2020). Representation of visual landmarks in retrosplenial cortex. *ELife*, 9. <https://doi.org/10.7554/eLife.51458>
- Keene, C. S., & Bucci, D. J. (2008). Neurotoxic lesions of retrosplenial cortex disrupt signaled and unsignaled contextual fear conditioning. *Behavioral Neuroscience*, 122(5), 1070–1077. <https://doi.org/10.1037/a0012895>
- Mao, D., Molina, L. A., Bonin, V., & McNaughton, B. L. (2020). Vision and Locomotion Combine to Drive Path Integration Sequences in Mouse Retrosplenial Cortex. *Current Biology*. <https://doi.org/10.1016/j.cub.2020.02.070>
- Powell, A., Connelly, W. M., Vasalaukaite, A., Nelson, A. J. D., Vann, S. D., Aggleton, J. P., ... Ranson, A. (2020). Stable Encoding of Visual Cues in the Mouse Retrosplenial Cortex. *Cerebral Cortex*. <https://doi.org/10.1093/cercor/bhaa030>
- Rigotti, M., Barak, O., Warden, M. R., Wang, X.-J., Daw, N. D., Miller, E. K., & Fusi, S. (2013). The importance of mixed selectivity in complex cognitive tasks. *Nature*, 497(7451), 585–590. <https://doi.org/10.1038/nature12160>
- Saleem, A. B., Diamanti, E. M., Fournier, J., Harris, K. D., & Carandini, M. (2017). Coherent representations of subjective spatial position in primary visual cortex and hippocampus, 1–19. <https://doi.org/10.1101/235648>
- Vélez-Fort, M., Bracey, E. F., Keshavarzi, S., Rousseau, C. V., Cossell, L., Lenzi, S. C., ... Margrie, T. W. (2018). A Circuit for Integration of Head- and Visual-Motion Signals in Layer 6 of Mouse Primary Visual Cortex. *Neuron*, 98(1), 179-191.e6. <https://doi.org/10.1016/j.neuron.2018.02.023>
- Voigts, J., & Harnett, M. T. (2020). Somatic and Dendritic Encoding of Spatial Variables in Retrosplenial Cortex Differs during 2D Navigation. *Neuron*, 105(2), 237-245.e4. <https://doi.org/10.1016/j.neuron.2019.10.016>

Reviewer #3 (Remarks to the Author):

Sun et al use a combination of a virtual linear track, DREADD inactivation's and two-photon calcium imaging to show that RSC is critical for learning not to seek a reward in an unrewarded context. Interestingly, they found that single neurons in RSC encoding many dimensions of the task related variables. This suggests RSC (like other higher order cortical regions) is well positioned to integrate and translate information. In general, the study represents an extensive undertaking, is well thought out, analyses are rigorous and thorough, and the findings are quite interesting. I have a few mostly

minor concerns.

I am struggling with the reversal learning aspect of this study. First, reversal learning wasn't really covered in the introduction, it just suddenly appeared when summarizing the study (in the introduction section). Second, if I am following, the findings for reversal learning generally mirror what is seen with initial learning. So is there really a deficit or differential encoding for reversal learning or do the encoding and dysfunction following inactivation actually reflect the same processes as with initial learning?

The period used to assess contextual encoding includes when other differential behaviors were present, suggesting this encoding might not actually be contextual. This is acknowledged in the manuscript, but as far as I can tell these two possibilities are never explicitly teased apart. Is it possible the contextual encoding actually represents behavioral or other differences between the two contexts (e.g., slowing vs not)? The post reward period is also examined, but again there is the presence or absence of a reward that is confounded with context. It is possible I am missing something, but if not, I am wondering if it might be better to temper some of the language about contextual encoding.

Minor:

There were a few references to "slightly but significantly" (p. 17) or a change that wasn't significant. I would recommend being careful with such descriptions. At the very least, if I am following, the claim at the end of the second paragraph on p. 9 is not supported by the preceding results (perhaps it just needs moved?). A difference that is not significantly different (day 12) is not different than a second non-significant result (day 17), but I think it is suggested that there is a difference between day 12 and day 17.

Supplementary Fig. 2. The title of the figure does not match the graph (says not significant but significant results are indicated on the figure) and the figure also does not seem to match references to it in the manuscript. Is this the wrong figure?

Supplementary Fig 7 appears out of order (between supplementary figure 2 and 3).

Fig. 3a. I think part of the schematic is shifted.

p. 21 reference #32. I believe motion encoding in RSC was shown first by McNaughton et al., 1994, should probably include this reference.

The discussion describes dense projections from visual cortex to RSC and the potential role of these inputs. However, RSC is also strongly reciprocally connected to PC which contains visual information, single cells there encode with high dimensionality, and PC has been suggested to be critical for integration and transformations. Thus, it might be worth also discussing a potential role of PC inputs/outputs.

We thank the reviewers for their insightful comments and suggestions. We have revised the manuscript accordingly, adding a significant number of additional Figure panels and reworking the overall flow of the manuscript to further elucidate the link between our behavioural readouts and neuronal activity in the retrosplenial cortex (RSC). Furthermore, we have expanded our analyses on a number of fronts and provide more in depth discussion of the current advances in the field. The manuscript is now greatly improved and significantly more comprehensive; we have addressed all comments from the reviewers, with specific details below. *Reviewers' comments are indicated in italics.*

Reviewer #1:

The authors investigated in mice the role of the retrosplenial cortex in the acquisition and recall of spatial contextual associations. Using a head-fixed VR assay, they had mice move repeatedly through 1 of up to 3 distinct visual VR environments (contexts) to reach a goal location with distinct reward associations (rewarded or not-rewarded contexts). Following repeated exposure to the environments, the mice begin to slow down their running in proximity of the goal location and do so selectively for the rewarded environment, a reward location association which can be changed by changing the rewarded environment. Interestingly, using chemogenetic silencing, they show that RSC activity is associated with acquisition of these associations (they don't learn to slow down upon silencing) but not with the recall of previously-learned associations (they still slow upon silencing). The authors then go on using 2-photon calcium imaging characterizing cellular activity in RSC during the task with focus on encoding of sensory and task variables.

This is a carefully-done study on an interesting topic. The approach is sound and the data interesting. The manuscript is well written. The behavioral experiments, thanks to the VR approach, have a level of control that is rare in the field. The task put forward by the authors is simple and has potential for broader use. The chemogenetics experiments provide an indication of a role of the RSC in the task, which raises the interest substantially. While this may not be entirely unexpected, the specificity to acquisition is interesting, and it's probably the first time that such a direct link is demonstrated with such a level of experimental control, which increases the potential impact. However, I have several concerns with the analysis of the behavioral data which I lay out below. The results of the calcium imaging seem comparatively weak. There are issues with analysis and experimental design, and with how the neural coding questions are addressed, which I also lay out below. Moreover, the overall motivation, the relation to the first part of the paper, and the significance of the results are not clearly explained. The focus being solely on encoding of task variables with little link with the animal's behavior the relation to the first part of the paper seems tenuous. The lack of link to behavior performance feels like a lost opportunity given the likely richness of the acquired data.

We greatly appreciate this positive evaluation of our work. We also appreciate the constructive comments and have made a significant improvement to the manuscript in this regard, following the reviewers' suggestions. In short, we have further validated our behavioural metric and added a number of supplementary Figures in this regard, performed deconvolution of the calcium signal in the ITI period to demonstrate the results after further removal of any direct influence of the preceding reward, extended the detail in our description of the analysis, improved the justification and flow of the writing in the manuscript, and drawn more explicit links from changes in behaviour to our observed neuronal activity - strengthening the results of the calcium imaging in the study. Specific details in response to individual comments are listed below.

Main comments

1.1. The study uses average running speed across trials as sole behavioral readout. This is convenient but the metric is not adequately validated and has clear weaknesses. With regards to validation, different speed trajectories could lead to the same average speed which can result in ambiguities. Very little behavioral data is shown and the extent to which the metric captures the differences between experimental conditions is not shown. As a result, the extent to which the behavior is affected by learning or the chemogenetic manipulations is not clear. Furthermore, the behavior during the inter-trial interval is not characterized although it is used in later parts of the paper. It would be important to show more of the behavioral data, more trajectories at different time points during learning and more trajectories under distinct chemogenetic manipulations along with the associated behavioral quantification and to include the inter-trial interval in the data visualization and quantification.

We have added a number of additional figures to address these concerns. First, we have elucidated the entire speed trajectories under the various conditions (Supplementary Figs. 2, 9); demonstrating that on baseline day (Day 1) there is no difference in the speed trajectory along the entire track and on day 4 the speed trajectory is decreased on average even within the first 60 cm indicating the ‘recognition’ of the rewarded track context as it appears along the corridor walls and then speed decreases even further to reach a maximal separation from the other contexts as the animal approaches the reward in the anticipation zone. With further learning, on day 7, the speed trajectory gets even more ‘efficient’ with animals slowing down closer to the reward zone in the rewarded context, starting around 90-120 cm along the corridor and then, again, reaches a maximal separation from the other contexts within the anticipation zone. Therefore, our use of the speed within the anticipation zone is a reliable metric across learning phases and a good reflection of the animal’s anticipation of the upcoming reward. Further, when we inactivate RSC neurons in the DREADD condition after training with context 1 rewarded, the speed of all conditions is decreased indiscriminately along the corridors, indicating the positive value bias effect. Subsequently, after training with CNO-off under various conditions we see the separation of the speed trajectories, again with maximal separation within the anticipation zone, supporting our conclusions regarding the effects of RSC inactivation (see lines 294-300, 327-331), and indicating that the effects of the CNO are not specific to dampening motor responses. Therefore, we thank the reviewer for this suggestion as this full elucidation of the trajectories, and additional trial-by-trial data both across and within session (see point 1.2), further validates our use of the speed as our metric, and the results in the anticipation area are shown to be reflective of the task-linked motor behaviour, removing this ambiguity.

For the ITI region, this is a special case due to the parameters of the task. Since it is known in these types of navigational tasks in mice that they will slow down/stop when they are in the act of consuming a reward, we did not want the rewarded/non-rewarded contexts to differ specifically in this regard. Therefore, to control this across contexts, the treadmill was stopped with external breaks after the corridor section, the reward spout was extended in all contexts, and animals were given a reward following the appropriate context, which was immediately consumed, then the reward spout was retracted in all contexts and this was followed by the ITI region (lines 121-129, see also Fig. 2e). Therefore, the ITI represents a sort of delay/waiting period for the mice and should not have any systematic differences in behaviour during this period. Therefore, we do not have movement data from the spherical treadmill during this period and there is always a speed of zero within this region for all contexts. Hence, we did not characterize the behaviour in this regard during the ITI. Later in the manuscript, we use the

neuronal activity during this ITI period, which is not systematically affected by movement (i.e. although mice may move randomly, all animals are ‘stationary’ in VR coordinates), to report context decoding accuracy. It is true that the average neuronal activity in the ITI will inherently take into account differences in behaviour across contexts during the ITI, but we do not expect systematic differences in behaviour here and can not further tease apart specific behaviours on a finer scale with our current paradigm (see also point 1.3). We have also added a number of Figure panels to further demonstrate the pattern of activity in the ITI period and other sections of the VR track (see Fig. 2, Fig. 4, Supplementary Figs. 5, 6).

1.2 With regard to weakness, the metric relies on trial averaging and therefore does not quantify single trial performance but it also does not take trial-to-trial variance into account hence and there is no quantification of overlap of behavior between experimental conditions. Trajectory shapes and trial-to-trial variance could vary significantly between experimental conditions and from the metric we would not know about it. Addressing this issue is important to understand the nature of the effects of learning and the chemogenetic manipulations. Furthermore, a better metric could potentially draw a direct link between neural activity and behavior.

We have now provided additional information in this regard and strengthened our behavioural metric in a number of respects. First, we have added an additional supplementary figure showing the trajectory of the speed along the entire track, including the standard error across all trials for all animals (Supplementary Figs. 2, 9-11, see also response to comment 1.1 above). Secondly, we have provided additional information regarding the trial-by-trial variance as suggested by the reviewer in Fig. 2 (see also Supplementary Fig. 2). Here we show the distribution of speed in the anticipation zone for each trial, comparing each context, and across all experimental conditions (see also Supplementary Figs. 9-11 for additional evaluation of DREADD experiments). In this way, the overlap of behaviour is clearly shown for each experimental condition by the overlapping regions of the histograms on each day. This clearly demonstrates the strength of our behavioural metric on a trial-by-trial basis as there is a clear shift in the distribution to slower speeds for the rewarded context (Fig. 2c, Supplementary Figs. 2, 10). Lastly, we have also added the within session data across animals and days (Supplementary Figs. 2, 11). Together, we have now demonstrated the results of the behavioural metric across all levels of analysis and we believe this greatly strengthens the confidence in our conclusions.

With regard to the possibility of using a different metric for behaviour, please also see our response to point 2.7 where we show that licking behaviour (a metric used in similar paradigms) is also highly correlated to our metric of speed and hence is further validation.

1.3 The paper makes a point that RSC activity during the intertrial interval in darkness is predictive of the visual environment that the animal just visited presenting this activity as evidence of contextual encoding. There are 2 issues with this. a) First, the experiments cannot distinguish between context and the behavioral responses to the reward. RSC receives a wide range of inputs (somatosensory, motor, neuromodulatory) and the reward may have long lasting effects on behavior such as increased movement speed, whisker activity, neuromodulatory inputs, which could be reflected in the activity. Here it would be important to show the behavior during those trials. b) Second, calcium responses themselves can be long lasting and responses to the reward likely are visible within the intertrial interval. I could not find it in the methods but it does not look that the calcium time courses have been deconvolved which would have removed some of the temporal dependencies.

With regard to a), we have also addressed this in comment 1.1 and agree that it is difficult to dissect in fine detail the behaviour of the mouse during this period, however, since the animal

has already consumed the reward, the spout has been removed and the treadmill is held stationary during this period, we do not expect any systematic differences in the behaviour in this period across contexts (see Results lines 121-129). We have also added an additional Supplementary Fig. 5 showing cell-by-cell comparisons of activity between the corridor, reward and ITI periods, demonstrating that the pattern of activity in the ITI period is not a simple reflection of the reward response itself. Any systematic behaviours that may remain are likely part of the natural repertoire of behaviours that help to form context-value associations (potentially due to increased arousal etc) and, hence, are of importance to the natural learning process (see also Discussion lines 381-386). While we can not predict/control all these sources of input from our experiments, we can conclude that this activity as a whole is integrative in nature as we find that multidimensional cells encode this content-value association more efficiently than single dimensional cells (see Fig. 4).

In relation to point b) and the deconvolution of calcium signals, we had previously not used deconvolved signals as many of our parameters are slow-changing or non-discrete events (e.g. speed, spatial location) for which deconvolution can struggle to represent accurately. However, we agree that our ITI period does pose issues for the long decay times associated with calcium transients in the previous reward period. Therefore, we have added additional analysis for this ITI period using the deconvolved signal (Pnevmatikakis et al., 2016; Methods lines 630-635). Supplementary Fig. 6 has been added and shows the decoding accuracy during ITI using the amplitude of deconvolved peaks to predict the contexts. We found the same pattern of results with the deconvolution, even after this removal of potentially conflicting decay-time signals leftover from the reward zone as suggested by the reviewer. Additionally, during the ITI period, we show that there is no difference in the mean population activity level across contexts (rewarded vs non-rewarded; Fig. 2f, g), while the *pattern* of encoding across cells does change across contexts. If signals here were heavily contaminated with reward responses, we would expect an increased mean activity across the population as reward responses in RSC have been shown to be generally potentiating. This has been discussed in the revised manuscript (lines 176-191) and further implications elucidated in the Discussion (lines 432-442).

1.4. The paper makes a point that the RSC shows activity linked to position in VR and that this activity is also informative of the specific VR environment the animal is visiting. The paper also makes multiple times the same point that cells that encode non-contextual variables such as running speed also carry information about context. The overall motivation behind these measurements is not made clear. RSC representations are intermingled so one expects some level of cross-talk between representations. Importantly, no links are drawn to behavior or learning which reduces to the overall interest.

Although the behavioural paradigm is relatively simple, indeed as the reviewer suggests, there are many parameters and multiple behavioural dimensions occurring simultaneously in behaving animals, even in this simple task - and they all contribute to a sense of 'contextual awareness' for the animal. The motivation for our measurements was to monitor as many of these parameters as possible and examine the task-related changes that occurred with learning as well as the task-related patterns of neuronal activity in the RSC to determine how this contextual information was integrated across learning. To say that RSC representations are intermingled '*so one expects some level of cross-talk between representations*', may be true, but this statement gives us no information about the quality of this cross-talk or how it may actually be represented on the single-cell level or change with learning. One can not just assume the precise dynamics here without investigation. Here, we have performed a quantitative analysis of these representations during learning and reversal learning within the RSC and observed an increase in the number of multidimensional neurons encoding context, position and speed during relearning due to increased recruitment of cells into this category (new Fig.

4) and further that this multidimensional population shows increased decoding accuracy linked to the behaviourally relevant context (see new Fig. 4e-h). Although we agree that future experimental paradigms and manipulations could strengthen the link of these multidimensional cells to specific behavioural performance even further (with more targeted inactivations, see Discussion lines 448-453), we have indeed provided quantitative evidence of how this multidimensional ‘cross-talk’ is represented on the single cell level and could be important across complex learning (see also point 2.4). We have clarified our motivation further in the introduction (e.g. see lines 32-35) and throughout transition periods in the revised manuscript (e.g. Results lines 116-120, 247-253) and hope our changes throughout satisfactorily addressed the previous ambiguity.

1.5 Second, the experimental design cannot distinguish between context encoding and sensory responses to the visual environments which bear distinct visual features and therefore are expected to evoke distinct visual responses from RSC neurons. The experiments also cannot distinguish visually-related modulations, behavioral responses to the reward, and contextual modulations so those points are of limited value.

We agree that context encoding and visual sensory responses are inherently entangled in this task - as they are in all ‘navigation’ tasks of this sort including investigations of place cells, spatial maps and contextual environments in general. Hence, this is why we did not designate ‘visually responsive’ as a particular dimension separate from context or position or speed, as visual responses could potentially affect all of these properties and could not be fully disentangled in our experimental design. Equally, we did not want to ignore the fact that some RSC cells showed clear visually-evoked responses (as has also been explored in excellent previous studies) and so we instead show specific visually-related activity in relation to the corridor wall patterns and gave the reader an indication of what proportion of RSC cells also code for visual features (Supplementary Fig. 4). In this way, we identified RSC cells that are specifically and robustly activated by dark-light transitions and show that the identification of the specific corridors are predominantly and rather accurately encoded by these “visual” cells (independently of reward or learning; see results lines 145-162). Although, we also note that these parameters that are inherently entangled in naturalistic behaviours and the quality of this entanglement does not change across learning, so our main conclusions comparing effects across days as well as learning phases are equally influenced by visual processing and therefore remain reflective of experience-dependent plasticity within the RSC. Therefore, in the revised manuscript we have acknowledged these factors while still being able to draw conclusions that do not exaggerate our ability to distinguish these particular parameters (see also Discussion 406-411).

1.6 Finally, one needs to be extra careful when applying decoding approaches to calcium imaging data which is inherently ambiguous particularly for questions about modulatory influences. Dendritic signals, neuropil contamination, motion artifacts, for instance, could correlated with context and induce spurious correlations that could lead to erroneous conclusions. While I trust the authors have been careful with analysis, there is not raw data shown that convince that the effects observed with the decoding actually reflect cellular spiking activity of the targeted population.

Indeed, we are fully aware that these factors can contribute to correlated $\Delta F/F_0$ signals and we have processed the changes in fluorescence accordingly and with standardized and previously validated and published methods (e.g. cNMF method, Pnevmatikakis et al., 2016) as already described in the methods section (lines 629-642). In relation to the decoding results specifically, we agree that an explicit demonstration of the neuronal dynamics underlying the changes in decoding accuracy would further convince the reader that these effects are a reflection of changes in cellular activity within the population, and provide additional information on the

underlying change in population dynamics in the RSC with learning. Hence, we have added a number of additional figures to demonstrate this (e.g. see Fig. 2f, g, Fig. 4e, g, Supplementary Fig. 5, 6) and discuss the implications of the patterns in activity evident in this readout of neuronal activity on the single-cell and population level (e.g. see lines 166-175).

1.7. Some comments about figures, data visualization, particularly in the analysis of the behavioral data. These visualizations are too complex and need to be streamlined optimized so the reader can quickly get through the essential points made by the data. The current figures are crowded and make a painful, time-consuming read. The illustrations of the experimental conditions in Figure 2a,c for example have too much redundancy in them which makes it unnecessarily difficult to read. The triple bar plots used to show the behavioral data (Figure 2, Figures S1-2) are extremely difficult to parse asking the reader to make multiway comparisons. The summary figures aim to address that (Fig. 2e-h, Figure S3) but they are not simple enough to allow efficient (e.g. symbol order changes unnecessarily across panels) and given the issues with the metric how they should be interpreted is not obvious.

We understand this point of the reviewer, however, we must also note that we are working with multidimensional parameters across a large number of days and various learning parameters, hence, we want to represent data as comprehensively as possible in a succinct number of Figures. This invariable means that simplistic representations are not always sufficient to capture the effects and for representing all statistically significant findings. Regardless, we have made every effort to simplify Figures wherever possible and we believe we have succeeded in this regard. For example, as suggested, we have reworked Figure 2, in the process removing the redundancy, and believe this is visually improved. We understand that the triple bar plots are relying on the reader to make multiway comparisons, however, the nature of the data as well as the statistically relevant comparisons to be made, unfortunately necessitate this layout. For instance, if we separate these plots into three panels (separated by context) it would be inconvenient to then visualize meaningful statistical comparisons between the three contexts - which encompasses our main hypotheses. We have, however, rearranged the format of the new Fig. 5 to streamline the important comparisons into learning and experimental phases with CNO on and off and feel this is much improved. Further, the summary plots following these graphs maintain the order of context 1,2,3 and this is carried through for all figure panels, simplifying them as much as possible. Additionally, we have moved a number of graphs to Supplementary Figures, keeping only our main points in the main Figure panels and made every effort to repeat graphical motifs and styles across various representations of different parameters in order to minimize the number of new visualizations that the reader must interpret. We trust this has increased the general readability of the manuscript and Figure representations.

Details:

1.8 Supplementary Videos: I could not find captions for these videos.

The Supplementary Videos have been removed in the revision. See comment 1.9 and 1.10 below.

1.9 Supplementary Video 1. The images shown don't appear to have been corrected for brain movement. If the images are the images obtained post-registration, then the registration algorithms appear to have failed to have removed the motion artifacts.

The video shown was indeed the raw data before any post-processing or registration and was meant to be a simple demonstration of the quality of the raw data. We agree that actually this is of little interest and the example field-of-view presented in Figure 1f is perhaps more informative here, so we have removed the video. If the reviewers think this is necessary and

would like to see an example of our calcium imaging we can provide a video of the same field-of-view after registration.

1.10 Supplementary Video 2 is not particularly informative. There is no indication of progress within the corridor and when the reward is delivered, and it is not possible to assess whether the animals is slowing down in anticipation of the reward.

The intent with this video was to show the experimental set-up and how the animal runs through the virtual corridors in general. This is from an example video we captured during baseline day. Unfortunately, we do not have video recordings of the animals during the task showing behaviour from such an angle that you can see the entire screen, reward spout, and the animal etc, because of the location of the two-photon microscope and surrounding screens, this is not so easy to obtain without obstructing one or the other. In hindsight, we agree that this example is not so informative since we show the general set-up schematic in Figure 1b and it has been removed.

1.11 Page 4. References to supplementary videos 1 and 2 appear to have been inverted and don't match the submitted manuscript items.

As noted above, supplementary videos have been removed in the revision.

1.12 Page 9: "These data clearly demonstrate that these mice can perform the context discrimination if learning was already acquired, hence, DREADDs-mediated inhibition did not affect the ability of the mice to recognize the visual stimuli." The latter part of this sentence is incorrect. It is unclear how the ability to recognize visual stimuli is involved. The metric is slow down in anticipation of reward delivery.

We found that if an association was already formed between a context (as indicated by a particular visual pattern on the corridor walls) and the reward when the DREADDs were not activated, then even upon DREADD activation (and hence RSC inactivation), the animals could still successfully differentiate between the three contexts, as they slowed down preferentially in only the rewarded context (which is only demarcated by a different visual stimulus wall pattern). It is true that our metric of task performance is a reduction in running speed, but if this reduction in speed is context-specific then it is linked directly to visual recognition of the specific rewarded context. Therefore, it is logical to state that the mice still maintain the ability to differentiate between these visual stimuli that make up the corridor walls, otherwise we would see no difference in our performance metric between the rewarded and non-rewarded context as mice would have no way to predict which context was which by the time they get to the anticipation zone. This is not to say that there may be more subtle changes in visual perception with the inactivation of the RSC, but it is certainly fair to say that in this task, the mice can still perceive and have the ability to distinguish between the different visual patterns on the corridor walls even when RSC is inactivated after previous learning. We have clarified the wording in the manuscript (lines 304-307), which now reads: "This also indicates that mice could still recognize the individual contexts (i.e. as defined by the visual stimuli presented on the corridor walls) when RSC was inactivated, hence, learning deficits in the initial phase with CNO-on cannot be explained simply by a general deficit in visual perception."

1.13 Page 9: Notably, on the first day of reversal learning while the RSC was inactivated (day 10), mice again reduced their speed in the newly rewarded context 2 to the same level as the previously rewarded context 1. This reward-related reduction of speed was specific to the anticipation zone (Figure S2c). This statement does not seem correct. S2c shows a decrease in speed outside the activation zone relative to baseline (day 10 vs. day 1).

We agree that this statement was confusing as we meant to convey simply that this effect was clearly seen in the anticipation zone, but as the reviewer points out, and is also evident from

our new Supplementary Fig. 9, the mice generally reduce their speed during the presentation of context 2 and the general trajectory of speed is not significantly different between context 1 and 2 on day 10. This has been made clear in the revised manuscript (lines 311-319).

1.14 Page 10. Supplementary Figure 2. Mean running speed is not modulated by context or reward. This statement is inaccurate. First, mean speed is consistently higher in the baseline session relative to subsequent sessions. Second, there are statistically significant differences in mean speeds between contexts in some sessions.

We agree that this Figure was not an effective way to present this speed data. We have replaced this figure with Figures that more comprehensively show the speed dynamics, both in the average speed trajectory across the whole corridor as well as the trial-by-trial and within session speed in the anticipation zone across all experimental conditions (see Supplementary Figs. 2 and 9-11; see also point 1.2). We have also commented further on the observed effect that baseline speeds are higher relative to early learning sessions, specifically with regard to a default positive-value bias that is evident in early learning phases (lines 287-294).

1.15 Page 10. “Considering these results during learning and reversal learning sessions, we suggest that inactivation of the RSC affects the ability of mice to update the context value to make it negative.” The reference to ‘learning’ and ‘reversal learning’ in the first part of the sentence is confusing because the results of the ‘learning’ and ‘reversal learning’ phases are entirely consistent with one another. An important point to make seem to be that ‘context 3’ has been assigned a neutral value at day 8 (which corresponds to the learning phase) during which no CNO was applied. With regards the latter part, the parsimonious interpretation seem to be that inactivation prevents all updates, not only the negative ones.

It is true that the processes of learning and reversal learning are consistent with each other. However, the underlying requirements for these two learning processes differ, in that learning requires forming an association while reversal learning requires inhibition of the formed association as well as forming a novel association (hence reference to the need for ‘cognitive flexibility’, see also point 3.1) - and there are subtle but important differences between these phases in relation to the inactivation of the RSC.

On the first learning day (Day 2), mice slow down in both contexts in comparison to Baseline (Day1). Then context 2 (the non-rewarded context) becomes devalued as learning proceeds in the control animals, so that by Day 4, mice are running faster through the non-rewarded context. Mice also generalize to the neutral context 3 here, and also run faster through it. So there is no positive-bias and mice have devalued both the non-rewarded and neutral contexts equally. With the RSC inactivated (CNO-on), by the test day 4, mice are still slowing down in anticipation of all contexts, i.e. they have a positive-value bias and have failed to devalue both the non-rewarded and the neutral contexts. Here one could indeed argue that inactivating the RSC prevents all updates, with a ‘default’ positive-value bias assigned to all contexts. The mice are then trained with CNO-off and learning proceeds normally, with both the non-rewarded and the neutral context devalued equally.

Then reversal learning begins with CNO-on and the RSC inactivated (Day 10). If there was a general inability to update indiscriminately, we would see that mice would continue to run fast through the previously non-rewarded context (context 2), as was learnt during Day 8/9. However, we see again that mice have assigned a positive-value bias to both contexts with the RSC inactivated. This is actually similar again to what we see in Day 5 reversal learning in control mice (see also Supplementary Fig. 12). The difference being that as reversal learning proceeds in control mice (Day 6), the previously rewarded context is then devalued and mice begin to run faster through context 1, and by Day 7, this previously rewarded context is devalued to again resemble the neutral context. In the RSC inactivated mice, the previously

rewarded context 1 is maintained as valued throughout the CNO-on phase and therefore, there is no significant difference in running speed between the previously and newly rewarded contexts (context 1 and 2). Therefore, with RSC inactivated, mice have failed to devalue the previously rewarded context. In relation to context 3, with the RSC inactivated, we do not see this context adopt a significant positive-value bias as it did during the initial learning phase with CNO-on (Day 4), but the association of this context as a neutral context from Day 8 is maintained on the test day even after RSC inactivation. There is no need for context 3 to be 'updated' from Day 8 to Day 12, as it remains neutral throughout, but this information is retained across Day 8-12 even with the RSC inactivated. Therefore, this is not evidence that inactivation prevents all updates, as context 3 did not need to be updated, but it was also not assigned a positive-value bias, as it was from Day 1 to Day 4, indicating that the status as neutral was retained even with RSC inactivation. Therefore, from Day 8/9 to Day 10, with RSC inactivated, context 2 is updated to be assigned a default positive-value bias, context 3 retains a neutral distinction, but context 1 is not updated to be devalued or assigned a negative value distinction.

We have made substantial revisions to this section of the results to make this point more explicit and clearer in the revised manuscript (lines 282-331). We also feel that the inclusion of the speed trajectories in Supplementary Fig. 9 as well as the trial-by-trial data in Supplementary Figs. 10-11 will aid the reader in this interpretation; here the full dynamics can be more easily compared across conditions - so we again thank the reviewer for this earlier suggestion.

1.16 Page 18. Additionally, there was no change in the modulation of the neuronal responses between context 1 and 2 (context modulation index; absolute value of $\text{context1} - \text{context2} / \text{context1} + \text{context2}$) across learning phases (Figure 3d), The graph in Figure 3d actually shows a trend of increase although it might not be significant.

Indeed there may have been a trend here, but the results were not significant and so we have made our interpretations based on this. As per reviewer comment 3.3, we are encouraged to refrain from commenting on results that show general trends but are not statistically significant. Additionally, point 2.11 suggests we remove these context modulation figures; therefore, these have been removed in the revised manuscript and the representations of the neuronal responses to each context have been simplified and presented in a more straightforward way to demonstrate more clearly the pattern of activity across contexts for individual cells (see Fig. 2f, g).

Reviewer #2 (Remarks to the Author):

In their manuscript (MS), Sun et al. study the behavioral role of retrosplenial cortex (RSC) during reward-associated contextual learning in a simple head-fixed virtual reality (VR) visual discrimination paradigm. They go on to record the neuronal activity of RSC in their paradigm to understand the potentially underlying changes in neuronal encoding of task-relevant features. The main behavioral finding is that RSC is necessary during both acquisition of the initial contextual memory as well as during reversal learning of a newly rewarded context. In contrast to previous RSC lesion studies of contextual fear conditioning (Keene & Bucci, 2008), recall of these memories was not affected by inhibitory DREADD expression in the presence of CNO. Interestingly, CNO+DREADD animals were able to anticipate rewards but not to distinguish rewarded from non-rewarded contexts, which the authors term 'positive value bias'. Their main neuronal finding is that coding of three task-related features in dysgranular RSC (i.e. visual context, position, and speed) is largely unchanged during learning and reversal learning with the exception of a modest increase in the already high fraction of 'multidimensional' neurons encoding all three analyzed task dimensions. The RSC has started

to draw considerable attention in recent years due to its newly appreciated role in learning and memory (L&M), navigation, and neurological disorders. The current study therefore is timely and both the behavioral and neuronal data are interesting and novel in their own regard. However, it is unclear how these findings relate to each other, to which extent the main neuronal finding of widespread mixed selectivity is non-trivial, why no longitudinal single-cell data is presented, and why the authors refrained from using standard head-fixed VR procedures like VR decoupling, VR gain modulation, and lick-scoring. I discuss these points below.

We appreciate this general evaluation of our work as timely and the statement that both the behavioral and neuronal data are interesting and novel. We also thank the reviewer for providing detailed recent literature citations, all of which we have incorporated into the revised manuscript. Indeed, at the time of our initial submission in Nov 2019 - as there was a substantial period of time passed between our initial submission and the review process due to some unfortunate administrative delays - many of these new 2020 studies were not yet published, but we have now incorporated these excellent resources into the revised manuscript and are pleased to see all the advances in the field.

In relation to the reviewers main concerns regarding the lack of longitudinal single-cell data as well as other aspects related to the experimental set-up, we can only put forth in all honesty, that although the authors have extensive previous imaging experience, this was the first set of experiments performed on a new in vivo two-photon behavioural set-up in the lab. While this is in no way reflected negatively in the *quality* of the data collected, which was at a high standard, there were certain limitations in the breadth of the functions that were fully implemented at the early stages of this experimental equipment. For instance, and as explained in detail below, the data from the lick sensor was not fully integrated at the time and the complexity of the experimental design and protocols run through the PhenoSys software limited the more advanced functionality of the VR procedures. As in all advanced imaging set-ups, the system has since grown in functionality, however, we are fully confident in the quality of our original dataset and even more so now since we have replicated these results, including adding new analysis to track single-cell activity longitudinally in a subset of our animals as suggested by the reviewer (see points 2.6, Fig 4d) and further validated our original behavioural metrics in comparison to data collected from an integrated lick sensor during the same learning task for a separate project (see point 2.7). We have unfortunately been somewhat limited in the volume of new experiments we could conduct recently, as have many labs, due to the ongoing COVID restrictions in 2020, but we have presented validating data from new animals as well as extensive additional analysis to the revised manuscript and have completed major revisions to the impact of our results as well as the general flow of the text. We feel as though we have addressed the main concerns of the reviewer adequately and to the best of our current ability and detail our changes in the points below.

Main points:

2.1 The behavioral and neurophysiological part of the study appear disjunct and the narrative of the MS is sometimes meandering. Most importantly, it is unclear how minor changes in the encoding of different task dimensions relate to the behavioral finding of a role of RSC in context value updating. The main learning-related neuronal changes the authors observe are i) a change in context discriminability in the intertrial intervals (Fig. 3), and ii) an increase in the fraction of neurons with mixed selectivity specifically after reversal learning (Fig. 6). Whereas the first finding is unlikely to contribute to contextual encoding because it is easily explainable by ongoing post-reward activity, it is unclear to me how the second finding of an increase in context-unspecific multidimensional coding relates to the observed behavior.

We have made every effort to streamline the narrative of the manuscript and have made major revisions throughout to help with the flow and clarity. In relation to finding i), while the reviewer speculates that this is easily explained by post-reward activity, in fact we do not find that this is the case and have now shown that the ability to decode the rewarded context from the RSC population activity in the ITI, even when no overt contextual stimulus remains, is still significant after deconvolution of the signal (removing even more of the potential ‘direct’ reward influence, see also point 1.3), and is due to more complex underlying changes in the pattern of neuronal activity across the population during this time for all contexts, not just the rewarded context (i.e. the pattern of activity between the neutral and non-rewarded context being less distinguishable from each other than to that of the rewarded context, without a change in the mean population activity level across any context - which is reflective of the task behaviour). This is of particular interest because, contrary to the reviewers original speculation, we find that this effect does contribute to contextual encoding, specifically of the valued (rewarded) context in comparison to the non-rewarded and also the neutral contexts together; revealing a sharp dependence of the accuracy of decoding on reward, but with an indirect relationship and even when no overt visual/contextual cues remain. Importantly, we also find that the populations of defined multidimensional neurons more efficiently encoding these context-value associations in comparison to single dimensional encoding cells that do not (Fig. 4e-h). Hence, we have added new Figures (Figs. 2g, 4e-h, Supplementary Figs. 5, 6) demonstrating this effect and amended the manuscript accordingly throughout (e.g. lines 163-191).

2.2 In fact, the probably more interesting finding matching their behavioral observation is the absence of observable context-related decoding differences during learning and relearning (Fig. 3cdi). Setting aside the fact that the chemogenetic interventions seem to address both the granular and dysgranular part of RSC (Fig. S7), a lack of persistent learning-related changes is consistent with the observation of a lack of effect of DREADD/CNO on the expression of contextual memory. RSCdys may therefore have a permissive role during learning of the paradigm but may not be part of the actual consolidated contextual memory trace.

We agree that this particular point was not made clear and have added this explicitly to the discussion in relation to both the effects we see in the virtual corridor and the activity in the ITI period, which has also been clarified throughout (see Discussion lines 425-433)

2.3 I suggest to phrase the interpretation of the neuronal data far more carefully and refrain from highly speculative interpretations in the context of the ill-described concept of ‘cognitive flexibility’ like “This [use of multiple parameters in the environment] is a role previously assigned to RSC [...], and we have now demonstrated this on a neuronal level.”

We have made substantial efforts in the revised manuscript to provide a more direct interpretation of the neuronal data and avoid speculative statements, including removing the suggested above statement. We have added a number of Figures that better demonstrate the representation of contextual information within the RSC neuronal population directly (see e.g. new Fig. 2 and 4, Supplementary Figs. 5, 6) and we have better defined the concept of cognitive flexibility, which is a well established term used in the field with reference to reversal learning tasks (Introduction lines 42-49).

2.4 Neurons in RSCdys have frequently been observed to encode visual, self-motion, head-direction, vestibular, and positional information and combinations thereof (the most recent reports (Fischer, Mojica Soto-Albors, Buck, & Harnett, 2020; Mao, Molina, Bonin, & McNaughton, 2020; Powell et al., 2020; Vélez-Fort et al., 2018; Voigts & Harnett, 2020) are not mentioned in the MS but should now be cited and, when necessary, discussed). The most important new finding therefore would be the high percentage (30%) of ‘multidimensional

neurons' and their increase (to 40%, Fig. 6) after reversal learning. Mixed selectivity is certainly not surprising for a structure with afferent and efferent connectivity as varied as the RSC's. But indeed: a gain in mixed selectivity could be an interesting finding in the context of L&M, given its postulated role in increasing the richness of potential input/output transformations (Rigotti et al., 2013).

We thank the reviewer for their comment and suggestions regarding new 2020 studies. Indeed, the majority of these excellent papers were published after our submission. We have now cited and discussed each appropriately (e.g. see lines 36-40, 148-151, 387-398) and agree that the highest impact of our study lies in the dynamic between the multidimensional neurons and learning phases. We have significantly strengthened this point throughout the manuscript.

2.5 However, I am skeptical if the authors show mixed representations or rather were unable to experimentally untangle the highly correlated task-dimensions they chose to investigate. Running speed is a covariate of visual responsiveness via visual flow. Similarly, position along a linear VR track is notoriously difficult to disentangle from simple visual responses to VR features. For instance, the double-peaked periodicity of many putative place cells in Fig. S6 could also be the result of on/off responses to the non-uniform visual features present in all contexts. Furthermore, vision is modulated by behavioral context such as running speed as early as in the thalamus (and potentially even on the level of retinal ganglion cells). It is not clear to me, why the authors did not try to take advantage of their VR setting to at least partially disentangle visual and self-motion signals using decoupled VR replay and treadmill-to-VR gain modulation as is common practice in the field, as has just recently repeatedly been shown also for visuo-spatial coupling in RSCdys (Fischer et al., 2020; Mao et al., 2020).

Indeed in these experimental paradigms it can be difficult to experimentally separate visually-based correlated task dimensions. As mentioned earlier, it is unfortunate that at the time of experiments, gain modulation of the system was not yet implemented in the software and further experiments altering this parameter at this time have been difficult due to resulting COVID restrictions. Regardless, we believe there are a few key points that allowed us to disentangle responses to enable us to make solid conclusions regarding the nature of the observed increase in multidimensional encoding cells with more complex learning. First, this interdependence of parameters would be the same during both learning and relearning phases of the experiment, the coupling of visual flow and running speed for instance will not change here, therefore, any changes we observed in the proportion of multidimensional encoding cells will be independent of these correlated task-dimensions (e.g. see lines 265-271). Additionally, in relation to reward-related value associations, the behaviour changed as the context (and therefore visual stimuli) changed across days, and also a proportion of responses changed with the valued (rewarded) context in the ITI period even in the absence of visual stimuli (see new Fig. 2). Therefore, this representation of the valued/non-valued and neutral contexts were not simply responses to the visual stimulus itself.

That being said, this is precisely the reason that we did not represent responses to visual properties (e.g. see Supplementary Fig. 4) as a singular dimension in our multidimensional analysis. This represents the most difficult factor to disentangle (for context, speed and position), as the reviewer points out. However, we have made significant efforts to include examples of the typical responses that underlie the single-dimensional responses of all of these parameters (e.g. Fig. 3, Supplementary Fig. 3, 4, 7) and it is clear that the pattern of responses is altered in unique ways for each dimension. Therefore, if some cells in RSC can represent a single dimension at all, it logically follows that the RSC can separate these parameters (and does in certain populations of cells, see also proportions of single dimension encoding cells in Fig. 4). If the RSC *can* separate out these parameters, then it also logically follows that if a cell's activity does not represent a clear single dimension parameter then there are levels of

integration on the single cell level (regardless of whether we can separate them experimentally or not) and therefore this is the natural basis for multidimensional encoding. Whether we could separate out the contribution of visual input with gain modulation, for instance, will not change the fact that in an environment where an animal is naturally navigating, these dimensions are overlapping within the encoding of single cells. We have made this argument more precisely in the revised manuscript to try to represent the strength of our conclusions as well as these experimental caveats (lines 406-414).

2.6 It is unclear to me why the authors chose to not take advantage of their recording modality of choice and analyze the short- and long-term stability of their representational classes. Chronic two photon Ca²⁺ imaging should have easily enabled them to ask whether or not the encoding classes they observe are stable across trials and days and whether or not neurons are recruited / silenced, stabilized / destabilized during learning and reversal learning. If the authors have this data, I strongly suggest to present it, also to distinguish this report from the four other recent publications (see above) with similar focus.

There were two main reasons that we did not previously present cell-matched chronic imaging data. Firstly, the analysis pipeline we used for preprocessing of our imaging data (cNMF method, Pnevmatikakis et al., 2016), while well respected, highly utilized in the field, and the most advanced processing method available at the time (published Jan 2016), was not readily applicable to chronic imaging datasets as the cells (ROIs) are segmented based on spatio-temporal characteristics within each imaging session. This was a technical limitation and we decided that for our purposes the important aspect was to highlight the information present within the population of RSC cells (i.e. decoding parameters, relative proportion of dimensions represented, etc), rather than the day by day single-cell dynamics. Secondly, for our imaging dataset one animal had a slightly different field of view in the z-axis on the baseline (day 1) compared to the other testing days (day 4 and Day 7). Therefore, to preserve our full dataset and address our hypotheses which were formulated around population dynamics, we did not further pursue single-cell matching across days. However, this does not mean that we perceive single-cell dynamics to be inconsequential and we fully agree with the reviewers' suggestion that examining these dynamics would be to take full advantage of our chronic imaging. Hence, we have reassessed our raw data through a relatively newer analysis pipeline (suite2p, Pachitariu et al., 2016; published on bioRxiv in June 2016 and further highly developed at Janelia since 2018), which we found to be more amenable to chronic imaging datasets. We have added an additional analysis specific to the single-cell matched imaging analysis, where we show the re-mapping properties of neurons across dimensions and show that the multidimensional neurons are the most stable across learning phases and that significantly more single/double dimension neurons get recruited to multidimensional neurons following reversal learning (Fig. 4d). However, we have maintained our original analysis pipeline in the main Figures, as this allows for the inclusion of all five mice.

2.7 Along similar lines: I do not understand why the authors chose to use a reduction in running speed before reward spout insertion as readout for reward anticipation. Running speed, again, is a covariate to vision and location. As the authors also state, the standard approach would have been to quantify reward-anticipation with licking as the motor output most closely related to reward consumption. Given that the authors state that their lick spout is also a lick sensor, the authors should explain their rationale for not using this data in the MS and ideally show lick responses when available. It would also be interesting to know why the lickspout was not present throughout the trials, enabling continuous lick recordings.

As mentioned above, unfortunately the lick sensory was not fully implemented at the start of experiments. We had some issues with the original vibration sensory that was supplied with the PhenoSys system as we found this was unreliable and we have since changed the lick sensor to

a capacitive touch sensor (see also Pakan et al., 2018), which improved reliability greatly. We have corrected the text in this regard, as it was misleading to state that the lick spout functioned as a sensor at the time of experiments (lines 522). During the pilot dataset, this change in running speed was observed and we found this to be a very robust metric for behaviour. We therefore decided to proceed with this metric and the full set of experiments, with the added benefit of quantifying and validating an alternative behavioural metric that could also prove to be useful for other working groups and future experiments. We have since fully implemented the new capacitive touch lick sensor in the experimental set-up and have also replicated the behavioural results using this paradigm for a separate project and a different set of mice. The results are presented below and demonstrate a high negative Spearman rank correlation coefficient between the number of licks and the mean running speed in the anticipation zone, indicating that lower speed is directly associated with higher number of licks and further validates our chosen behavioural metric. Since this is data from a separate project we have not included it in the manuscript at this time, and instead relied on improving our presentation of the reliability and validity of speed as our behavioural metric (new Fig. 2c and Supplementary Figs. 2, 9-11, see also points 1.1 and 1.2), however if the reviewers deem it necessary, we are happy to include this as an additional supplementary figure in the manuscript.

Figure. Spearman correlation between the number of licks and the log(mean running speed in the anticipation zone) from an additional dataset after learning (a) and reversal learning (b). Different colors code data from different animals, each point represents the mean speed and the number of licks in the anticipation zone per trial. The regression lines fit data for individual animals. (c) Demonstrates a significant negative correlation between the number of licks and the running speed during both learning phases.

Minor points

2.8 The finding of a role of RSC during memory formation but not expression is in disagreement with the conventionally assigned role of RSC as relevant for recall but not encoding during contextual fear conditioning (Corcoran et al., 2011; Keene & Bucci, 2008). This difference should be discussed in greater depth. Furthermore, it is unclear given these and other widely known studies why the authors claim that [o]ur study is the first to explore the contribution of RSC in the contextual rather than cued learning”.

We have further discussed the findings with regard to the role of the RSC in learning and memory, and specifically recall, in light of our study in comparison to the previous literature, as well as included the references suggested by the reviewer. In short, both of the noted references (and much of the previous literature) have utilized contextual fear conditioning as a

paradigm, and it is unclear how comparable contextual learning is in relation to fear conditioning versus appetitive conditioning. These involve an appetitive motivational system that mediates approach and reward versus an aversive motivational system that mediates defense and fear, and while there may be complex relationships between these two systems (e.g. Bulganin et al., 2014), and certainly the RSC plays a role in both (Smith et al., 2004), they are in fact mutually inhibitory (e.g. Dickinson and Dearing 1979, Belova et al., 2008). Additionally, the difference between a visuospatial task and cue-based association learning may also be significant given the role of RSC in spatial processing. However, the comparison remains of great interest and has been extensively discussed in the revised manuscript (Discussion lines 415-453). In light of our own results regarding multidimensional neurons, it is interesting to note that in fear conditioning experiments that specifically utilized multiple cues simultaneously, the RSC is necessary for memory formation as well (e.g. Keene & Bucci, 2008; Robinson et al., 2011).

We have additionally removed the noted statement above and revised the Discussion throughout to better reflect the novel contribution of our study to the, already knowledgeable, field in this regard. This particular statement was originally meant to be with regard to a ‘reversible’ chemogenetic approach rather than a lesion study and was in reference specifically to the preceding sentence.

2.9 In the discussion the authors state that "single neurons in the RSC integrate multiple aspects of information while forming and updating context-reward value associations." This is misleading. None of the single cell data show robustly changing context-reward association (except in terms of post-reward ITI activity).

We concede that we did not adequately represent the full nature of the signals within the ITI region in the previous version of the manuscript. We have revised the analysis (see also point 1.3), presentation (Fig. 2 and 4, Supplementary Figs. 5, 6), and interpretation (e.g. see results lines 163-197, 272-280) of this data and have amended this sentence to be more precise, noting that ‘neurons that showed multidimensional responses along the corridor also efficiently encoded the context-reward associations during the delay period between trials while neurons encoding for a single dimension along the corridor did not. This supports a translational role for the RSC in forming context-value associations based on integrative information.’ (lines 373-377). We have additionally included examples of neurons that we have followed across days that clearly show a context-specific change in responses according to reward (context-value associations).

2.10 The chemogenetic intervention addresses both granular and dysgranular RSC – the recordings, however, are exclusively from dysgranular RSC. The authors should state this clearly throughout and not speak of RSC in general when referring to their recordings in L2/3 of RSC_{dys}.

We clearly state this now at various locations throughout the manuscript and in the discussion we specifically point out the distinction in relation to chemogenetic experiments (lines 448-453).

2.11 The figures are often difficult to read. For example, most figures have recording day labels. It would be helpful if additionally the learning state would be indicated (e.g. day1, baseline; day 4 post-learning, day 7 post-reversal). Furthermore, some figures display highly derived data to the extent of being cryptic (e.g. Fig. 3e, h). The authors should consider leaving these out.

We have added additional labels to the figures according to this suggestion wherever feasible and included clearer descriptions in all figure legends. We have also have made a number of

changes to the figures to present more non-derived data along with our summary metrics (e.g. Fig. 2f, g, 4e, g, Supplementary Figs 3, 4a, 5, 6) in place of the more derived context modulation index graphs, which have been removed as suggested.

2.12 The linear models and procedures for cell selection and cross validation should be described in much greater depth in the methods.

We have extended the description of these methods in this regard (see lines 650-697).

2.13 Authors should refrain from indicating ‘barely non-significant’ comparisons in their figures (e.g. Fig. S1c)

We have removed this designation within the Figures and provided documentation of all actual p-values, see a supplementary file.

2.14 When feasible, single observation points should be shown together with the bar graphs. Especially in the case of repeated measures data, individual connected animal datapoints are crucial to be able to judge across-animal variance.

Where feasible, we have included this information (e.g. Fig 2d, f, g, Fig. 5b). Given the request of the reviewers to also simplify Figure illustrations wherever possible (see point 1.7) we have refrained from including this in some of the panels (e.g. Fig. 2b and Fig. 5a, d, e), as it would again lead to overly-complicated visualizations, and some plots represent comparisons across different animal groups and/or different populations of cells, but we have added this wherever possible.

2.15 Neither in figures nor legends the authors indicate cell numbers and only show animal numbers. Most of the displayed data is subsampled. Especially in the case of the linear models where sum-of-squares analysis was used, the number of significantly coding neurons is important information. Does this change over the course of learning (e.g. does sparsity increase)?

We originally discussed cell numbers largely in the *Data analysis* portion of the methods section, but we have now included additional Figures in which the cell numbers are more prominently displayed directly in Figures (e.g. Fig 2, 4) and have also added reference to them where appropriate in Figure legends. With regard to the number of significantly encoding neurons, we note that this information is provided for each parameter in the percentage of encoding cells (see Fig. 2i, k, Fig. 3d, i), including information on the dynamics of this proportion over the course of learning.

2.16 It would be interesting to see the within session trial-wise running speed data (similar as in Fig. 1) for at least reversal learning in the presence of CNO (d9 vs. d10). Do mice slow down instantly after receiving reward in the new context? How does this develop during d10?

We have now included this data and related measures in a number of new Figures (Supplementary Fig. 2, 9-11). An example of the within session running speed is now shown (Supplementary Fig. 11) for all days. Specifically, during the transition from day 9 to day 10 we observe that in the first few trials, with RSC blocked, mice continue to assign a positive-value bias to context 1 (continue to slow down), even after it has ceased to be rewarded. Additionally, within the first few trials context 2 is also assigned a positive-value bias, such that the speed matched that of context 1. Since positive-value bias occurs during normal learning as well (e.g. slowing down from baseline day to Day 2), we conclude that the major deficit in learning with the inactivation of the RSC is in devaluing the previously rewarded context, or in other words, a failure to update a positive to a negative context-value association (see Results 289-293, Discussion lines 396-398). We have also included two related Figures that more

comprehensively show the speed dynamics, both in the average speed trajectory as well as the trial-by-trial speed in the anticipation zone across all experimental conditions (see Supplementary Fig. 9 and 10).

2.17 Given the VR wall patterning used, visual responses to the three different visual contexts shown could easily be highly context-specific even with simple spatio-temporal visual receptive fields (S4e). It is unclear why the authors come to the conclusion “[...] indicating complex context-specific processing of visual information at the RSC rather than generalized responses to light [...]”.

Here we meant to convey that the processing of visual information was not a simple response to light (as would occur in a whole field flash on/off for instance). We completely agree with the reviewer that given spatio-temporally organized receptive fields the context-specificity could be established - our point being that there is at least a necessity for organized spatio-temporal visual receptive fields for RSC neurons. This is not a new finding, as visual receptive field properties have already been established in the RSC (e.g. Murakami et al., 2015 and Wang et al., 2012 which we had cited already and Powell et al., 2020 which we have added here); here, we confirmed this concept for a set of diverse virtual environments. We have amended this description in the revised manuscript (lines results 145-162).

2.18 Fig S6: What do the authors mean with ‘classical place cell’ - I assume cells with only a single place response on the linear track? But many cells in Fig.S6 show clear double peaks. ‘Snake plots’ (peak-sorted positional responses) as in Fig.S6 should be sorted on one half of the trials and displayed using that sort-index for the other half (see (Saleem, Diamanti, Fournier, Harris, & Carandini, 2017)).

By ‘classical place cells’ we meant as defined along similar criteria as previously published criteria (e.g. Mao et al., 2017, 2018), which generally means that place fields are singular within an environment, but this is not strictly an exclusion criteria. We described our specific criteria in the methods section. It is true that this method does not completely excluding the possibility of a ‘second’ peak for position encoding, but namely the criteria that ‘the mean in-field activity had to be at least three times higher than the mean out-of-field activity, and the position of the global maximum was required to be the same in >30% trials’ had to be met. This would exclude most ‘double peak’ cells, but if the response to the second position is much weaker than the first, the cells may still be classified as a ‘place cell’, as was the case in a subset of our position encoding neurons.

That being said, we agree with the reviewer that this is not a such a clear distinction and the aim of the study was not to specifically define ‘place cell’ activity, which is inherently more broad than that found in the hippocampus, but has also already been investigated in the RSC by previous excellent papers (e.g. Smith et al., 2012; Mao et al., 2017, 2018, etc). Therefore, we have removed this Supplementary Figure from the paper. If the reviewers determine it is vital information we are happy to put it back in with the additional snake plot sorting analysis, but otherwise we decided it is somewhat tangential to our main conclusions and would be better investigated with a targeted study.

Reviewer #3 (Remarks to the Author):

Sun et al use a combination of a virtual linear track, DREADD inactivation's and two-photon calcium imaging to show that RSC is critical for learning not to seek a reward in an unrewarded context. Interestingly, they found that single neurons in RSC encoding many dimensions of the task related variables. This suggests RSC (like other higher order cortical regions) is well positioned to integrate and translate information. In general, the study represents an extensive

undertaking, is well thought out, analyses are rigorous and thorough, and the findings are quite interesting. I have a few mostly minor concerns.

We thank the reviewer for their encouraging review and specifically for the suggestion to clarify the reversal learning paradigm in general and also to further explore and clarify the patterns of behavioural and neuronal activity within the ITI period, in reference to confounding reward responses, etc. We address these main concerns below and have further corrected all of the additional minor points.

3.1 I am struggling with the reversal learning aspect of this study. First, reversal learning wasn't really covered in the introduction, it just suddenly appeared when summarizing the study (in the introduction section). Second, if I am following, the findings for reversal learning generally mirror what is seen with initial learning. So is there really a deficit or differential encoding for reversal learning or do the encoding and dysfunction following inactivation actually reflect the same processes as with initial learning?

We have added an introduction to reversal learning including the relevance to the study aims (lines 46-49). We agree that we did not make the distinction between learning and reversal learning entirely clear and explicit, especially in relation to the inactivation of the RSC as well as the full implications of our results. We have updated the manuscript to reflect this improvement (lines 91-114, see also point 1.15, which discusses this issue and our changes in more detail in relation to the RSC inactivation experiments).

3.2 The period used to assess contextual encoding includes when other differential behaviors were present, suggesting this encoding might not actually be contextual. This is acknowledged in the manuscript, but as far as I can tell these two possibilities are never explicitly teased apart. Is it possible the contextual encoding actually represents behavioral or other differences between the two contexts (e.g., slowing vs not)? The post reward period is also examined, but again there is the presence or absence of a reward that is confounded with context. It is possible I am missing something, but if not, I am wondering if it might be better to temper some of the language about contextual encoding.

We further discuss the influence of differential behaviours (see also point 1.1 and 1.3) and generally clarify the parameters of the experiment that can be separated based on behavior versus contextual processing. Since we did not observe systemically significant differences between decoding accuracy for context, position or speed that was linked to the rewarded context when we look specifically along the virtual corridor, it is unlikely that the contextual encoding representation was changed according to the slowing down per se. Further, we have clarified by strengthening the discussion surrounding differences in both behaviour and neuronal activity between the corridor and ITI (post-reward) periods (e.g. see lines 130-197; see also point 2.2 and Discussion 428-442); in this way we also now refer to the context-value associations that are represented in the ITI period rather than the specific contextual decoding as this activity is clearly linked to the behaviourally-relevant context itself, but also not simple reward-onset event responses. We think this has significantly helped to clarify the underlying representation of this contextual encoding.

Minor:

3.3 There were a few references to "slightly but significantly" (p. 17) or a change that wasn't significant. I would recommend being careful with such descriptions. At the very least, if I am following, the claim at the end of the second paragraph on p. 9 is not supported by the preceding results (perhaps it just needs moved?). A difference that is not significantly different (day 12) is not different than a second non-significant result (day 17), but I think it is suggested that there is a difference between day 12 and day 17.

We have reviewed the manuscript carefully to remove statements of speculation based on non-significant results and to avoid discussions of tendencies. With specific reference to the description of day 12 and day 17, this section has been heavily amended and the direct comparison removed (line 327-331).

3.4 Supplementary Fig. 2. The title of the figure does not match the graph (says not significant but significant results are indicated on the figure) and the figure also does not seem to match references to it in the manuscript. Is this the wrong figure?

We agree that this Figure was not an effective way to present this speed data and has caused confusion. We have replaced this figure with Figures that more comprehensively show the temporal speed dynamics: in the average speed trajectory across the whole corridor, as well as with the trial-by-trial and within session speed in the anticipation zone across all experimental conditions (see Fig. 2c, Supplementary Fig. 2, 9-11; see also point 1.2)

3.5 Supplementary Fig 7 appears out of order (between supplementary figure 2 and 3).

This has been corrected.

3.6 Fig. 3a. I think part of the schematic is shifted.

This has been corrected.

3.7 p. 21 reference #32. I believe motion encoding in RSC was shown first by McNaughton et al., 1994, should probably include this reference.

The reference has been added (line 230).

3.8 The discussion describes dense projections from visual cortex to RSC and the potential role of these inputs. However, RSC is also strongly reciprocally connected to PC which contains visual information, single cells there encode with high dimensionality, and PC has been suggested to be critical for integration and transformations. Thus, it might be worth also discussing a potential role of PC inputs/outputs.

In general, we expanded our discussion to include further references to circuit dynamics and the potential influence of reciprocal and top-down projections to the RSC, specifically in relation to responses observed within the ITI period (see Discussion lines 359-367, 432-442).

Reviewers' comments:

Reviewer #1 (Remarks to the Author):

I appreciate the effort that went in the revision. However, it has alleviated my overall concerns. Multiple conceptual issues as well as technical issues related to data analysis, quantification, visualization, and writing remain unaddressed, and new issues were introduced along with the new material. As a result, my assessment remains largely unchanged. The results on the effects of visual reward associations on running behavior (Figs. 1 and 5) are of significant potential interest but the results on neuronal representations (Figs 2-3) are still weak, conceptually and methodologically. The results of Figure 4 are a welcome addition but given that they follow (and presumably rest on) the weak framework set forth by Figs. 2-3, these results are difficult to get excited about. About of the issues of writing, the description of the behavioral data are inaccurate, the results of neural recordings in Figs. 2-3 seem overinflated, and the conceptual framework underlying the neural data is not clearly expressed. Furthermore, there is frequent blurring between results and data interpretation that does not inspire confidence.

Main points

The revised manuscript (text and figures) still does not define clearly context and is not sufficiently clear and explicit about limitations of the paradigm in distinguishing the contribution of visual and reward contexts. In any given experiment, each reward context (reward, non-reward, neutral) is tied to one particular visual environment so the specific effects of reward are not independently assessed. Yet the manuscript and figure panel frequently refer to reward and visual context separately giving impression that they are addressed or manipulated independently. This issue affects the manuscript text, captions, as well as icons and label in all main figures and in Supp. Figs. 1, 2, 3 and 4 in that individual panels specify either visual or reward context failing to clarify how these variables are conflated.

The new analyses of behavioral data show that the behavioral metric chosen (mean speed in AZ) is only weakly related to the learning, if at all. The supplementary data show that a portion of the variance at the AZ is not context sensitive so the depiction of the AZ as a readout of learning in Figure 1 and Figure 5 is misleading.

The conceptual framework underlying the analysis of RSC representations (Figs 2-3) is not explained clearly. While the behavioral data in Figs. 1-2 is focused on learning, this aspect is not examined in Figs. 2-3. It is brought up in Fig. 4 but the link between activity and behavior in the previous figures is not make clear. Methodologically, the new population activity plots (e.g. Figs. 2f,g) have issues and are not quantified properly. The decoding analysis which is focused on the entire trial is not matched to the behavior which already settles at 1/6th of the trial (see below). The metric used (decoding accuracy) may have issues. It is certainly not properly motivated.

With respect to the writing, there are inaccuracies in the description of the behavioral and neural data. Furthermore, with regards to behavior, language is used inappropriately in results to describe what is going on in the animal's head. Such language is subject interpretation and should be replaced with language that objectively describes the animal's behavior.

Points 1.1 and 1.2 : behavioral data

The speed traces (Supp Fig 2,9) are a welcome addition as they illustrate the behavior and show that the animals can robustly discriminate between contexts not only in the anticipation zone but throughout the trial.

However, the traces clearly show that the 'anticipation zone' is not the best choice for the quantification of behavioral data and is in fact quite misguided. In Figure 1, the authors point to the sudden drop in running speed in the AZ as the signal of interest, as the correlate of learning.

However, it is the overall difference in running speed which starts early in the trial and increases progressively during the trial, and not the sudden reduction in speed at the AZ, that is context sensitive. Though this may not influence the conclusions (average running speed is reflected in the AZ), for the purpose of explaining the behavior the focus on the AZ will confuse and mislead the reader. This issue of interpretation is exacerbated by the decision not to show the average speed for different contexts in Figure 1. I don't feel it is a good practice to relegate critical data like these to supplementary figures.

In that regards, the description of changes in running behavior across days (lines 122-132) is also inaccurate and confusing. First, the reduction in running speed during the trial and the sudden drop in speed the AZ develop independently. While the former is clearly visible in the first day of learning, the latter is not (Supp Fig 2). Second, the learning is not reflected in a decrease in running speed for the rewarded context but a running speed increase for non-rewarded contexts. An accurate description of the data would clarify that, on first day of learning, the reward is associated with a reduction in running speed and that this reduction is indiscriminate w.r.t. context. In the following days, animals progressively learn to increase their running speed in the non-rewarded contexts, effectively racing through non-rewarded trials. Again, there is a progressive increase in the reduction of running speed in the AZ but that sudden reduction in running speed seems to be largely insensitive to context.

These comments have implications for lines 135-146 as well as for the results of the chemogenetic experiments.

Related to point 1.5 : activity in visual corridors

Line 162 to 194 and Fig. 2f,h: this text and these figures are misleading and need to be corrected.

The text contains multiple assertions of aiming to investigate differences in neural activity and neural decoding accuracy across "rewarded, non-rewarded or neutral contexts" (lines 164, 169, 174, and 176). The way they are expressed, these assertions are misleading because in the experiments each reward context (reward, non-reward, neutral) is tied to a particular visual environment. Because both visual environment and reward are varied, these two variables are conflated.

To support their assertions, the authors would have to show data for combinations of visual environment and reward context, which may not be experimentally feasible.

The fact that RSC neurons show distinct activity in visual environments is in itself neither interesting nor informative not to say trivial. This point can be made succinctly in the context of other more interesting findings but the large amount space currently devoted to it in the text and main figures is not warranted and detrimental to the paper's message.

As for the text, the visual language and labels in the figures with regards to visual and reward contexts lack consistency, are confusing, and are even misleading. In some figure panels, 'context' ('ctx') is used to point to distinct visual environments (fig. 2b) in others it used in reference to combination visual environment and reward (supp. Fig. 2). Furthermore, in many panels, reward, non-reward and neutral icons (fig. 2d,f,g) are to refer to data acquired in distinct visual environments, without explicitly stating so.

This visual language and usage of labels is difficult to comprehend and risks of biasing the reader towards specific interpretations rather than serving interpretation of objective facts.

Point 1.3 : activity during ITI

I appreciate the effort to study the activity during ITI to control for sensory and behavioral variables but these data are simply not interpretable. The main finding is a slightly elevated decoding accuracy following reward delivery, which in itself is not surprising as such as salient sensory and motor event will have a powerful effect on behavioral state and triggered sensory inputs that may have activated a

new set of cells. Three seconds is short. The behavior differs between context and the finding that recent visual experience can be decoded from neural activity in darkness does not imply that the cells encode context. It may simply indicate that the distinct visual environments used induce distinct sensory adaptational states.

Figures and data analysis

The activity plots in Fig. 2f-g, Fig. 4e,g and Supp. Fig. 5 and elsewhere are problematic. First these plots are not quantitative and subject to perceptual artefacts. Second, they pre-suppose that complex temporal activity patterns (Fig. 3a-f) can be meaningfully summarized as single numbers, a very unlikely scenario. Finally, sorted data are shown without cross-validation and are plotted with arbitrary, non-monotonic blue-white-yellow color maps. If the authors want to make a point about similarity of activities they should quantify them by, for example, computing the cross-correlation between activity vectors. The blue-white-yellow color maps are arbitrarily applied to data with different ranges. Such non-monotonic color maps create patterns where there are none.

The standard errors in speed traces (Supp Fig 2,9) seem to be computed across both animals and trials. This is not a good practice as makes the error bars smaller than they would be if computed across animals. The figure captions report the number of animals so this is what should be used to compute the standard errors. I have not looked carefully but this could also occur at other places in the manuscript.

The bar plots throughout the manuscript are not up to current standards for plots in high quality journals. Whenever possible the authors should plot the data points. When matched data exist across sampling points, the data points should be linked by lines.

Ref: <https://www.nature.com/articles/s41551-017-0079#citeas>

When data points are plotted (e.g. Fig. 5b-e; Supp. Fig. 12), they often are too small to be seen.

Supp. Fig 2C y axes should be on the same scale as in Fig. 2A. Same for Supp. Fig. 11.

The sorted activities in populations activity plots (Fig. 2f,g; Fig. 5 e,g; Supp Fig. 5, Supp. Fig. 6) need to be cross-validated (see below).

Details

I quote below text from the manuscript followed by my comments.

123 is no explicit measure of success rate. However, we found that as learning progressed, animals
124 naturally increased their speed in the non-rewarded context (i.e. context 2) and decreased their
125 speed selectively in the rewarded context (i.e. context 1) in preparation to consume the water
126 reward (n = 5 mice, Figs. 1h, 2b, and Supplementary Fig. 1). This reduction in speed was

As explained above, this description is not a faithful representation with the data shown in Supp. Fig. 2. The addition of the reward is associated with a unspecific reduction in average speed. During 'learning' average speed in non-rewarded context increases whereas average in rewarded context stays relatively constant. There is also a progressive reduction in running speed in the anticipation zone but that reduction does not appear to be specific to rewarded non-rewarded contexts.

136 learning, when the rewarded and non-rewarded contexts were switched but the neutral context
137 (context 3) remained neutral. Here, mice need to additionally devalue the previously rewarded

This description of reversal learning is misleading because the neutral context is not presented until the end of reversal learning.

139 context (context 2). Similar to the initial learning progress, we found that even on the second
140 day of exposure to the new task, mice already showed significantly slower running speed in

This is not accurate description of the data presented in Suppl. Figure 2. The average speed on day 6
is higher than on day 5. Speed is slowest upon change of reward context association then speed to
non-rewarded context increases.

144 Therefore, mice were very efficient at this reversal learning task; they readily de-valued the
145 previously rewarded context to match the neutral context, and flexibly formed a new positive

The use of a term like 'very efficient' require definition/quantification and the term 'de-value' is an
interpretation not a description of the behavior. Furthermore, it is unclear how the term 'positive
association' rewarded context could be justified. The behavioral data shows an effect for non-
rewarded context (increase in running speed) not for the rewarded context. On the first day upon
reversal of the rewarded context, animals show similar average speeds for rewarded and non-
rewarded contexts. Later, as observed in the 'learning phase', animals show increase average speeds
for the non-rewarded context. The statement about the neutral context also does not seem to be
justified. The neutral and non-rewarded context have in common that they don't lead to a reward.

150 related signals during learning 36 we wanted to investigate the underlying activity in the 151
dysgranular RSC during this learning and reversal learning VR paradigm to determine how 152
information related to these combined factors were represented within the RSC. We,

Given that the factors are tied to one another and not manipulated independently, the study of how
factors are represented does not seem to be an accurate description of the neural data analysis that
follows.

162 We first examined the neuronal activity along the virtual corridor and found that, in
163 relation to the neutral context, RSC cells displayed heterogeneous response patterns for both
164 the rewarded and non-rewarded contexts - during both the learning phase (day 4) and after 165
reversal learning (day 7; averaged responses across trials, Fig. 2f). Indeed, when we used a

There are multiple unaddressed issues with this section. First, it makes a really weak point that does
not really advance the paper and certainly does not justify the large amount of text and space that is
allocated to it. The previous section focuses on differences over learning. Here without warning the
differences of activity patterns from one training session to the next are not examined. This is a
missed opportunity since response patterns across visual environments cannot be easily compared
whereas examining the evolution of response patterns could have been instructive.

Second, the text and new panels Fig. 2F and 2G are flawed. The authors mean to say that RSC cells
show distinct activation patterns in distinct environments/context (as is much more clearly
demonstrated for example cells in Fig. 3A). This is entirely expected as RSC neurons selectively
respond to visual stimuli and the 3 VR visual environments tested strongly differ in visual structure.
This is first acknowledged in line 178. By not making the expectation clear from the start, there is
implicit suggestion of a finding, which is misleading. Furthermore, the point is not made quantitatively.
The similarity in responses is not quantified (Fig. 2f). Also, the average response over trial does not
capture difference in response timing which can occur without difference in average amplitudes (e.g.
Fig. 3A, cell 3's responses to contexts 2 and 3). Finally, Fig. 2F and 2G plot sorted data without cross-
validation so comparison between data in third column and data in the first or second column is
meaningless. Furthermore, the plots fail to represent how the differences in responses observed
between visual environments exceeds differences expected from the variance of responses across
trials. This statement is indirectly supported by the decoding analysis but this is not explained
properly.

168 see methods), we found that there were no systematic differences in decoding accuracy 169
between the rewarded, non-rewarded or neutral contexts (Fig. 2h). Additionally, we found no

Here and below, it is not clear from the writing why the focus on decoding accuracy as a metric of differences in response pattern between environments and, considering the experiment design, it is not clear why discrimination between contexts, irrespective of learning, is the question to be addressed. Decoding accuracy is likely at saturation, making it a weakly sensitive metric, that is influenced by factors that have little to do with the quality of neural representations such as population size and measurement noise of the somatic calcium signals. Finally, the decoding uses activity from the entire trial whereas the behavioral data (Supp. Fig. 2) show animals discriminate between environments at 1/6th of the trial, making the parallel to the behavior very weak.

195 Next, we examined neuronal activity within the ITI period. Although animals no
196 longer saw the context-specific visual stimulus during this time interval, it is possible that the
...

This section is problematic. The finding is that reward can be decoded from RSC activity seconds after it was delivered. Because visual environment and reward were not independently manipulated it is not possible to determine whether the conjunction of reward and visual context is encoded.

200 rewarded contexts, in relation to the neutral context. Here the activity of individual neurons 201
was similar for the non-rewarded and neutral contexts, and differed for the rewarded context 202 (Fig.
2g); this was reflective of the behavioral changes in running speed observed across

As for Fig. 2f, similarity of responses is not quantified, and sorted responses to 'neutral' context cannot be visually compared to responses to 'rewarded' and 'non-rewarded' contexts without the use of cross-validation.

205 Accordingly, we also found a higher decoding accuracy for the rewarded context (Fig. 2j) as 206
well as a significantly higher proportion of context-encoding cells (Fig. 2k), following both

Similar issue with the use of decoding accuracy in the VR corridor. Why choosing it and how should this metric be interpreted? What is its meaning?

Reviewer #2 (Remarks to the Author):

The authors are to be applauded for this revision - the MS is clearly much improved. Sun et al. have included a range of different new analyses (up and foremost longitudinally matched single cell data), flipped the flow of argument (now closing with the behavior data), and provide a much more detailed and balanced discussion of their results in the context of previous work, including very recent findings. This is a significant effort that clearly paid off since the main message of the paper - further evidence for a role of RSC in value coding - is now supported by both the functional and behavioral data which now appear much less disjunct than in the first submission. Of course: It remains unclear how exactly an increase in multidimensional neurons (now nicely supported with longitudinal data) should lead to context devaluation upon reversal learning - but this is now much more carefully discussed and also clearly beyond the scope of a single study. Also the finding of ITI-coding for a preceding reward, one of my criticisms of the first version, is now more carefully interpreted and factors like reward-related ongoing neuromodulation are acknowledged. The presented data is convincing that simple "bleed-through" of reward-locked activity is not leading to this difference.

My remaining points are all minor.

- The authors now include longitudinal data (which I know was a significant effort!). However, it would be nice to see a some quantification of similarity for the qualitative sorted matched-cell plots in Fig. 2fg and 4ge (and the respective supplements). Simple population vector correlation matrices across contexts over learning phases in different blocks would suffice. Also: Did I miss the quantification of

the (lack of) amplitude differences over time which is strongly referred to in the results (L210-212)?

- These sorted longitudinal data figures would also be more telling if the sorting would be done cross-validated by using half of the trials and using these indices for all conditions. This would reveal better how much of the dissimilarity is trial-to-trial fluctuation and how much is showing representational differences. This is, for instance, relevant for supplementary Figure 5: To me it rather looks as if the ITI activity is much more similar to the activity in the reward epoch than in the corridor itself. This is, of course, expected because only the corridor has visual stimulation. But to make the point that ITI activity and reward activity are very different (L216,217) one should also take trial to trial fluctuations into account (i.e.: how similar are subsequent reward epochs). Simple population vector correlation matrices across reward, corridor and ITI trials again could make this measure quantitative.
- It should be clearly stated in the legend to fig 1 and results that the treadmill is blocked in the reward zone already - and not only in the ITI.
- Figure 3 e,j - please use "single context", "two contexts" and "three contexts" - it took me a bit to see that not the standard context index (ctx 1, 2 3 etc.) is used here.
- In the results and discussion there are several remarks towards "efficient encoding" of task variables (i.e. L280). I am not certain if evoking Horace Barlow was in the intention of the authors - but given that no quantification of common 'efficient coding' parameters like sparsity etc. are given, the authors should probably refrain from using this term to not lead readers onto the wrong path.

Reviewer #3 (Remarks to the Author):

It was a little hard to track to the changes in the manuscript at times because the line numbers given in the rebuttal letter did not seem to match the manuscript at times, but perhaps the authors were not at fault for this and regardless I think I was able to track everything down. All of my original concerns were addressed and I have no further or new concerns.

REVIEWER COMMENTS AND AUTHOR REPLIES

Reviewer #1 (Remarks to the Author):

I appreciate the effort that went in the revision. However, it has alleviated my overall concerns. Multiple conceptual issues as well as technical issues related to data analysis, quantification, visualization, and writing remain unaddressed, and new issues were introduced along with the new material. As a result, my assessment remains largely unchanged. The results on the effects of visual reward associations on running behavior (Figs. 1 and 5) are of significant potential interest but the results on neuronal representations (Figs 2-3) are still weak, conceptually and methodologically. The results of Figure 4 are a welcome addition but given that they follow (and presumably rest on) the weak framework set forth by Figs. 2-3, these results are difficult to get excited about. About of the issues of writing, the description of the behavioral data are inaccurate, the results of neural recordings in Figs. 2-3 seem overinflated, and the conceptual framework underlying the neural data is not clearly expressed. Furthermore, there is frequent blurring between results and data interpretation that does not inspire confidence.

Main points

1. The revised manuscript (text and figures) still does not define clearly context and is not sufficiently clear and explicit about limitations of the paradigm in distinguishing the contribution of visual and reward contexts. In any given experiment, each reward context (reward, non-reward, neutral) is tied to one particular visual environment so the specific effects of reward are not independently assessed. Yet the manuscript and figure panel frequently refer to reward and visual context separately giving impression that they are addressed or manipulated independently. This issue affects the manuscript text, captions, as well as icons and label in all main figures and in Supp. Figs. 1, 2, 3 and 4 in that individual panels specify either visual or reward context failing to clarify how these variables are conflated.

We greatly appreciate very constructive and helpful comments of the reviewer. We have amended the text in this regard and worked to improve the Figures to clarify this as much as possible. We now make a clearer semantic definition of what we mean by the 'context' at the beginning of the results (lines 76-78), which we specifically define as the 'virtual environment'. This now reads:

“Animals were presented with three different contexts across a series of days, **which are defined by the parameters of the virtual environment - including the spatial properties of the linear corridor as well as the visual pattern present along the length of the corridor.**”

We believe this further improves the semantics associated with this distinction and will assist the reader.

In terms of separating out the contribution of visual and reward parameters, we had already specified that the context is strongly represented by the visual parameters of the experiment (results lines 90-95) and discussed how these elements are not separable within each experimental day (discussion lines 465-470), and this is why this is not a separate category

for determining multidimensional responses. For the effects of reward: we speak of context-value associations to represent the fact that the reward is linked to different contexts. For example, while the reward is linked to a specific context and virtual environment within each experimental day, the context that is linked with the reward is alternated across days so that reward responses are present in context 1 on day 4 and context 2 on day 7. This has allowed us to examine uniquely reward-related responses across days and separate these parameters from purely visual responses to the particular virtual environment (i.e., lines or checkerboard patterns). We have explicitly added this to the results (lines 85-95), which now read:

“The aim was to establish an experimental paradigm that would passively, but consistently, pair a particular context with a reward, thereby assigning value to a specific context through associative learning - as may occur when animals are navigating through an environment and discover a food source at a particular location. Therefore, rewards were given to the animals consistently at a default location in a particular context without the need for the animal to perform an extraneous behavior to receive the reward. Within each experimental day this context is then inseparable from the parameters of the virtual environment (e.g., visual pattern on the corridor walls) and its behaviourally-relevant value, however, on subsequent days a different context is paired with reward (i.e., reversal learning) to assess changes in context-value associations and to dissociate the effects of the reward-association from responses specific to the visual pattern on the corridor walls (Fig. 1i).”

For instance, in Fig. 2f, we can see the population responses change across days according to the reward (or context-value association: rewarded context, non-rewarded context, neutral context). These three categories of context-value associations are present on each experimental day when imaging was performed - but, importantly, the visual wall pattern that underlie the rewarded and non-rewarded context change across learning. Therefore, we can say that changes in responses to context 1 from day 4 to day 7 are linked specifically to the presence (day 4) or absence (day 7) of the reward - since all other parameters of the virtual environment (or context) have been kept consistent except for this one context-value association change. Hence, the reward and the context are manipulated independently across days. These are the important transfers of context-value associations that we are examining. We have amended the text to better define this and throughout when we refer to context-decoding - to more clearly indicate that what we are assessing by looking at changes across days is the decoding of context-value associations. For instance, one added passage now reads (lines 157-164):

“To determine how the context (parameters of the virtual environment) and the context-value associations (specific context linked to reward, no-reward, or neutral value associations) were represented in this heterogeneous population activity in the RSC, we examined the ability to decode the individual contexts (baseline day 1) and the change in context-value associations formed with learning (day 4) and after reversal learning (day 7). We use decoding accuracy as a metric to quantify the extent to which external and task-related variables are represented by neuronal population activity within the RSC.”

We have also added an additional analysis that uses longitudinal data to follow the responses of individual neurons across days to group reward-associated data across days, which spans different visual contexts (new Supplementary Fig. 5). This allowed us to show that these context-value associations can be rather accurately predicted across days during both the

length of the corridor in which we also observe differences in our behavioural metric (40-180 cm) as well as during the ITI period. This is now included in the revised results (lines 184-206) and in the discussion (lines 428-432).

We hope this is now clear and that our efforts to convey this effectively in the figures and text have helped with this distinction.

2. The new analyses of behavioral data show that the behavioral metric chosen (mean speed in AZ) is only weakly related to the learning, if at all. The supplementary data show that a portion of the variance at the AZ is not context sensitive so the depiction of the AZ as a readout of learning in Figure 1 and Figure 5 is misleading.

We disagree that the supplementary figures show that the mean speed in the AZ is weakly related to learning. The supplementary figures show a trajectory of speed that begins to deviate along the virtual corridor but does not reach the maximum deviation point until the end of the virtual corridor (i.e., within the anticipation zone). It is true that there are ubiquitous decreases in the relative speed across all context indiscriminately due to the upcoming deployment of the 'breaks' in the behavioural paradigm for all contexts (used in order to reduce locomotion confounds in the reward and ITI periods, which has already been discussed, lines 98-102), but this contributes across contexts and therefore this region is still a valid representation of the max difference in speed trajectories as summed along the length of the corridor. This is evident from Supplementary Fig. 15 (e.g., Control_Day 7) and the trajectories have now also been highlighted in main Fig. 1j.

To further address this issue for the reviewer, we used the linear discriminant analysis to predict if an animal will be rewarded or not at the end of a trial from the mean speed in the anticipation zone for all trials across both learning and reversal learning combined. The accuracy of reward prediction was $84.8 \pm 1.1\%$ ($n=5$), which was significantly better than when the mean speed in the whole virtual corridor was used ($71.9 \pm 4.4\%$, $n = 5$, $p = 0.022$). Moreover, when we artificially selected the optimal combination of speed values in all spatial bins to maximize the accuracy of reward prediction, it was $87.7 \pm 1.3\%$ ($n = 5$), i.e., not much better ($p = 0.104$) when the mean speed in the anticipation area was employed, thus justifying the use of the latter measure.

We have added this analysis into the Results (lines 107-111):

“Using linear discriminant analysis, we predicted whether an animal was rewarded or not at the end of a trial from the mean speed in the anticipation zone for all trials in contexts 1 and 2 across both learning and reversal learning. The accuracy of prediction was $84.8 \pm 1.1\%$ ($n=5$), which was significantly better than when the mean speed in the whole virtual corridor was used ($71.9 \pm 4.4\%$, $n = 5$, $p = 0.022$).”

If deemed necessary by the Editor, we can also include this as an additional supplementary figure.

3. The conceptual framework underlying the analysis of RSC representations (Figs 2-3) is not explained clearly. While the behavioral data in Figs. 1-2 is focused on learning, this aspect is not examined in Figs. 2-3. It is brought up in Fig. 4 but the link between activity and behavior in the previous figures is not make clear. Methodologically, the new population activity plots (e.g. Figs. 2f,g) have issues and are not quantified properly. The decoding analysis which is focused on the entire trial is not matched to the behavior which already settles at 1/6th of the trial (see below). The metric used (decoding accuracy) may have issues. It is certainly not properly motivated.

The aspects of learning are continually examined in Figures 2-3 with the use of analysis across days (e.g., baseline, learning, reversal learning) as well as with the analysis of speed-related cells in Figure 3, which is the exact behavioural metric that we are measuring in Fig 1-2. Then, as the reviewer mentions, this is again linked back in Figure 4. We have been previously criticized regarding the length of the results, and without adding substantial text to reiterate these relationships across figures, we are unclear as to what further changes we can make here. We hope that the many overall changes we have made throughout the manuscript will now make this clearer to the reviewer in general. With regard to the population activity plots, we have done new analysis to significantly strengthen the presentation of this effect by adding cross-validation (Supplementary Figs. 4) as well as quantification using cross-correlation (Fig. 2e, 2f and new Supplementary Figs. 6-7). The decoding accuracy metric is a very common metric used in the field. Since the aim of the paper is to investigate the dynamics of multidimensional responses and integration of task dimensions, this analysis of population decoding is justified. We have additionally made changes to the text to further explain the motivation for using this metric (lines 157-164):

“To determine how the context (parameters of the virtual environment) and the context-value associations (specific context linked to reward, no-reward, or neutral value associations) were represented in this heterogeneous population activity in the RSC, we examined the ability to decode the individual contexts (baseline day 1) and the change in context-value associations formed with learning (day 4) and after reversal learning (day 7). We use decoding accuracy as a metric to quantify the extent to which external and task-related variables are represented by neuronal population activity within the RSC.”

Our additional analyses performed to address this issue are discussed below (21).

4. With respect to the writing, there are inaccuracies in the description of the behavioral and neural data. Furthermore, with regards to behavior, language is used inappropriately in results to describe what is going on in the animal's head. Such language is subject interpretation and should be replaced with language that objectively describes the animal's behavior.

We have made substantial efforts to use precise language and clearly separate results from interpretations throughout the revision and have made changes in various places (e.g., lines 105-111, 113-116, and 132-133). However, we must also note that for scientific communication to be accessible to a wide audience (from PhD students, to scientific journalists and even the general public), a less technical style of writing is often preferred. In the least this is a matter of personal writing style and while we have been careful to remove

any perceived inaccuracies or misleading language, we do not agree that our language was overly inappropriate or misleading. Many of the confusions seem to stem from misunderstandings about what comparisons the text was specifically referring to, so we have tried to amend this throughout without being overly repetitive in our description of the results (which was also a criticism from the previous revision. Regardless, we have made significant efforts to use precise language throughout (also explained in the responses to the 'Details' section below, points 16-26).

5. Points 1.1 and 1.2 : behavioral data: The speed traces (Supp Fig 2,9) are a welcome addition as they illustrate the behavior and show that the animals can robustly discriminate between contexts not only in the anticipation zone but throughout the trial. However, the traces clearly show that the 'anticipation zone' is not the best choice for the quantification of behavioral data and is in fact quite misguided. In Figure 1, the authors point to the sudden drop in running speed in the AZ as the signal of interest, as the correlate of learning. However, it is the overall difference in running speed which starts early in the trial and increases progressively during the trial, and not the sudden reduction in speed at the AZ, that is context sensitive. Though this may not influence the conclusions (average running speed is reflected in the AZ), for the purpose of explaining the behavior the focus on the AZ will confuse and mislead the reader. This issue of interpretation is exacerbated by the decision not to show the average speed for different contexts in Figure 1. I don't feel it is a good practice to relegate critical data like these to supplementary figures.

We appreciate the reviewer's note that "animals can robustly discriminate between contexts not only in the AZ but throughout the trial", apparently after the first 40 cm. Although this was mentioned already in the previous version, we now state this even more clearly in lines 102-120:

"This change in speed across contexts was maximally represented in the 10 cm before the water reward was given, in the reward 'anticipation zone', and was consistently reproduced across experimental groups (Fig. 2a and Supplementary Fig. 1). Additionally, we found consistent speed trajectories across trials, with mice starting to show alterations in running speed after the first ~40cm along the corridor (Figure 1j). Using linear discriminant analysis, we predicted whether an animal was rewarded or not at the end of a trial from the mean speed in the anticipation zone for all trials in contexts 1 and 2 across both learning and reversal learning. The accuracy of prediction was $84.8 \pm 1.1\%$ ($n=5$), which was significantly better than when the mean speed in the whole virtual corridor was used ($71.9 \pm 4.4\%$, $n = 5$, $p = 0.022$). We also found a clear shift in the distribution of speeds within the anticipation zone on a trial-by-trial basis across various learning stages (Fig. 2b) as well as within learning sessions (Supplementary Fig. 2). Although mice slowed down in the anticipation zone across all contexts following training (due to the enforced stationary period at the end of each trial), this anticipation zone region represents the culmination of the difference in speeds across the length of the corridor (Fig. 1j, Supplementary Fig. 2). Therefore, in this type of passive training paradigm that has no explicit measure of success rate per se, we found that this change in speed along the virtual corridor, culminating in a maximal difference between rewarded versus non-rewarded contexts at the anticipation zone, was a reliable behavioral metric of learning and context discrimination."

Moreover, as explained in our reply to point 2, our analysis shows that as a metric predictive of reward, the mean speed in the anticipation zone is slightly better than the total mean speed, hence, it is a valid metric for learning in this regard. Indeed, as already mentioned by the reviewer, this distinction does not change the results of the study and either measure could be used in future experiments. We believe this point is now clear in the revised manuscript.

Additionally, we have now moved a condensed version of speed trajectories for the behaviour into the main Figure 1 (Figure 1j, showing baseline, learning and reversal learning days).

6. In that regards, the description of changes in running behavior across days (lines 122-132) is also inaccurate and confusing. First, the reduction in running speed during the trial and the sudden drop in speed the AZ develop independently. While the former is clearly visible in the first day of learning, the latter is not (Supp Fig 2). Second, the learning is not reflected in a decrease in running speed for the rewarded context but a running speed increase for non-rewarded contexts. An accurate description of the data would clarify that, on first day of learning, the reward is associated with a reduction in running speed and that this reduction is indiscriminate w.r.t. context. In the following days, animals progressively learn to increase their running speed in the non-rewarded contexts, effectively racing through non-rewarded trials. Again, there is a progressive increase in the reduction of running speed in the AZ but that sudden reduction in running speed seems to be largely insensitive to context. These comments have implications for lines 135-146 as well as for the results of the chemogenetic experiments.

We have already noted in the manuscript that: *on the first day of learning, the reward is associated with a reduction in running speed and that this reduction is indiscriminate w.r.t. Context.* This is described in more detail in relation to the results from the DREADD experiments. We have added additional text early in the results to indicate this here as well:

“However, we found that as learning progressed, **after an initial decrease in speed across all contexts**, animals naturally increased their speed in the non-rewarded context (i.e., context 2) and decreased their speed selectively in the rewarded context (i.e., context 1), **presumably** in preparation to consume the water reward (n = 5 mice, Fig. 1h, 1j, 2a, and Supplementary Fig. 1). This **change in speed across contexts was maximally represented** in the 10 cm before the water reward was given, in the reward ‘anticipation zone’, and was consistently reproduced across experimental groups (Fig. 2a and Supplementary Fig. 1). Additionally, we found consistent speed trajectories across trials, **with mice starting to show alterations in running speed after the first ~40cm along the corridor (Figure 1j).** Using linear discriminant analysis, **we predicted whether an animal was rewarded or not at the end of a trial from the mean speed in the anticipation zone for all trials in contexts 1 and 2 across both learning and reversal learning. The accuracy of prediction was $84.8 \pm 1.1\%$ (n=5), which was significantly better than when the mean speed in the whole virtual corridor was used ($71.9 \pm 4.4\%$, n = 5, p = 0.022).** We also found a clear shift in the distribution of speeds within the anticipation zone on a trial-by-trial basis across various learning stages (Fig. 2b) as well as within learning sessions (Supplementary Fig. 2). **Although mice slowed down in the anticipation zone across all contexts following training (due to the enforced stationary period at the end of each trial), this anticipation zone region represents the culmination of the difference in speeds across the length of the corridor (Fig. 1j, Supplementary Fig. 2).** Therefore, in this type of passive training

paradigm that has no explicit measure of success rate per se, we found that this change in speed **along the virtual corridor, culminating in a maximal difference between rewarded versus non-rewarded contexts at the anticipation zone**, was a reliable behavioral metric of learning and context discrimination.” (lines 98-120).

We have also changed the wording in relation to the old lines 135-146 as follows: “Similar to the initial learning progress, we found that even on the second day of exposure to the new task, mice already showed significantly slower running speed for the newly rewarded context **(i.e., context 2) compared to the non-rewarded context** (i.e., context 1; Fig. 2a; see Supplementary Fig. 2).”

To the second point, actually learning is reflected in both the increase in speed in the non-rewarded and neutral contexts and a decrease in speed in the rewarded context (as mentioned already in the text indicated above). Perhaps this is now also better appreciated in the additional presentation of the data in Figure 1j, where graphs can be compared between baseline, learning and reversal learning across the same scale horizontally.

7. Related to point 1.5 : activity in visual corridors: Line 162 to 194 and Fig. 2f,h: this text and these figures are misleading and need to be corrected. The text contains multiple assertions of aiming to investigate differences in neural activity and neural decoding accuracy across “rewarded, non-rewarded or neutral contexts” (lines 164, 169, 174, and 176). The way they are expressed, these assertions are misleading because in the experiments each reward context (reward, non-reward, neutral) is tied to a particular visual environment. Because both visual environment and reward are varied, these two variables are conflated. To support their assertions, the authors would have to show data for combinations of visual environment and reward context, which may not be experimentally feasible.

Perhaps the confusion here partially comes from the use of specific terms. We have edited the text to make this clearer. In our experimental design, within a specific learning phase, one reward is tied to a particular context (i.e., virtual environment), however, in the reversal learning phase, the specific context that is linked to the reward is switched, even though the same virtual environments are presented. Therefore, by comparing results across days, the virtual environments (contexts) remain the same while the context-value association switches from context 1 to context 2. Additionally, we have the baseline responses of the RSC population to each virtual environment (context) from day 1. When we see changes in the accuracy of the decoding across days that are specific to the context that is rewarded (context 1 for day 4 and context 2 for day 7) in comparison to baseline values for these virtual environments on day 1, this is then an indication that positive context-value associations are more easily decoded by the RSC population, as this is the only factor that changes (i.e., the association with a reward for that particular virtual environment) across these days. We have edited this portion of the results to make this clearer (lines 152-175 see also point 1 and new Supplementary Figures 5).

8. The fact that RSC neurons show distinct activity in visual environments is in itself neither interesting nor informative not to say trivial. This point can be made succinctly in the context of other more interesting findings but the large amount space currently devoted to it in the text and main figures is not warranted and detrimental to the paper’s message.

The comparisons made across days and learning phases in relation to the responses in the different virtual environments linked to different context-value associations are not trivial. Especially considering our different results in the corridor versus the ITI region of the paradigm and the data pointing to the “ceiling” effect in accuracy of decoding of environments with multiple repeated visual features. During revision we have consistently been instructed to expand on this analysis and so we have in a number of ways; regardless, our results will be of interest to many readers. For the section that is devoted to the analysis of specific visual responses, which indeed are already well established in the RSC, we have compressed the text in the results as much as possible for this section (see lines 177-184) and replaced them with a more in-depth discussion about the context-value associations across learning phases (see point 1). We do not believe however that a single supplementary figure is over represented in this regard as these visual properties are still interesting in relation to our other results and may give interested readers some insight for responses observed in related studies. Lastly, and we are sure that the reviewer would agree, it is important that the evidence for these responses in our dataset are established clearly in this manuscript before our further analysis of multidimensional responses based on these individual parameters.

9. As for the text, the visual language and labels in the figures with regards to visual and reward contexts lack consistency, are confusing, and are even misleading. In some figure panels, ‘context’ (‘ctx’) is used to point to distinct visual environments (fig. 2b) in others it used in reference to combination visual environment and reward (supp. Fig. 2). Furthermore, in many panels, reward, non-reward and neutral icons (fig. 2d, f, g) are to refer to data acquired in distinct visual environments, without explicitly stating so. This visual language and usage of labels is difficult to comprehend and risks of biasing the reader towards specific interpretations rather than serving interpretation of objective facts.

We have taken additional care to ensure consistency between figures and the visual symbols used in this regard. Along with the additional definitions of context and text edits separating context (virtual environment) and context-value associations (context-reward pairings), we feel as though this is now clearer. We however do not see the potential for bias with regard to the reward/non-reward symbols. We always have put all information - for instance in Figs 2c, e, f, these were already listed as Ctx 1/2/3 at the bottom of these panels - we have tried to make this more salient in the revised figure by adding the visual pattern illustration as well, but otherwise need to keep this consistent across figures, as the reviewer suggests. The reward/no-reward/neutral symbols are used in combination with the context 1/2/3 labels and show no bias - they are simply the conditions which the data has come from for each particular panel. Without these indications (due to the learning and reversal learning conditions) it would be difficult for readers to keep track of where the reward was given for the particular day/environment combination. We disagree that this leads to significant bias in interpretation, regardless, we have tried to simplify the visual representation of our multi-condition experimental paradigm, which can become complicated due to our many experimental manipulations. We hope this has improved readability and eased comprehension throughout.

10. Point 1. 3 : activity during ITI : I appreciate the effort to study the activity during ITI to control for sensory and behavioral variables but these data are simply not interpretable. The main finding is a slightly elevated decoding accuracy following reward delivery, which in itself is not surprising as such as salient sensory and motor event will have a powerful effect on

behavioral state and triggered sensory inputs that may have activated a new set of cells. Three seconds is short. The behavior differs between context and the finding that recent visual experience can be decoded from neural activity in darkness does not imply that the cells encode context. It may simply indicate that the distinct the visual environments used induce distinct sensory adaptational states.

The fact that this increase in decoding accuracy changes with the rewarded context (virtual environment) across days (i.e., context 1 on day 4 and context 2 on day 7) are evidence that these responses are separable from the specific virtual environment but instead representative of context-value associations. The fact that the responses in the ITI period are not a simple reflection of the reward itself further suggests that populations in the RSC integrate more complex task-information in this ITI period following each trial, and the evidence from Figure 4 that the neurons which encode more multidimensional parameters contribute to this effects more strongly suggests that this more complex encoding is underlying the ability for the network to decode these meaningful context-value associations and, further, the fact that this proportion of multidimensional neurons increase with reversal learning, which requires cognitive flexibility, indicated these populations are important in this regard. Given all this evidence, it is unlikely that these ITI responses represent a simple reflection of behavioural state - which is also generally represented by an increase in gain in networks and not a specific change in selectivity or parameter encoding.

Figures and data analysis

11. The activity plots in Fig. 2f-g, Fig. 4e,g and Supp. Fig. 5 and elsewhere are problematic. First these plots are not quantitative and subject to perceptual artefacts. Second, they presuppose that complex temporal activity patterns (Fig. 3a-f) can be meaningfully summarized as single numbers, a very unlikely scenario. Finally, sorted data are shown without cross-validation and are plotted with arbitrary, non-monotonic blue-white-yellow color maps. If the authors want to make a point about similarity of activities they should quantify them by, for example, computing the cross-correlation between activity vectors. The blue-white-yellow color maps are arbitrarily applied to data with different ranges. Such non-monotonic color maps create patterns where there are none.

We have added further quantification of the cross-correlation matrix between activity vectors in different contexts and the mean correlation (Supplementary Figs. 6, 7, 9, 11). We have also changed the color map to monotonic "Viridis" with monotonically increasing luminance and a pleasant smooth arc through blue, green, and yellow hues which is widely used in scientific journals. We have also cross validated the sorted data with odd and even trials following the same sorting orders, which resulted in the same pattern of correlations. While these plots are meant to specifically represent the mean activity across cells, our further decoding analysis and many other figure representations throughout (e.g., cell example heatmaps etc) represent different temporal aspects of the activity. It is not feasible to represent all dimensions of the data in coherent plots. We are here representing a summary of the mean activity for all individual cells, across all different contexts, across all different days - this is an important main point that is then expanded on in our further analysis throughout the paper. As the reviewer has noted earlier, the ITI period is not that long (3 seconds), so additionally this is not likely to be much of a factor here. For the corridor (Fig 2e) this is indeed a mean across a longer time period, however in the decoding analysis this is separated into 18 bins along the

track and therefore represents the temporal aspects, and yet we find the same pattern of results with no systematic difference between decoding accuracy for context 1 and 2 across days.

12. The standard errors in speed traces (Supp Fig 2,9) seem to be computed across both animals and trials. This is not a good practice as makes the error bars smaller than they would be if computed across animals. The figure captions report the number of animals so this is what should be used to compute the standard errors. I have not looked carefully but this could also be occur at other places in the manuscript.

This is the only place this form of presentation was used because the original comment in the previous review round was in relation to the trial-by-trial variability. However, we have changed this figure and now we report the standard errors across animals for all figures throughout the manuscript.

13. The bar plots throughout the manuscript are not up to current standards for plots in high quality journals. Whenever possible the authors should plot the data points. When matched data exist across sampling points, the data points should be linked by lines. Ref: <https://www.nature.com/articles/s41551-017-0079#citeas>. When data points are plotted (e.g. Fig. 5b-e; Supp. Fig. 12), they often are too small to be seen.

In our last revision we were also asked to simplify our visualizations - we found that the already complex conditions with the addition of dots and connecting lines made for very busy visualizations. Where possible we have added these, but we insist on keeping the dots small and the lines very light grey, otherwise this additional information completely masks the presented effects and has a detrimental outcome on the comprehension of our figures. For those who are interested in the individual data points, these will be visible, especially in the online version where figures can be zoomed in etc.

14. Supp. Fig 2C y axes should be on the same scale as in Fig. 2A. Same for Supp. Fig. 11.

This has been changed as requested.

15. The sorted activities in populations activity plots (Fig. 2f,g; Fig. 5 e,g; Supp Fig. 5, Supp. Fig. 6) need to be cross-validated (see below).

This has been added as requested (new Supplementary Figures 4, 9, 10, and 12; see also points 3 and 11)

Reviewer's comments: Details

I quote below text from the manuscript followed by my comments.

16. 123-126 *...is no explicit measure of success rate. However, we found that as learning progressed, animals naturally increased their speed in the non-rewarded context (i.e. context 2) and decreased their speed selectively in the rewarded context (i.e. context 1) in preparation to consume the water reward (n = 5 mice, Figs. 1h, 2b, and Supplementary Fig. 1). This reduction in speed was...*

As explained above, this description is not a faithful representation with the data shown in Supplementary Fig. 2. The addition of the reward is associated with an unspecific reduction in average speed. During 'learning' average speed in non-rewarded context increases whereas average in rewarded context stays relatively constant. There is also a progressive reduction in running speed in the anticipation zone but that reduction does not appear to be specific to rewarded non-rewarded contexts.

We have reworded the passage as follows (see also point 5 and 6):

“However, we found that as learning progressed, **after an initial decrease in speed across all contexts**, animals naturally increased their speed in the non-rewarded context (i.e., context 2) and decreased their speed selectively in the rewarded context (i.e., context 1), **presumably** in preparation to consume the water reward (n = 5 mice, Fig. 1h, **1j**, **2a**, and Supplementary Fig. 1).”

This 'unspecific' reduction in speed in all contexts at the end of the corridor comes from learning that for all contexts the breaks will be applied in the reward region. We have clarified this further in multiple places in the text (e.g., lines 113-116):

“**Although mice slowed down in the anticipation zone across all contexts following training (due to the enforced stationary period at the end of each trial), this anticipation zone region represents the culmination of the difference in speeds across the length of the corridor (Fig. 1j, Supplementary Fig. 2).**”

And lines 143-147:

“Next, the period, where **external breaks were applied and therefore** the treadmill was blocked so all animals were stationary, the reward spout was extended for all trials, then a water reward was dispensed following only the appropriate context and animals consumed the reward, and finally the reward spout was retracted for all trials (termed the reward period).”

17. 136-137 *...learning, when the rewarded and non-rewarded contexts were switched but the neutral context (context 3) remained neutral. Here, mice need to additionally devalue the previously rewarded...*

This description of reversal learning is misleading because the neutral context is not presented until the end of reversal learning.

We have reworded the passage for clarity (lines 121-124):

“Using this behavioral metric, we then examined how mice perform during reversal learning, when the **reward-association was switched from context 1 to context 2** and the neutral context (context 3) **was not presented during training days and, hence,** remained neutral **on the test day (day 7).**”

18. 139-140 ...context (context 2). Similar to the initial learning progress, we found that even on the second day of exposure to the new task, mice already showed significantly slower running speed in...

This is not accurate description of the data presented in Suppl. Figure 2. The average speed on day 6 is higher than on day 5. Speed is slowest upon change of reward context association then speed to non-rewarded context increases.

We have reworded the passage as follows (lines 126-129):

“Similar to the initial learning progress, we found that even on the second day of exposure to the new task, mice already showed significantly slower running speed for the newly rewarded context (i.e., context 2) compared to the non-rewarded context (i.e., context 1; Fig. 2a; see Supplementary Fig. 2).”

19. 144-145 Therefore, mice were very efficient at this reversal learning task; they readily de-valued the previously rewarded context to match the neutral context, and flexibly formed a new positive...

The use of a term like ‘very efficient’ require definition/quantification and the term ‘de-value’ is an interpretation not a description of the behavior. Furthermore, it is unclear how the term ‘positive association’ rewarded context could be justified. The behavioral data shows an effect for non-rewarded context (increase in running speed) not for the rewarded context. On the first day upon reversal of the rewarded context, animals show similar average speeds for rewarded and non-rewarded contexts. Later, as observed in the ‘learning phase’, animals show increase average speeds for the non-rewarded context. The statement about the neutral context also does not seem to be justified. The neutral and non-rewarded context have in common that they don’t lead to a reward.

During reversal learning, mice do show a decrease in running speed for the newly rewarded context (context 2) compared to both baseline days as well as the end of learning (day 4), as well as an increase in running speed for context 1 (non-rewarded) and the neutral context stays high. Both effects are clear across days (please see especially Ctx 2 speed day 4 vs Ctx 2 speed day 7, Figure 2a). The passage has been reworded (lines 129-135):

“By the testing day (day 7), mice again showed significantly slower running speeds in the rewarded context (context 2) compared to baseline days and the end of the initial learning phase, and the same faster running speed in both the non-rewarded and neutral contexts (Fig. 2a, c). Therefore, mice changed their running behaviour during this reversal learning task; indicating that they de-valued the previously rewarded context to match the neutral context, and flexibly formed a new reward association with the previously non-rewarded context.”

20. 150-152 ...related signals during learning 36 we wanted to investigate the underlying activity in the dysgranular RSC during this learning and reversal learning VR paradigm to determine how information related to these combined factors were represented within the RSC.

Given that the factors are tied to one another and not manipulated independently, the study of how factors are represented does not seem to be an accurate description of the neural data analysis that follows.

This has been addressed in point 1 above.

21. 162-165 *We first examined the neuronal activity along the virtual corridor and found that, in relation to the neutral context, RSC cells displayed heterogeneous response patterns for both the rewarded and non-rewarded contexts - during both the learning phase (day 4) and after reversal learning (day 7; averaged responses across trials, Fig. 2f). Indeed, when we used a...*

There are multiple unaddressed issues with this section. First, it makes a really weak point that does not really advance the paper and certainly does not justify the large amount of text and space that is allocated to it. The previous section focuses on differences over learning. Here without warning the differences of activity patterns from one training session to the next are not examined. This is a missed opportunity since response patterns across visual environments cannot be easily compared whereas examining the evolution of response patterns could have been instructive.

The intention here for the description of the results observed along the corridor, as with the following description of the responses within the ITI, is to compare across days, as stated in the first sentence. Perhaps this was not as evident here since there were no significant differences across days, but the comparisons are still made. We have revised the text to make this clearer (see lines 152-175). We have also added a new analysis to more directly separate out the visual responses for individual neurons across days in our subset of longitudinal data that is later used for the remapping in Figure 4. This includes an additional Supplementary Fig. 5 (which we could include in a main Figure if deemed necessary) and a description in the results as follows:

“In order to more directly separate out the responses of individual neurons to the specific visual pattern of the virtual environment versus the context-value associations, we followed the neuronal responses for a subset of neurons longitudinally across days 1, 4, and 7. Using principal component analysis we found that representations of neuronal activity during both learning phase and context-value associations were separable (e.g., Supplementary Fig. 5a). Therefore, we further performed linear discriminant analysis to test the decoding accuracy for the context value from neuronal responses, i.e., % of correct prediction of the rewarded context (context 1 on day 4 and context 2 on day 7) in relation to all other non-rewarded contexts across all learning phases. We focused on responses along 40-180 cm of the virtual corridor, which corresponds to the length of the virtual environment where we found the divergence of speed trajectories in the running behaviour across contexts with training (see Fig. 1j, Supplementary Fig. 2). We found that the decoding accuracy of the rewarded context-value association was significantly higher for the observed compared to shuffled data ($83.7 \pm 3.5\%$ vs $55.5 \pm 1.2\%$, $n = 4$; $p = 0.006$; see Supplementary Fig. 5c). Thus, a large proportion of the RSC neuronal population, even in naïve animals, showed context-specific patterns of activity along the virtual corridor and allowed for accurate discrimination of all virtual environments. Moreover, the response properties of individual neurons across days could be predictive of context-value

associations across learning phases during periods when behaviour also diverged across contexts.”

One reason there is so much space dedicated to this section in general is that we are explaining our methods and analysis, which is then also applied to the ITI region in the same way. We don't feel as though it is fair to ignore the reporting of the results along the corridor just because we don't see significant changes across days from this analysis - this is an indication of the stability of this context encoding in relation to the visual responses across changes in behaviour, which is also interesting in relation to the more dynamic responses we see on the population level in the ITI period. We have shortened the description of the results based on the visual parameters as suggested (see point 8).

22. Second, the text and new panels Fig. 2F and 2G are flawed. The authors mean to say that RSC cells show distinct activation patterns in distinct environments/contexts (as is much more clearly demonstrated for example cells in Fig. 3A). This is entirely expected as RSC neurons selectively respond to visual stimuli and the 3 VR visual environments tested strongly differ in visual structure. This is first acknowledge in line 178. By not making the expectation clear from the start, there is implicit suggestion of a finding, which is misleading. Furthermore, the point is not made quantitatively. The similarity in responses is not quantified (Fig. 2f). Also, the average response over trial does not capture difference in response timing which can occur without difference in average amplitudes (e.g. Fig. 3A, cell 3's responses to contexts 2 and 3). Finally, Fig. 2F and 2G plot sorted data without cross-validation so comparison between data third column and data in the first or second column is meaningless. Furthermore, the plots fail to represent how the differences in responses observed between visual environments exceeds differences expected from the variance of responses across trials. This statement is indirectly supported by the decoding analysis but this is not explained properly.

We are not sure where the confusion comes from here, but the intention of Fig 2e, f is not to show distinct activation patterns for specific contexts, but that the activation pattern to specific contexts changes according to the context-value association across learning phases. Even though the context (or virtual environment) stays the same, i.e., the visual pattern does not change etc, when associated with a reward, the neurons in RSC show a different activation pattern in relation to a neutral context. Here the reward-association effects across learning phases are separated from the effects of visual stimulation for each specific context (see also point 1 above). We have further quantified this pattern of responses using cross-correlation and additional cross-validation methods (see also points 3 and 11).

23. 168-169 ...see methods), *we found that there were no systematic differences in decoding accuracy between the rewarded, non-rewarded or neutral contexts (Fig. 2h). Additionally, we found no...*

Here and below, it is not clear from the writing why the focus on decoding accuracy as a metric of differences in response pattern between environments and, considering the experiment design, it is not clear why discrimination between contexts, irrespective of learning, is the question to be addressed. Decoding accuracy is likely at saturation, making it a weakly sensitive metric, that is influenced by factors that have little to do with the quality of neural representations such as population size and measurements noise of the somatic

calcium signals. Finally, the decoding uses activity from the entire trial whereas the behavioral data (Supp. Fig. 2) show animals discriminate between environments at 1/6th of the trial, making the parallel to the behavior very weak.

We have added a statement to clarify our motivation to use the decoding (classification/prediction based on the linear model) methods, which are a common approach (for a recent example, see doi: 10.1093/cercor/bhy292) for analysis in the field (lines 157-164):

“To determine how the context (parameters of the virtual environment) and the context-value associations (specific context linked to reward, no-reward, or neutral value associations) were represented in this heterogeneous population activity in the RSC, we examined the ability to decode the individual contexts (baseline day 1) and the change in context-value associations formed with learning (day 4) and after reversal learning (day 7). We use decoding accuracy as a metric to quantify the extent to which external and task-related variables are represented by neuronal population activity within the RSC.”

The decoding uses the activity from the entire trial to capture the temporal dynamics that the reviewer criticised were not captured in point 11.

24. 195-196 *Next, we examined neuronal activity within the ITI period. Although animals no longer saw the context-specific visual stimulus during this time interval, it is possible that the...*

This section is problematic. The finding is that reward can be decoded from RSC activity seconds after it was delivered. Because visual environment and reward were not independently manipulated it is not possible to determine whether the conjunction of reward and visual context is encoded.

This has been addressed in point 1 as well as in relation to point 22. We have made amendments to this section of the text for clarity (lines 203-227).

25. 200-202... *rewarded contexts, in relation to the neutral context. Here the activity of individual neurons was similar for the non-rewarded and neutral contexts, and differed for the rewarded context (Fig. 2g); this was reflective of the behavioral changes in running speed observed across...*

As for Fig. 2f, similarity of responses is not quantified, and sorted responses to ‘neutral’ context cannot be visually compared to responses to ‘rewarded’ and ‘non-rewarded’ contexts without the use of cross-validation.

We have added the cross-validation analysis as well as cross-correlation quantification (see also points 3, 11, 22).

26. 205-206 *Accordingly, we also found a higher decoding accuracy for the rewarded context (Fig. 2j) as well as a significantly higher proportion of context-encoding cells (Fig. 2k), following both...*

Similar issue with the use of decoding accuracy in the VR corridor. Why choosing it and how should this metric be interpreted? What is its meaning?

The utilization of decoding techniques is widely used in the field of calcium imaging - this is not unique to our manuscript and is well established. This analysis provides a metric to quantify the extent to which external and task-related variables can be encoded in neuronal populations and individual neurons. These established computational tools are used to predict animal behavior merely using the information from neuronal activity and are essential in understanding the role of neuronal activity in behavior, especially in awake-behaving experimental paradigms that can often be difficult to interpret due to the high dimensionality of data. Here, the success rate of predictions (or accuracy) can be used as a metric of understanding of a given system. We have added a short justification along these lines in the manuscript (lines 157-16, see also points 1, and 3 above).

“To determine how the context (parameters of the virtual environment) and the context-value associations (specific context linked to reward, no-reward, or neutral value associations) were represented in this heterogeneous population activity in the RSC, we examined the ability to decode the individual contexts (baseline day 1) and the change in context-value associations formed with learning (day 4) and after reversal learning (day 7). We use decoding accuracy as a metric to quantify the extent to which external and task-related variables are represented by neuronal population activity within the RSC.”

Reviewer #2 (Remarks to the Author):

The authors are to be applauded for this revision - the MS is clearly much improved. Sun et al. have included a range of different new analyses (up and foremost longitudinally matched single cell data), flipped the flow of argument (now closing with the behavior data), and provide a much more detailed and balanced discussion of their results in the context of previous work, including very recent findings. This is a significant effort that clearly payed off since the main message of the paper - further evidence for a role of RSC in value coding - is now supported by both the functional and behavioral data which now appear much less disjunct than in the first submission. Of course: It remains unclear how exactly an increase in multidimensional neurons (now nicely supported with longitudinal data) should lead to context devaluation upon reversal learning - but this is now much more carefully discussed and also clearly beyond the scope of a single study. Also the finding of ITI-coding for a preceding reward, one of my criticisms of the first version, is now more carefully interpreted and factors like reward-related ongoing neuromodulation are acknowledged. The presented data is convincing that simple "bleed-through" of reward-locked activity is not leading to this difference.

We thank the reviewer for very positive comments and the recognition of the major improvements that were made with our previous revision efforts. We have addressed all remaining minor comments below.

My remaining points are all minor.

1. The authors now include longitudinal data (which I know was a significant effort!). However, it would be nice to see a some quantification of similarity for the qualitative sorted matched-cell plots in Fig. 2fg and 4ge (and the respective supplements). Simple population vector correlation matrices across contexts over learning phases in different blocks would suffice. Also: Did I miss the quantification of the (lack of) amplitude differences over time which is strongly referred to in the results (L210-212)?

As suggested, we have added additional quantification along these lines, now including the cross-correlation values in the main Figure 2, 4 as well as additional Supplementary Figures with correlation matrix presented and quantification and statistics of these results across all animals (Supplementary Figs. 6, 7, 9, and 11). We feel as though this has significantly strengthened our main point here. The lack of amplitude differences is indeed represented at the base of the heatmaps in Figure 2e, f as connected line plots for each animal and as p-values in the supplementary table, but we have also now included an indication of the non-significant p values in the text as well (line 155-156 Fig. 4e day 4, $P=0.958$, day 7, $P=0.088$; Fig. 4f day 4, $P=0.055$, day7, $P=0.117$).

2. These sorted longitudinal data figures would also be more telling if the sorting would be done cross-validated by using half of the trials and using these indices for all conditions. This would reveal better how much of the dissimilarity is trial-to-trial fluctuation and how much is showing representational differences. This is, for instance, relevant for supplementary Figure 5: To me it rather looks as if the ITI activity is much more similar to the activity in the reward epoch than in the corridor itself. This is, of course, expected because only the corridor has visual stimulation. But to make the point that ITI activity and reward activity are very different (L216,217) one should also take trial to trial fluctuations into account (i.e.: how similar are subsequent reward epochs). Simple population vector correlation matrices across reward, corridor and ITI trials again could make this measure quantitative.

We have provided the suggested new analysis in this regard and included cross-validation using odd and even trials as well as correlation matrices (Supplementary Figs 4, 9, 10, and 12). While the correlation between the reward and ITI periods are higher than the correlation between the corridor (as expected since these values are also averaged across different time scales and highly temporally separated), these correlation values are still significantly lower than those observed across contexts for instance (Supplementary Figures S6, S7, S9)

3. It should be clearly stated in the legend to fig 1 and results that the treadmill is blocked in the reward zone already - and not only in the ITI.

We have made this point more explicit in the Figure legend and results (lines 113-116) in addition to the previous mention in the methods.

E.g. Results: "Next, the reward period, where external breaks were applied and therefore the treadmill was blocked so all animals were stationary, the reward spout was extended for all trials,..."

4. Figure 3 e,j - please use "single context", "two contexts" and "three contexts" - it took me a bit to see that not the standard context index (ctx 1, 2 3 etc.) is used here.

We have made this change as requested.

5. In the results and discussion there are several remarks towards "efficient encoding" of task variables (i.e. L280). I am not certain if evoking Horace Barlow was in the intention of the authors - but given that no quantification of common 'efficient coding' parameters like sparsity etc. are given, the authors should probably refrain from using this term to not lead readers onto the wrong path.

Thank you for the suggestion, we have reworded the instances where this term was used to either remove the 'efficiently' designation, or in one instance (discussion line 433-434 changed this to 'more accurately' encoded).

Reviewer #3 (Remarks to the Author):

It was a little hard to track to the changes in the manuscript at times because the line numbers given in the rebuttal letter did not seem to match the manuscript at times, but perhaps the authors were not at fault for this and regardless I think I was able to track everything down. All of my original concerns were addressed and I have no further or new concerns.

We apologize for the lack of specific tracked changes in the last revision, it became difficult since the manuscript flow was significantly revised. However, we have tracked specific changes in the current revision and thank the reviewer for taking the time to re-evaluate our submission and for the positive comments.

Reviewers' comments:

Reviewer #2 (Remarks to the Author):

Concerning my own review, I am happy with the changes made. I am not opposed to publication. However, the recent pre-print of the Goard lab on the role of RSC in contextual processing (Franco & Goard 2021, <https://www.biorxiv.org/content/10.1101/2020.12.20.423684v1>) should be cited and discussed.

Reviewer #1 has withdrawn from the review process. I have been asked to evaluate this reviewer's comments and assess if the replies of the authors would address her/his points. I find it rather unfortunate that reviewer 1# made the decision to not comment on the responses to her/his often very insightful and elaborate points her/himself.

Even though I share many of the concerns of this reviewer, as should have become clear in my first review, I still consider the MS improved and more carefully interpreted. To sum up my view of the paper: Its strength is the behavioral data on the role of RSC in appetitive contextual learning - although a more context-independent motor behavior (i.e. licking instead of reward-anticipatory slow-down) as readout would have been preferable. The main problem of the paper is the tenuous link between the neuronal and behavioral data and the (acknowledged) difficulty to disentangle contextual variables like movement, position, visual input, and reward.

Addressing the specific points of reviewer 1:

Main point 1:

This point regards the aforementioned difficulty of disentangling contextual variables due to the limits of the experimental design.

I share the general concerns of reviewer #1. However, I do think that the authors now sufficiently clarify that context/reward associations are only separable across reversal of reward-contingency and not within-day.

I still consider it unfortunate that slow-down in the AZ was used as a metric (as opposed to licking). Given that slower running in the rewarded context is visible even outside the AZ, this means that it is impossible to say how much of the activity in RSCdys that is specific for rewarded contexts is simply driven by differences in self-motion.

Main point 2:

In this point reviewer #1 criticizes the weak link between AZ speed and learning.

Here, I tend to agree with the authors. Figure 2a shows a fairly clear reduction in AZ speed in the rewarded and newly rewarded contexts in comparison to the non-rewarded contexts. I therefore do not fully understand this point.

Main point 3:

Here reviewer #1 raises general points regarding the analysis of RSC representations.

I find it a bit difficult to comment on this since the points remain vague. Fig. 2-3 indeed addresses learning and decoding accuracy is indeed a rather standard measure.

Main point 4

This point regards the language used.

Also here I find it difficult to comment. I agree that the paper is no easy read and that many of the links between neural activity and behavior remain highly speculative (e.g. the role of mixed selectivity in cognitive flexibility). However, the reviewer remains too vague for me to assess her/his specific concerns.

Main point 5

This point regards the choice of running speed in the AZ as a behavioral readout.

I agree with the authors. I do not see a major problem in using AZ speed as a learning readout - but would have preferred licking.

Main point 6

This point regards a lack of precision in the description of the behavioral changes.

The description in text and depiction in figures is clearer now.

Main point 7

Effectively this point mirrors point 1

I agree with the authors that the contingency switch during reversal learning partially addresses this issue.

Main point 8

Space allocated to the analysis of visual responses too much

I agree with the authors. I do not think that this is overly excessive and consider it relevant to discuss this.

Main point 9

Problems with visual language.

I find the visual presentation improved over the first version. However, also I do not understand why rewarded contexts are not marked as such in the bar graphs throughout.

Main point 10

Point: Increased decoding accuracy after reward in ITI is not surprising.

This echoes one of my previous points - I agree with this. Post-reward activity, even when not directly reflecting reward-consumption motor behavior will induce a lingering state-change that is decodable during ITIs. This is indeed not very surprising and to call this post-reward activity "context-value association" without further controls is still a bit of a stretch. One way to address if the activity is indeed specific for the learned association of context and reward may potentially be to compare the first learning and reversal learning trials of ITI activity (i.e. during "Learning" not during "Test" sessions) and check if RSC activity is already discriminative for rewarded contexts. If yes, this would point towards simple, non-associative reward-related state changes.

Main point 11

Problems with data presentation by sorted colormaps.

I find this unproblematic. Whereas it is an important point that sorted maps and visible patterns therein should be treated with caution, I find this here much less of an issue as, for instance, in other publications where a sequential activity 'snake' over time is shown. Here, the only point made is the visualization of similarity between non-rewarded and neutral contexts during ITIs. Using single PV

correlation values as similarity metric is, admittedly, simplifying - but certainly standard in the field.

Main point 12
Standard errors.

The authors have addressed the reviewer's concern.

Main point 13
Plotting of data points.

I agree - but the authors have done this now.

Main point 14
Done

Main point 15
Done.

Main point 16
This seems addressed.

Main point 17
This seems addressed.

Main point 18
This seems addressed.

Main point 19
The description of the data is now less interpretive.

Main point 20
ok

Main point 21
ok

Main point 22
It is very difficult to assess whether or not the changes address the reviewer's concerns. I refrain from doing so.

Main point 23
Even though not unwarranted, these are very general points made against decoding accuracy as a metric that could be made against very many studies these days. I think this is fine in this context.

Main point 24
As stated above, I fully agree with this point. I don't think this is corrected in the MS.

Main point 25
This is addressed

Main point 26
See above.

Reviewer #2 (Remarks to the Author):

Concerning my own review, I am happy with the changes made. I am not opposed to publication. However, the recent pre-print of the Goard lab on the role of RSC in contextual processing (Franco & Goard 2021, <https://www.biorxiv.org/content/10.1101/2020.12.20.423684v1>) should be cited and discussed.

We have added reference to this interesting new study into the manuscript (e.g. discussion lines 467, 487-489, highlighted in red in the original Word file; the numbering may be slightly changed by the submission system generating the merged PDF).

Reviewer #1 has withdrawn from the review process. I have been asked to evaluate this reviewer's comments and assess if the replies of the authors would address her/his points. I find it rather unfortunate that reviewer 1# made the decision to not comment on the responses to her/his often very insightful and elaborate points her/himself.

Even though I share many of the concerns of this reviewer, as should have become clear in my first review, I still consider the MS improved and more carefully interpreted. To sum up my view of the paper: Its strength is the behavioral data on the role of RSC in appetitive contextual learning - although a more context-independent motor behavior (i.e. licking instead of reward-anticipatory slow-down) as readout would have been preferable. The main problem of the paper is the tenuous link between the neuronal and behavioral data and the (acknowledged) difficulty to disentangle contextual variables like movement, position, visual input, and reward.

We thank the reviewer for taking the time and effort to evaluate our manuscript again and appreciate the positive comments and suggestions. We have addressed the few remaining points below. Specifically, we have further clarified the responses in the ITI period to the best of our ability and strengthened the link between our neuronal and behavioural data (see point 10). We have also further clarified the limitations of the study in the discussion, edited for increased readability wherever possible, and added the reward symbols to the bar plots throughout. Although future studies should address the specific nature of the contextual variables, as the reviewer suggests, we agree that our manuscript provides valuable insight into the role of the RSC in contextual learning. We feel that the paper has been substantially improved with all the comments and suggestions from the reviewers who have taken their time to improve the presentation of our results. We hope the paper is now ready for publication.

Addressing the specific points of reviewer 1:

Main point 1:

This point regards the aforementioned difficulty of disentangling contextual variables due to the limits of the experimental design.

I share the general concerns of reviewer #1. However, I do think that the authors now sufficiently clarify that context/reward associations are only separable across reversal of reward-contingency and not within-day.

I still consider it unfortunate that slow-down in the AZ was used as a metric (as opposed to licking). Given that slower running in the rewarded context is visible even outside the AZ, this means that it is impossible to say how much of the activity in RSCdys that is specific for rewarded contexts is simply driven by differences in self-motion.

We are glad that the reviewer agrees that these contingencies are now clarified and the confounds appropriately addressed in the revised manuscript. We have additionally even further expanded on the confound of movement in the context of point 10 and 24 below and strengthened our link between the

neuronal activity and our behavioural metric.

Main point 2:

In this point reviewer #1 criticizes the weak link between AZ speed and learning.

Here, I tend to agree with the authors. Figure 2a shows a fairly clear reduction in AZ speed in the rewarded and newly rewarded contexts in comparison to the non-rewarded contexts. I therefore do not fully understand this point.

We also believe that there was some initial confusion on the part of reviewer 1 and are reassured that, in the revised manuscript, reviewer 2 agrees that this is clearly demonstrated.

Main point 3:

Here reviewer #1 raises general points regarding the analysis of RSC representations.

I find it a bit difficult to comment on this since the points remain vague. Fig. 2-3 indeed addresses learning and decoding accuracy is indeed a rather standard measure.

We agree that this is a standard measure in the field and feel that it is indeed appropriate for the study.

Main point 4

This point regards the language used.

Also here I find it difficult to comment. I agree that the paper is no easy read and that many of the links between neural activity and behavior remain highly speculative (e.g. the role of mixed selectivity in cognitive flexibility). However, the reviewer remains too vague for me to assess her/his specific concerns.

We have again gone over the manuscript to try to increase the ease of reading as much as possible and made small changes throughout. Perhaps these help to improve readability even further.

Main point 5

This point regards the choice of running speed in the AZ as a behavioral readout.

I agree with the authors. I do not see a major problem in using AZ speed as a learning readout - but would have preferred licking.

We have noted this for future experiments and already confirmed major findings with licking data in new project data and will have both measures going forward. We would additionally reiterate that while licking behaviour is indeed already an established metric in this regard, our speed in the anticipation zone may also be used in comparison to experiments with freely moving animals for instance or other diverse paradigms where licking behaviour is not possible to have as a metric. At least we feel it may be useful that this is now established as an additional option and has been shown to be a robust readout in this study.

Main point 6

This point regards a lack of precision in the description of the behavioral changes.

The description in text and depiction in figures is clearer now.

Excellent.

Main point 7

Effectively this point mirrors point 1

I agree with the authors that the contingency switch during reversal learning partially addresses this issue.

Great.

Main point 8

Space allocated to the analysis of visual responses too much

I agree with the authors. I do not think that this is overly excessive and consider it relevant to discuss this.

Great, we feel strongly about this point and are glad that reviewer 2 agrees.

Main point 9

Problems with visual language.

I find the visual presentation improved over the first version. However, also I do not understand why rewarded contexts are not marked as such in the bar graphs throughout.

We are glad the reviewer agrees that the presentation was improved. As suggested, we have further added the rewarded context on the bar graphs throughout.

Main point 10

Point: Increased decoding accuracy after reward in ITI is not surprising.

This echoes one of my previous points - I agree with this. Post-reward activity, even when not directly reflecting reward-consumption motor behavior will induce a lingering state-change that is decodable during ITIs. This is indeed not very surprising and to call this post-reward activity "context-value association" without further controls is still a bit of a stretch. One way to address if the activity is indeed specific for the learned association of context and reward may potentially be to compare the first learning and reversal learning trials of ITI activity (i.e. during "Learning" not during "Test" sessions) and check if RSC activity is already discriminative for rewarded contexts. If yes, this would point towards simple, non-associative reward-related state changes.

Here we have added new data analysis and a new main Figure (new Figure 3) that we feel addresses this point to the best of our ability and simultaneously also provides a stronger link between our neuronal and behavioural data. It was not possible to compare the neuronal activity of the first learning trials during the 'Learning' phase as this was not an imaging day in our protocol. However, on Test day 4, the animals still showed a significant learning curve across trials according to our behavioural metric, as the difference in speed between the early trials was much smaller than during the later trials in that session (this is clear in our new Figure 3c). Therefore, we compared the decoding accuracy based on the neuronal activity within the ITI period for these trials with a small overall difference in speed with the trials with the largest difference in speed during this session. Importantly, we found that the trials where the mice did not show behavioural discrimination between contexts (i.e. had no substantial difference in speed between contexts) showed no significant difference in decoding accuracy based on the neural activity in the ITI period. However, for the trials where there was a large difference in speed in the anticipation zone, the rewarded context had significantly higher decoding accuracy compared to the non-rewarded and neutral contexts (Figure 3d). This new data analysis makes two main points: 1) since the animals received rewards in context 1 for all trials, the effect we see in the decoding accuracy is not *only* a reflection of the reward alone, and 2) when the animals showed a larger difference in our behavioural metric (better 'performance'),

the underlying RSC activity in the ITI period had increased decoding accuracy in relation to the other contexts (see lines 268-284).

These new findings do not entirely exclude that there may be behavioural state-related changes following the reward, but we would argue that these changes are related to the learning process and likely contribute to the discriminability between the context-value associations. We have also clarified these details further in the discussion (line 493-495).

Interestingly, on day 7, the animals did not show the same type of learning curve (Figure 3c), presumably as they already had more prior experience with the task, and here we did not see significant changes in decoder accuracy between trials (Figure 3d) - indicating that if the behavioural metric was dissociable across contexts, the RSC neuronal activity consistently resulted in significantly higher decoding accuracy for the rewarded context.

Main point 11

Problems with data presentation by sorted colormaps.

I find this unproblematic. Whereas it is an important point that sorted maps and visible patterns therein should be treated with caution, I find this here much less of an issue as, for instance, in other publications where a sequential activity 'snake' over time is shown. Here, the only point made is the visualization of similarity between non-rewarded and neutral contexts during ITIs. Using single PV correlation values as similarity metric is, admittedly, simplifying - but certainly standard in the field. Great, we are glad that reviewer 2 agrees.

Main point 12

Standard errors.

The authors have addressed the reviewer's concern.

Great.

Main point 13

Plotting of data points.

I agree - but the authors have done this now.

Great.

Main point 14

Done

Great.

Main point 15

Done.

Great.

Main point 16

This seems addressed.

Great.

Main point 17

This seems addressed.

Great.

Main point 18
This seems addressed.
Great.

Main point 19
The description of the data is now less interpretive.
Great.

Main point 20
ok
Great.

Main point 21
ok
Great.

Main point 22
It is very difficult to assess whether or not the changes address the reviewer's concerns. I refrain from doing so.

Here, we can only reiterate that we added extensive additional quantification regarding the pattern of responses using cross-correlation and additional cross-validation methods in the last revision, but indeed the original reviewer's concerns were unclear and so we addressed them to the best of our ability.

Main point 23
Even though not unwarranted, these are very general points made against decoding accuracy as a metric that could be made against very many studies these days. I think this is fine in this context.
We agree.

Main point 24
As stated above, I fully agree with this point. I don't think this is corrected in the MS.
Please see response to point 10.

Main point 25
This is addressed
Great.

Main point 26
See above.

Indeed, related to point 23 and has been addressed.

Reviewers' comments:

Reviewer #2 (Remarks to the Author):

The authors have addressed my concerns, pending some remaining minor suggestions - I am looking forward to seeing the paper published:

With their new analysis and figure, the authors provided reasonable evidence that a reward-consumption state change alone is unlikely to be the sole reason that activity during the ITI contains information about the previous rewarded context: They show that early trials during learning with indiscriminate AZ speed also have indiscriminate context decodability, whereas in later trials context-dependent AZ slow-down is associated with information about the rewarded context in ITIs - even though the reward was given in all reward-context trials. This does not exclude that different behavior during movement in the rewarded context (i.e., reward-anticipatory slow-down) or ITI motion-attempts with blocked-treadmill (i.e., running attempts in non-rewarded context) are leading to decodable activity, but I agree that this analysis is an important step forward to disentangling these issues and the data is described with sufficient care (L287-289).

I have a few remaining minor points :

- I have a bit of trouble understanding the new text describing the new figure 3.

1. The authors write, "Further, when we used deconvolution methods in the ITI to remove potential effects of the remnant calcium decay signal, the pattern of results in the ITI period was unaltered (Fig. 3...)" L272-274.

How does the new figure 3 show an "unaltered pattern of results"? What is meant by that? Also, is data in Fig 3 deconvolved but not in Fig 2? Why is that not written anywhere?

2. The authors then write "The activity level of neurons within the reward period does not necessarily correlate to that within the ITI period; Fig. 3a, 3b, Supplementary Figs. 9, 10)" L275-276

Fig. 3 a shows that reward correlates best with IT period - the later the trials the better 3b - so what is this sentence supposed to mean? Exactly that? If yes, it is cryptic. Also, the punctuation in this sentence is off.

- This came up a few times during the review, but even though I have seen the figures now many times I still sometimes struggle to understand the panels without extensive re-reading of main text and captions. It would be helpful to use the axis labels to more precisely describe what exactly is plotted. For example: When applicable use "AZ speed" instead of "speed" only, "ITI decoding accuracy" instead of "decoding accuracy" etc.

- This is, admittedly, partially the result of a long review process, but the paper is rather complex already and adding further figure panels will make it even less digestible. Therefore, even though I like the analysis of the new Fig3 I suggest to attempt to join Fig2 and 3, moving some of their content to supplement. For instance current 2c, 2b, 2, 3b could be moved out maybe 3a associated with current 2f and 3c,d moved below 2g-j. But I would leave this to the authors.

- Please indicate the AZ in the task sketch in 2d

REVIEWERS' COMMENTS

Reviewer #2 (Remarks to the Author):

The authors have addressed my concerns, pending some remaining minor suggestions - I am looking forward to seeing the paper published:

With their new analysis and figure, the authors provided reasonable evidence that a reward-consumption state change alone is unlikely to be the sole reason that activity during the ITI contains information about the previous rewarded context: They show that early trials during learning with indiscriminate AZ speed also have indiscriminate context decodability, whereas in later trials context-dependent AZ slow-down is associated with information about the rewarded context in ITIs - even though the reward was given in all reward-context trials. This does not exclude that different behavior during movement in the rewarded context (i.e., reward-anticipatory slow-down) or ITI motion-attempts with blocked-treadmill (i.e., running attempts in non-rewarded context) are leading to decodable activity, but I agree that this analysis is an important step forward to disentangling these issues and the data is described with sufficient care (L287-289).

We thank the reviewer again for their time, positive comments and helpful suggestions! All remaining minor points are addressed below.

I have a few remaining minor points :

- I have a bit of trouble understanding the new text describing the new figure 3.

1. The authors write, "Further, when we used deconvolution methods in the ITI to remove potential effects of the remnant calcium decay signal, the pattern of results in the ITI period was unaltered (Fig. 3...)" L272-274.

How does the new figure 3 show an "unaltered pattern of results"? What is meant by that? Also, is data in Fig 3 deconvolved but not in Fig 2? Why is that not written anywhere?

The statement was meant in reference to the non-deconvolved data, such that the pattern of results is comparable whether data is deconvolved or not. Note, this is also the reason why the main results of the manuscript remained as the non-deconvolved data because this was not consequential to the conclusions - this was made clear and addressed in comment 1.3b of review 1, during the first round of review. This indicates that the activity in the ITI period is not just a reflection of the activity during the preceding reward period and the pattern of results kept the same even after deconvolution to remove potentially conflicting decay-time signals leftover from the reward zone, which was the concern of the reviewer's original comment. We have reworded this sentence to clarify this point: "Further, when we used deconvolution methods in the ITI to remove potential effects of the remnant calcium decay signal (Fig. 3; Supplementary Figs. 10, 11), the pattern of results in the ITI period was consistent with the non-deconvolved data." (Lines 270-272).

The main figures are all with the non-deconvolved data. The deconvolved data was originally added as a supplementary figure to address a reviewer's comment (point 10 from review 2, during the 3rd round review). This was noted in the manuscript as above: "Further, when we used deconvolution methods in the ITI to remove potential effects of the remnant calcium

decay signal...(Supplementary Figs. 10, 11)", as well as in the appropriate figure legends. The new Figure 3 was added to address the concerns about the potential contamination of the reward period into the ITI and this is why we again chose to add Figure 3 with the deconvolved data (this is clearly indicated in lines 270-272: "...we used deconvolution methods in the ITI to remove potential effects of the remnant calcium decay signal (Fig. 3f Supplementary Figs. 10, 11)").

We have now also added a clarification for each Figure used with deconvolved data in the methods: "To extract the neuronal change in fluorescence, we used a method that automatically identified ROIs (including spatially overlapped ones), de-noised signals, and when comparing the ITI period to the immediately preceding reward period (Fig. 3f Supplementary Figs. 10, 11), deconvolved signals⁸⁷ with open source and adapted MATLAB code (MathWorks, MA, USA)." (Lines 740-744).

2. The authors then write "The activity level of neurons within the reward period does not necessarily correlate to that within the ITI period; Fig. 3a, 3b, Supplementary Figs. 9, 10)" L275-276

Fig. 3 a shows that reward correlates best with IT period - the later the trials the better 3b - so what is this sentence supposed to mean? Exactly that? If yes, it is cryptic. Also, the punctuation in this sentence is off.

We have fixed the punctuation in this sentence (bracket missing), and also clarified this sentence to read: "We sorted the average activity of each neuron according to their responses during the reward period and found that the activity within the reward period is only moderately correlated to that within the ITI period, and only increases in the later trials (Supplementary Figs. 9, 10)." (Lines 272-275).

- This came up a few times during the review, but even though I have seen the figures now many times I still sometimes struggle to understand the panels without extensive re-reading of main text and captions. It would be helpful to use the axis labels to more precisely describe what exactly is plotted. For example: When applicable use "AZ speed" instead of "speed" only, "ITI decoding accuracy" instead of "decoding accuracy" etc.

We have made additional efforts to improve this throughout all Figures as the reviewers suggested (e.g. see Figures. 2a-2c, 3a-3f, 5c, 5e-5h, 6a-6e and Supplementary Figures. 1b, 2b-2c, 10b, 12a-12b, 16, 17, and 18a-18h edited axis labels, etc).

- This is, admittedly, partially the result of a long review process, but the paper is rather complex already and adding further figure panels will make it even less digestible. Therefore, even though I like the analysis of the new Fig3 I suggest to attempt to join Fig2 and 3, moving some of their content to supplement. For instance current 2c, 2b, 2, 3b could be moved out maybe 3a associated with current 2f and 3c,d moved below 2g-j. But I would leave this to the authors.

Indeed, we have often found it difficult with conflicting requests from reviewers. For instance, Figure 2b was specifically requested to be moved from supplementary to the main Figure

(comment 5, review 1, during the 2nd round of review). With all this in mind, we agree with the reviewer that we could merge parts of Figure 2 and 3, however to keep Figure sizes manageable, we have kept both as main Figures. We have therefore moved panels 2g-j to Figure 3, since these panels all represent the decoding accuracy measures and, as the reviewer suggests, tend to go together. We have also moved Figure panels 3a, b to supplementary as suggested. We believe this has improved the readability of both Figures.

- Please indicate the AZ in the task sketch in 2d

This has been added to Figure 2d.